# Body orientation change of neighbors leads to scale-free correlation in collective motion

Collective motion, such as milling, flocking, and collective turning, is a common and captivating phenomenon in nature, which arises in a group of many self-propelled individuals using local interaction mechanisms. Recently, vision-based mechanisms, which establish the relationship between visual inputs and motion decisions, have been applied to model and better understand the emergence of collective motion. However, previous studies often characterize the visual input as a transient Boolean-like sensory stream, which makes it challenging to capture the salient movements of neighbors. This further hinders the onset of the collective response in vision-based mechanisms and increases demands on visual sensing devices in robotic swarms. An explicit and context-related visual cue serving as the sensory input for decision-making in vision-based mechanisms is still lacking. Here, we hypothesize that body orientation change (BOC) is a significant visual cue characterizing the motion salience of neighbors, facilitating the emergence of the collective response. To test our hypothesis, we reveal the significant role of BOC during collective U-turn behaviors in fish schools by reconstructing scenes from the view of individual fish. We find that an individual with the larger BOC often takes on the leading role during U-turns. To further explore this empirical finding, we build a pairwise interaction mechanism on the basis of the BOC. Then, we conduct experiments of collective spin and collective turn with a real-time physics simulator to investigate the dynamics of information transfer in BOC-based interaction and further validate its effectiveness on 50 real miniature swarm robots. The experimental results show that BOC-based interaction not only facilitates the directional information transfer within the group but also leads to scale-free correlation within the swarm. Our study highlights the practicability of interaction governed by the neighbor's body orientation change in swarm robotics and the effect of scale-free correlation in enhancing collective response.

Large-scale collective motion is a widely observed phenomenon in biological systems, e.g., flocks of starlings[1], schools of zebrafish[2], and herds of sheep[3]. The mechanism underlying collective motion has been extensively studied, offering valuable insights for fields ranging from ecology to physics and engineering[4–8]. It is well known that such spectacular phenomena emerge from individual-level interactions within a local neighborhood. Extensive studies have been conducted to reveal interaction mechanisms through computational theories and handcrafted designs. For example, the metric interaction[7–12], topological interaction[1,13], and selective interaction[14–16].

✉ e-mail: pxg@nwpu.edu.cn

Despite the explanatory success at the emergence of collective motion, the proposed metric, topological, or selective interaction mechanism primarily reproduces the collective motion in a phenomenological manner. These interaction mechanisms often rely on physical measurements that may not be directly available from the first-person perspective of the animal[8,9,14]. Beyond these phenomenological interaction mechanisms, the vision-based modeling paradigm has recently become popular in current research[2,17–20], as these models are more biologically plausible in mimicking how animals perceive and react to their neighbors. Plenty of studies have achieved collective motion by simply relating visual inputs and movement decisions[20–24]. However, in previous studies, the complex process of visual perception was often simplified to a transient binary visual field for decision-making[20,22,24]. On the one hand, it is difficult to extract meaningful visual cues from simplified visual information, limiting the ability to capture the salient motions of neighbors. This further poses a gap between vision-based models and the emergence of collective response, which is essential for many collective tasks such as collective anti-predation[7] and collective avoidance[25]. On the other hand, relying on a binary visual field presents practical difficulties for real-world applications in swarm robotics, as it is challenging to achieve with onboard visual sensing devices. There is a pressing need to bridge this gap by uncovering an explicit and context-related visual cue as the sensory input to facilitate the emergence of collective response in vision-based swarm models.

In this study, we aimed at revealing what visual cue individuals adopt from the first-person perspective to select their interacting neighbors during collective responses. Inspired by the theory success of body orientation (or attitude) coordination investigated in macro-level swarm models[26,27], we hypothesized that body orientation change (BOC) might be a significant visual cue that could effectively capture the swift maneuvers made by neighbors in response to the external stimuli, i.e., the motion salience of neighbors. Since the BOC of neighbors directly leads to variations in the projections of their body shape on the focal individual's retina, reflecting the real-time adjustments in the movement of neighbors.

To test this hypothesis, we analyzed experimental data from spontaneous U-turn behavior in groups of rummy-nose tetra (*Hemigrammus rhodostomus*)[16,28,29], aiming to uncover the significance of BOC in achieving the collective response. First, we introduced a quantitative method to characterize the BOC of neighbors by computationally reconstructing the scene from a fish's perspective. Second, we determined the leadership of each individual during U-turns by leveraging a state-of-the-art leader-follower relationship analysis method[30]. Third, we revealed the relationship between the leadership of individuals and their BOC-based motion salience by conducting a Spearman correlation analysis. Interestingly, the analysis of real fish data revealed that an individual fish with the larger BOC tends to lead other fish during the U-turn behaviors. Moreover, to gain deeper insights into this empirical finding, we modeled a pairwise interaction mechanism based on the BOC and thoroughly investigated the information transfer dynamics governed by this bio-inspired mechanism. Through extensive experiments of collective spin and collective turn, we found that BOC-based interaction not only facilitates the directional information transfer within the group but also provides each individual with an effective interaction range much larger than the perception range, indicating the emergence of scale-free correlation. Notably, scale-free correlation in this study refers to the correlations among the velocity fluctuations of individuals, which has been widely adopted to understand the onset of collective responses in biological swarms[31,32]. Finally, we adopted the BOC-based interaction in a robotic swarm consisting of 50 real mobile robots. The results of robotic experiments further support the advantages and feasibility of BOC-based interaction in swarm robotics, highlighting the ability of BOC to enhance the collective response in real-world applications.

## Results

### Quantifying body orientation change to measure motion salience of neighbors

Previous research has revealed that the directional decisions of individuals in biological swarms are influenced by neighbors with salient movement changes[33–35]. However, the way individuals assess the salience of a neighbor's movements from the first-person visual perception remains not fully elucidated. Here, we introduce a direct observational visual cue, i.e., body orientation change (BOC) to measure the motion salience of neighbors from an individual's own view.

To quantify the BOC of neighbors, we first characterize each individual as a non-transparent ellipse with lengths of major axis $a$ and minor axis $b$ (Fig. 1a), which suggests that neighbors near the focal individual may occlude neighbors that are further away (detailed approaches for identifying the occluded neighbor are presented in Supplementary Note 4 and Supplementary Fig. 15). Then, we reconstruct the visual field of each elliptical individual by using the same computational method in Ref. 36 (see Supplementary Note 4 for the detailed computation method). Specifically, the visual field of each individual is defined as the triangular region formed by connecting the focal fish's eye to two points where the rays emanating from its eye are tangent to the edges of each elliptical neighbor. For example, as shown in Fig. 1a, the tangent points on the elliptical neighbor $j$ observed from the focal individual's eye at times $t-1$ and $t$ are denoted as $[p_1 = (x_1, y_1), p_2 = (x_2, y_2)]$ and $[p_3 = (x_3, y_3), p_4 = (x_4, y_4)]$, respectively. As a result, the visual fields of individual $i$ observing neighbor $j$ at times $t-1$ and $t$ are the regions shaded in yellow and orange, respectively. Next, we calculate the distance between pairs of tangent points at times $t-1$ and $t$, denoted as the $\beta_j(t-1) = \sqrt{(x_1 - x_2)^2 + (y_1 - y_2)^2}$ and $\beta_j(t) = \sqrt{(x_3 - x_4)^2 + (y_3 - y_4)^2}$.

Moreover, we consider the variation between these two distances as the transient measure of the neighbor's body orientation change, i.e., $|\beta_j(t) - \beta_j(t-1)|$. Finally, we define the accumulation of these transient variations over a period from $T - \tau$ to $T$ as the magnitude of BOC at time $T$, denoted as $g_{ij}(T, \tau)$. The BOC of neighbor $j$ observed by the focal individual $i$ at time $T$ can be calculated as follows:

$$\begin{cases} g_{ij}(T, \tau) = \sum\limits_{t=T-\tau+1}^{T} \frac{|\beta_j(t) - \beta_j(t-1)|}{\Delta t} \cdot \mathrm{fp}(t) \\ \mathrm{fp}(t) = \left( \frac{1 + \hat{\mathbf{v}}_i(t) \cdot \hat{\mathbf{x}}_{ij}(t)}{2} \right)^{\alpha} \end{cases} \quad (1)$$

where $t$ is the time index and $\Delta t$ is the time interval between steps $t-1$ and $t$. $\tau$ represents the time period (or duration) over which individual $i$ considers the past BOC of neighbor $j$. The accumulation of transient BOC over $\tau$ simulates the process by which sensory information is often gathered over a time period before being updated, aligning with findings in biological swarms[37,38].

The term $\mathrm{fp}(t) = \left( \frac{1 + \hat{\mathbf{v}}_i(t) \cdot \hat{\mathbf{x}}_{ij}(t)}{2} \right)^{\alpha}$ in Eq. (1) quantifies the frontal preference of individual $i$ to perceive the movement of neighbor $j$ from the first-person perspective, and the $\mathrm{fp}(t)$ ranges from [0, 1]. $\hat{\mathbf{v}}_i(t)$ is the velocity of individual $i$ and $\hat{\mathbf{x}}_{ij}(t)$ is the relative position vector between individual $i$ and $j$ at time $t$. Here, frontal preference refers to an individual's tendency to interact (or maintain alignment) with its neighbors primarily within its frontal field of view. In the context of perceiving BOC, the frontal preference causes an individual to be more attentive to the BOC of neighbors positioned in front. Specifically, the dot product of $\hat{\mathbf{v}}_i(t) \cdot \hat{\mathbf{x}}_{ij}(t)$ measures the relative bearing of neighbor $j$. For example, for the neighbor-$j_1$ shown in Fig. 1b, $\angle_{(\hat{\mathbf{v}}_i, \hat{\mathbf{x}}_{ij}) = 0}$ signifies that neighbor-$j_1$ is located directly in front of focal individual $i$, and the focal individual could perceive the neighbor-$j_1$ since the $\frac{(1 + \hat{\mathbf{v}}_i \cdot \hat{\mathbf{x}}_{ij})}{2}$ is 1. Once the neighbor $j$ moves from the front to back, i.e., $\angle_{(\hat{\mathbf{v}}_i, \hat{\mathbf{x}}_{ij})}$ goes from 0 to $\pi$ or $-\pi$ (neighbor-$j_2, j_3, j_4$, as shown in Fig. 1b), the ability of

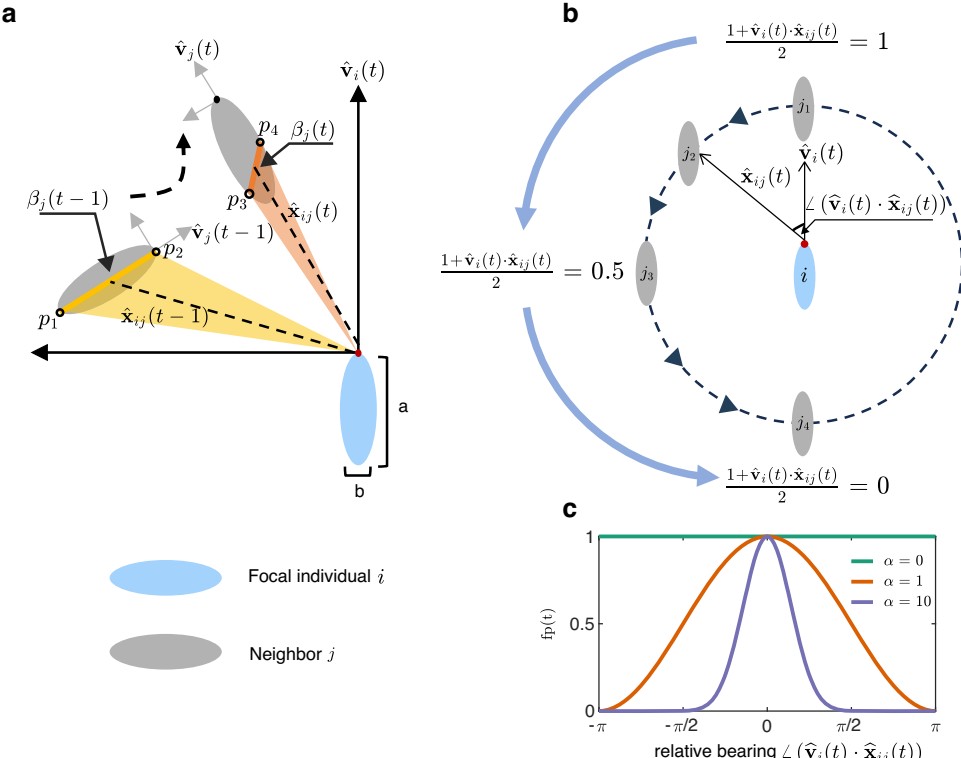

**Fig. 1 | Quantify the BOC of neighbors from the first-person perspective. a** The geometric definition of neighbor-$j$'s body orientation change over two consecutive steps from the view of individual $i$, which is calculated as $g_{ij} = \frac{|\beta_j(t) - \beta_j(t-1)|}{\Delta t} \cdot \mathrm{fp}(t)$, where $\Delta t$ is the time interval between two steps. $\beta_j(t-1)$ is the relative distance between tangent points $p_1$ and $p_2$. $\beta_j(t)$ is the relative distance between tangent points $p_3$ and $p_4$. $\hat{\mathbf{v}}_i(t)$ and $\hat{\mathbf{v}}_j(t)$ are the velocity vectors of individuals $i$ and $j$, respectively. $\hat{\mathbf{x}}_{ij}(t)$ and $\hat{\mathbf{x}}_{ij}(t-1)$ are the relative position vectors between individuals $i$ and $j$ at times $t$ and $t-1$, respectively. The term $\mathrm{fp}(t) = \left(\frac{1+\hat{\mathbf{v}}_i(t)\cdot\hat{\mathbf{x}}_{ij}(t)}{2}\right)^{\alpha}$ is a frontal weighting factor that incorporates the influence of frontal preference in perceiving BOC, where $\alpha$ is an anisotropic parameter used to tune the degree of frontal preference. **b** As the neighbor $j$ moves from the front to the back ($j_1 \rightarrow j_4$), the perception of the neighbor $j$ by focal individual $i$ diminishes as the $\mathrm{fp}(t)$ decreases from 1 to 0 ($\alpha = 1$). **c** Effect of the anisotropic parameter $\alpha$ ($\alpha \geq 1$) on tuning the degree of frontal preference. Each individual's eye is positioned at the front of the ellipse along the major axis (red points drawn on the focal individual). The aspect ratio was chosen as $a/b = 0.1$ in an effort to approximate the shape of the fish in experimental data.

the focal individual $i$ to perceive neighbor $j$ is reduced as the $\mathrm{fp}(t)$ decreases from 1 to 0.

Additionally, $\alpha$ is an anisotropic parameter that tunes the effect of the forward-oriented preference on the perception of BOC. Specifically, when $\alpha = 0$, the focal individual $i$ ignores its forward-oriented preference in perceiving BOC. As $\alpha$ increases ($\alpha \geq 1$), individuals gradually narrow their perception of nearby neighbors towards the front of their visual field. For example, as shown in Fig. 1b, c, if $\alpha = 10$, the perception of BOC decreases to $g_{ij}(T, \tau) \approx 0$ when the relative positions of the neighbors are out of the visual sight $(-\pi/2, \pi/2)$.

**Individuals with larger body orientation changes often take on leading roles during collective U-turns**

To reveal the importance of body orientation change in biological swarms, we analyzed experimental data from spontaneous collective U-turn behaviors observed in rummy-nose tetra groups (*Hemigrammus rhodostomus*) recorded in a circular tank[16,28,29]. In this controlled environment, the fish groups exhibit movement predominantly in two directions: clockwise and counterclockwise. A U-turn is often initiated when a single fish, acting as the leader, abruptly changes its swimming direction. This sudden maneuver serves as a signal for the rest of the group, leading to rapid and cohesive directional changes across the entire fish school.

In this work, we collected a total of 44 and 400 tracks of collective U-turns from groups of 10 and 8 fish, respectively. The details of the U-turn detection method are provided in Supplementary Note 1, and overviews of the U-turn trajectories are shown from Supplementary

Fig. 3 to Supplementary Fig. 8. This analysis is performed with the aim of revealing the relationship between BOC and leadership during U-turn behaviors in fish schools, shedding light on the significant role of BOC in achieving collective responses. Notably, the analysis of experimental data from the fish schools is based on two key assumptions: first, a leader is present during U-turns, typically initiating the abrupt maneuver in the group (see Supplementary Movies 1, 2); second, leadership within the group is associated with certain visual or motion cues of neighbors, as it has been elucidated in many other biological swarms[15,30,33,39–41]. To comprehensively capture the relationship between BOC and leadership in fish schools, we divide each U-turn trajectory into several segments $[T - \tau, T]$ with different combinations of time $T$ and time period $\tau$ ($\tau$ is defined in Eq. (1) and the details of the trajectory segmentation can be found in Supplementary Note 2).

For each segment $[T - \tau, T]$ of a U-turn trajectory, we quantified the BOC of each individual's neighbor from the first-person perspective according to Eq. (1). This further yielded a BOC matrix in which the rows represent the focal individuals (or viewers), and the columns represent their perceived neighbors. The entries in the BOC matrix represent the magnitude of each neighbor's BOC as perceived by the focal individual within a segment $[T - \tau, T]$ (see Methods for the mathematical definition of the BOC matrix). Then, to evaluate how prominently the focal individual's movements are perceived by its neighbors, we averaged each column in the BOC matrix, resulting in a vector known as the BOC-based motion salience that encapsulates the quantification of each individual's degree of salient movement from the perspective of others within $[T - \tau, T]$ (see Methods for more

detailed information of BOC-based motion salience). Next, we quantified the leadership of each individual based on the leader-follower network derived from a segment $[T − τ, T]$ of the U-turn trajectory. In brief, the leader-follower network is essentially an unweighted directed graph, where connections flow from leaders at higher hierarchy to followers at lower (see Methods for the detailed reconstruction of the leader-follower network). Consequently, an individual with more outgoing connections with followers lower in the hierarchy is considered to have stronger leadership within the group (see Methods for the detailed definition of leadership).

By investigating the correlation between the leadership of individuals and their BOC-based motion salience during U-turns within each segment $[T − τ, T]$, we can evaluate the importance of BOC in guiding the group to change its moving direction. For example, according to the U-turn trajectory shown in Fig. 2a, we constructed a BOC matrix (Fig. 2b) and leader-follower network (Fig. 2c) within a period of [2.22 s, 2.82 s]. Then, we obtained the BOC-based motion salience $G_i(T, τ)$ of each individual (Fig. 2d) by calculating the column-wise average of the BOC matrix and normalizing it using the maximum value. Next, we derived each individual's leadership $L_i(T, τ)$ from the leader-follower network (Fig. 2e). Finally, we adopted the Spearman correlation analysis between two vectors comprising $L_i(T, τ)$ and $G_i(T, τ)$. Interestingly, as shown in Fig. 2f, we found a positive correlation between the BOC-based motion salience $G_i(T, τ)$ and leadership of individuals $L_i(T, τ)$ with the involvement of frontal preference ($α = 1$).

Moreover, we extended the Spearman correlation analysis for each U-turn trajectory over different combinations of $T$ and $τ$. As shown in Fig. 2g, we found that in the absence of frontal preference ($α = 0$), the peak of the distribution of the correlation coefficient $ρ$ is close to zero, indicating neither a positive nor a negative correlation between the BOC-based motion salience and leadership. However, when the frontal preference in visual perception ($α ≥ 1$) is introduced, the peak of the distribution of $ρ$ is close to 1, which indicates that individuals with larger BOC often take on leading roles during collective U-turns. Heatmaps of the correlation coefficient $ρ$ with $α = 0$ and $α = 1$ can be found in Supplementary Fig. 11. The prevalence of positive correlation coefficient $ρ$ occurred in the context of frontal preference might find its explanation in the front-to-back transfer of turning information during U-turns (See Supplementary Note 2 and Supplementary Figs. 9, 10 for the spatial distribution of turning orders in fish schools). Although the front fish has a strong influence on the movement direction of individuals during U-turns, we further investigated whether the statistical results presented in Fig. 2g are exclusively attributable to the frontal preference of individuals. Through an examination of the sole correlation between leadership and the effect of frontal preference, i.e., omitting the influence of the neighbor's BOC from Eq. (1) (described as $g_{ij}(T, τ) = \sum_{t=T−τ+1}^{T} \mathrm{fp}(t)$), we found that the peak of $ρ$ distribution is only confined to the range of 0.3–0.5 (see Supplementary Fig. 13 for the distribution of $ρ$ considering only the effect of frontal preference), which provides compelling evidence of the impact of BOC on influencing the emergence of leadership during U-turn behaviors.

To further elucidate the crucial role of BOC, we conducted the same correlation analysis for two other classical motion cues, i.e., the Euclidean distance[9] and bearing change of neighbors[14] (the detailed mathematical definitions of these two motion cues can be found in Supplementary Note 2). Following the same procedures used to calculate the BOC-based motion salience (the detailed procedure can be found in Fig. 2a–f or Supplementary Fig. 31), we derived the corresponding distance-based motion salience and bearing change-based motion salience (see Supplementary Note 2 for detailed information). On the one hand, the correlation analysis with distance-based motion salience examines whether neighbors closer to the focal individual exert a greater influence on shaping leadership during U-turns. On the other hand, the correlation analysis with bearing change-based motion

salience investigates whether the neighbors with greater relative position changes tend to be leaders within the group. As shown in Fig. 2g–i, we found that the Spearman correlation between the leadership and these two motion cues were both much weaker than the correlation with BOC-based motion salience, even when frontal preference is strongly involved ($α = 10$), indicating that neither distance nor bearing change serves as the primary factor in shaping leadership within the group. These results further highlight the crucial role of BOC during U-turn behaviors.

Moreover, it is noteworthy that the aspect ratio of the elliptical individual might be an influential factor in the experimental data analysis of real fish schools, which is defined as the ratio of the major axis length (parameter $a$ shown in Fig. 1a) to the minor axis length (parameter $b$ shown in Fig. 1a)) of an ellipse, i.e., denoted as $a/b$. To investigate the influence of the aspect ratio, we repeated the same analysis shown in Fig. 2a–f with different aspect ratios from $a/b = 0.1$ to $a/b = 0.5$. These additional results showed that the aspect ratio had a negligible effect on the experimental outcomes that confirmed the significant role of BOC during U-turns in fish schools (see Supplementary Fig. 14 for detailed analysis results). Overall, our empirical findings indicated that the local interaction mechanism underlying the U-turn behaviors is closely associated with the body orientation change, in which individuals prefer to follow neighbors with larger BOC.

## BOC-based interaction leads to non-trivial scale-free correlation

The empirical findings in fish schools led us to hypothesize a pairwise interaction mechanism based on BOC, where the focal individual selectively interacts with the neighbor exhibiting the largest BOC, referred to as BOC-based interaction. To explore how this bio-inspired mechanism drives the overall collective response, we introduced BOC-based interaction into the self-propelled model[9,12,42], a theoretical framework of swarm model used to simulate how individuals move based on certain local interaction mechanisms, which suggests that the motion decisions of individuals are guided by the most influential neighbor characterized by BOC and the focal individual can adjust its motion by behavior imitation or velocity alignment with this selected neighbor to achieve the collective response (see Methods for detailed definitions of the swarm model with BOC-based interaction).

To deeply understand the dynamics of information transfer governed by BOC-based interaction, we conducted simulation experiments of collective spin and collective turn in a real-time physics simulation environment[43] with a group of two-wheeled differential simulated robots. On the one hand, collective spin is an idealized and simplified scenario designed to study information transfer of BOC-based interaction. In the collective spin, individuals can adopt one of two motion states (or behavior), i.e., either spinning or remaining stationary, and update their motion states based on the states of their selected neighbors, where the spin refers to an in-place rotational movement and the stationary means an individual halts all movement. A spin initiator begins the rotational movement at a predetermined time, serving as the trigger for propagating the spinning state (or behavior) within the group. When individuals adapt their motion states (or behavior) to either spin or stay stationary, it enables the spinning state to disseminate within the group. Importantly, the goal of the collective spin is to achieve in-place rotation collectively throughout the group before the spin initiator completes a $2π$ rotation (see Methods for detailed descriptions of collective spin experiments). On the other hand, collective turn represents a more complex and realistic scenario, in which individuals are required to adjust their movement in response to a sudden directional change initiated by one informed individual (or turn initiator) through velocity alignment. Unlike the stationary nature of the collective spin, the collective turn involves the actual movements of individuals and demands rapid reorientation as the group shifts moving direction, posing a more challenging task of

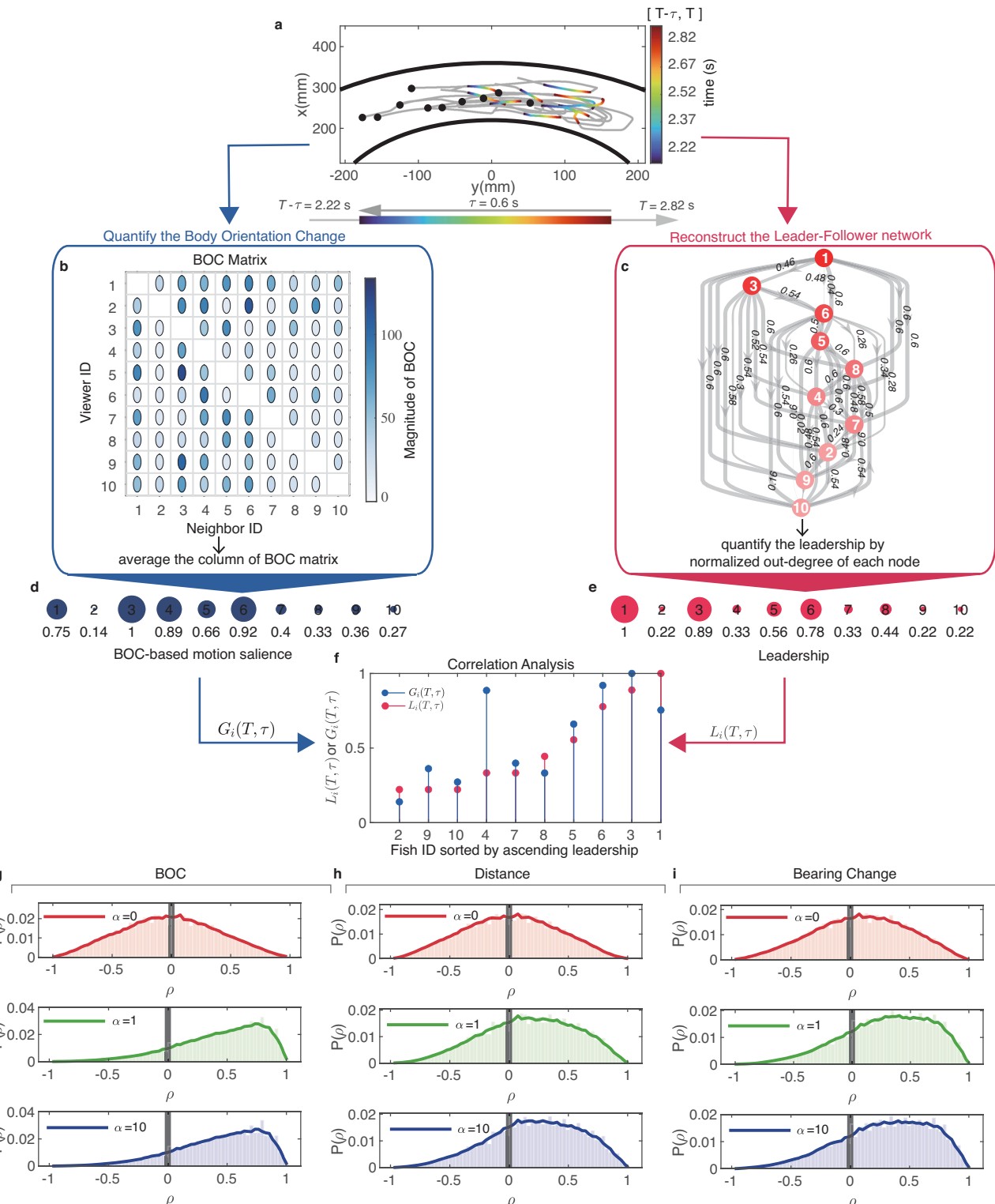

**Fig. 2 | Analysis of experimental data from collective U-turn behaviors in fish schools.** In panels (**a**–**f**), we presented the procedure of experimental data analysis used to reveal the role of BOC during U-turns. Specifically, for a given U-turn trajectory (**a**), we could construct a BOC matrix (**b**), and leader-follower network (**c**) within the period trajectory [2.22s, 2.82s] ($T = 2.82s$ and $\tau = 0.6s$). The analyzed part of the trajectory (**a**) is drawn with a gradient color that increases with time. Consequently, we derived the BOC-based motion salience $G_i(T, \tau)$ and leadership $L_i(T, \tau)$ of each individual (see Methods for more detailed information), as shown in (**d**) and (**e**), respectively. Interestingly, two vectors composed of $L_i(T, \tau)$ and $G_i(T, \tau)$ are positively correlated. Finally, we calculated the Spearman correlation coefficient $\rho$ between two vectors composed of $L_i(T, \tau)$ or $G_i(T, \tau)$ for each U-turn trajectory across different combinations of $T$ and $\tau$. **g**–**i** The distributions of $\rho$ for the BOC, distance, and bearing change with $\alpha = 0$, $\alpha = 1$, and $\alpha = 10$, respectively. Boxplots of the correlation coefficients $\rho$ of the BOC, distance, and bearing change are shown in Supplementary Fig. 12.

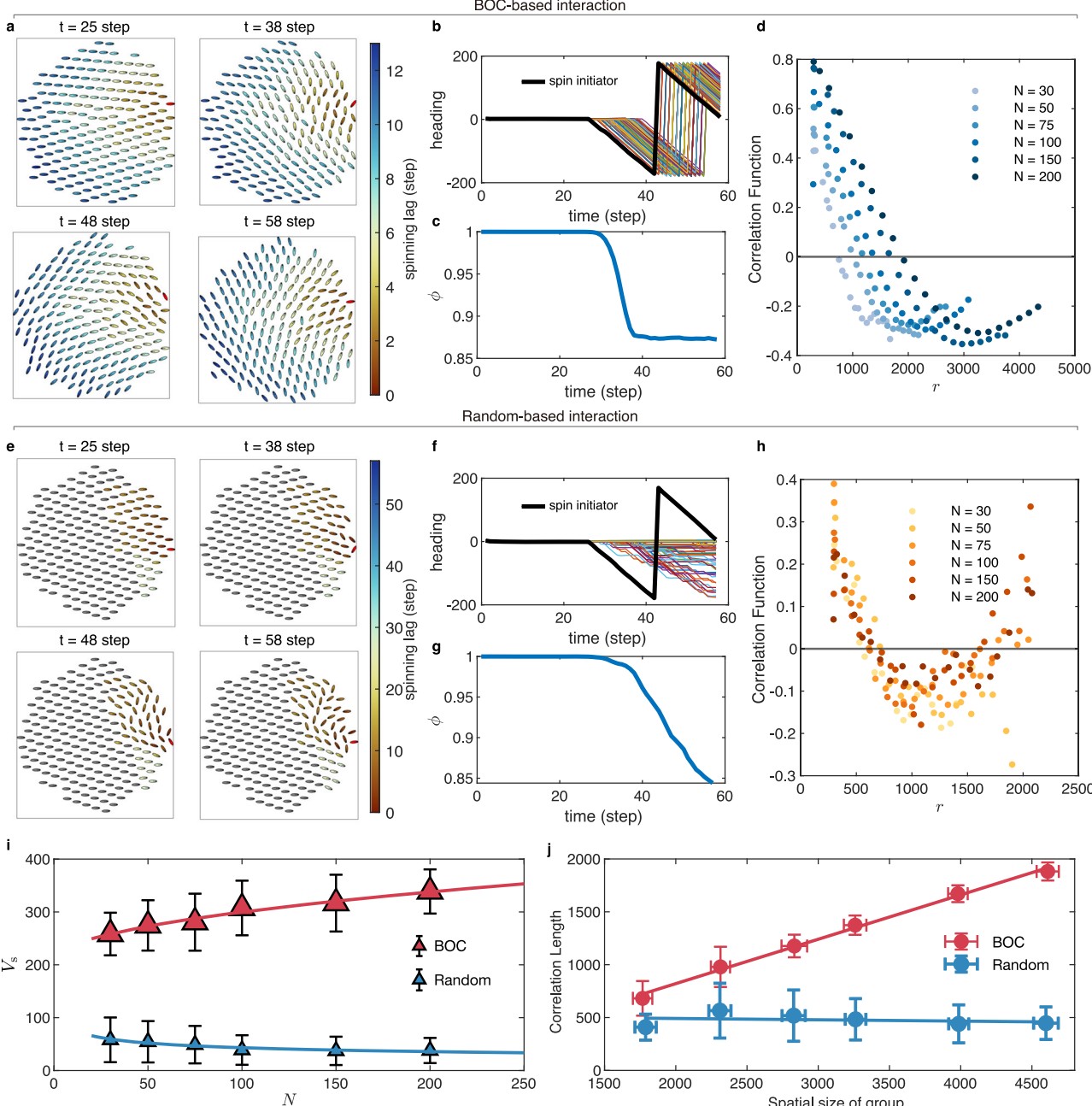

**Fig. 3 | Simulation results for the collective spin experiments. a** Snapshots of the collective spin with a group size of $N = 200$ from the activation of the spin initiator ($t = 25$ step) to its completion of a $2\pi$ rotation ($t = 58$ step) in the group with BOC-based interaction. **b, c** Headings and polarization of the group with BOC-based interaction as a function of time. **d** The correlation function varies as the increasing distance $r$ with different group sizes $N$ for the group with BOC-based interaction. **e** Snapshots of the collective spin with a group size of $N = 200$ from the activation of the spin initiator ($t = 25$ step) to its completion of a $2\pi$ rotation ($t = 58$ step) in the group with random-based interaction. **f, g** Headings and polarization of the group with random-based interaction as a function of time. **h** The correlation function varies as the distance $r$ with different group sizes $N$ for the group with random-based interaction. **i** Relationship between the information transfer speed and group size for the groups with BOC-based and random-based interaction. **j** Relationship between the correlation length and spatial size of group for BOC-based and random-based interaction. The spatial size of the group is calculated as the maximal relative distance among individuals. All the data points are averaged over 50 times independent simulations. The unit of the information transfer speed is mm s⁻¹. The units of the spatial size of the group and correlation length are mm. In panels **i, j** the error bar represents the standard deviation calculated from 50 times independent simulations. The color of each individual represents the spinning lag, defined as the time delay relative to the spin initiator. Stationary individuals are colored gray, and the spin initiator is red. Arrows on the elliptical individuals indicate their headings.

information propagation through the BOC-based interaction (see Methods for more detailed information on collective turn simulation experiments). Notably, we used the random-based interaction as the baseline model to highlight the advantages of BOC-based interaction. In the random-based interaction, an individual randomly selects a neighbor to react within the perception range irrespective of the

neighbors' BOC (see Methods for the detailed description of the baseline swarm model with random-based interaction).

For the collective spin simulation experiments, as shown in Fig. 3a, b, we found that the spinning state has been rapidly propagated within the group through BOC-based interaction. This further causes individuals to spin collectively (Fig. 3a), with their headings continuously

changing with that of the spin initiator (Fig. 3b). Additionally, we evaluated the velocity consensus of the group during the collective spin by calculating the polarization $\phi$[9] (see Supplementary Note 3 for a detailed description of the mathematical definition). In brief, the polarization $\phi$ is calculated as the normalized sum of the unit velocity vectors of all individuals in the group, with values ranging from 0 to 1. High polarization indicates a strong alignment of velocities among individuals, whereas low polarization indicates a dispersed group with weaker directional alignment. As shown in Fig. 3c, polarization sharply decreased when the informed individual began spinning but gradually reconverged to a new consensus level. Such reconvergence occurs only when all individuals have begun spinning collectively, further confirming the rapid transfer of the spinning state in the group with BOC-based interaction.

In contrast, as shown in Fig. 3e, in the group with random-based interaction, the spinning state cannot propagate among individuals before the spin initiator completes its $2\pi$ rotation, and only a small number of individuals close to the spin initiator begin spinning. Notably, these individuals frequently switched between the spinning and stationary states, as evident from their intermittent changes of heading (Fig. 3f). This further caused the group's polarization to consistently decrease without reconvergence (Fig. 3g). Additionally, the impacts of the group size on the polarization, max spinning lag, and information transfer direction were also investigated to show the advantages of BOC-based interaction in the propagating spinning state (see Supplementary Note 11 and Supplementary Fig. 30 for detailed results and mathematical definitions of these indicators).

Moreover, we analyzed the information transfer speed $V_s$ governed by BOC-based interaction in simulation experiments of collective spin, which is a measure of how quickly a new motion state or information, is disseminated within a group. In the context of collective spin, $V_s$ evaluates how fast the spinning state (or behavior) triggered by one initiator spreads in the group. The detailed computation process of $V_s$ in simulation experiments of collective spin can be found in Methods. As shown in Fig. 3i, we found that in groups with BOC-based interaction, the information transfer speed $V_s$ is significantly greater than that in groups with random-based interaction. Interestingly, as the group size increases, the information transfer speed in groups with BOC-based interaction shows an upward trend, which suggests the emergence of avalanche-like information transfer[44,45]. In contrast, the $V_s$ in groups with random-based interaction remained nearly constant and did not increase with group size. These results showed that the BOC-based interaction is an effective mechanism for information transfer with great efficiency and scalability.

To explore the reason underlying the advantage of BOC-based interaction in information transfer, we hypothesized that it might be the consequence of scale-free correlation, which is a macro-level phenomenon characterized by the correlations among velocity fluctuations of individuals. To identify the occurrence of scale-free correlation within the group, we analyzed the relationship between the correlation length and spatial sizes of the group. The correlation length indicates the distance at which velocity fluctuations among individuals are no longer aligned, providing an estimate of the spatial range over which the influence of one individual on another diminishes. To obtain the correlation length, we first calculated the correlation function of the velocity fluctuation[31]$C(r)$, which measures the average directional alignment of velocity fluctuations among individuals at a distance $r$ (see Methods for a detailed mathematical definition of $C(r)$). In particular, $r$ is a variable that represents the distance between two individuals, which can be varied to investigate how the directional correlation between velocity fluctuations evolves with increasing distance. Specifically, when distance $r$ is small, the velocity fluctuations among individuals are closely aligned, leading to a high value of $C(r) > 0$. However, as the distance $r$ increases, the directions of

these vectors gradually diverge, causing $C(r)$ to continuously decrease until it eventually crosses the x-axis $C(r) < 0$.

The correlation length is then defined as the distance $r_0$ at which the correlation function $C(r)$ first reaches zeros, i.e., $C(r_0) = 0$ (see Methods for the details of the correlation length calculation or Supplementary Fig. 33). For example, we illustrated the correlation function $C(r)$ varies with the increasing distance $r$ for BOC-based and random-based interaction in Fig. 3d, h, respectively. From Fig. 3d, h, we found that under the influence of BOC-based interaction, the point where the correlation function $C(r)$ first crosses the x-axis shifts forward as group size increases. In contrast, for the random-based interaction, the crossing points of the correlation function on the x-axis approximately coincide regardless of the group size. Additionally, we determined the exact distance $r_0$ that makes $C(r_0) = 0$ for different spatial sizes of group with BOC-based and random-based interaction. In particular, the spatial size of a group is estimated as the maximal relative distance among individuals. As shown in Fig. 3j, we found that the correlation length is linearly proportional to the spatial size of the group in the case of BOC-based interaction, which indicates the emergence of scale-free correlation in the group. On the contrary, random-based interaction do not lead to scale-free correlation, as the correlation length remains approximately constant with the increasing spatial size of group.

In addition to the collective spin, it is important to show the advantages of BOC-based interaction and explore the emergence of scale-free correlation in a more realistic scenario, i.e., collective turn (see Methods for more detailed information), which is a common scenario in real-world applications of swarm robotics[4,46]. To increase the difficulty of the group to respond to an abrupt turn, we set two typical turning angles $\theta_{info}$ for the informed individual (or turning initiator): $\theta_{info} = \frac{\pi}{2}$ and $\pi$. While the experiment setup of collective turn shares similarities with the U-turn behavior observed in real fish schools, it is important to note that these simulation experiments are not modeled upon the U-turn data of real fish. Notably, to prevent collisions among the simulated robots in the collective turn experiments, we complemented the velocity alignment (described in Eq. (7)) with additional repulsive interactions among individuals. These repulsive interactions occur when individuals come too close to each other, causing them to move apart gently (see Supplementary Note 7 for detailed mathematical definitions of repulsive interactions). In addition, to compare BOC-based interaction with random-based interaction quantitatively, we analyzed the response accuracy $\delta_{resp}(t)$ and responsiveness $R$ in collective turn experiments (see Methods for detailed mathematical definitions). On the one hand, response accuracy $\delta_{resp}(t)$ measures how closely the group aligns with the direction of an informed individual. In particular, $\delta_{resp}(t) = 1$ indicates the most accurate response, whereas $\delta_{resp}(t) = -1$ signifies the worst response. On the other hand, responsiveness $R$ evaluates how quickly and efficiently the group responds to the direction change initiated by the informed individual. Notably, the lower the $R$, the higher the responsiveness of the group.

To begin with, we presented the trajectories of individuals with different turning angles in the groups with BOC-based and random-based interaction in Fig. 4a–d. It is evident that the group with BOC-based interaction successfully achieves the collective turn with high response accuracy $\delta_{resp}$ for both turning angles (Fig. 4a, c). The group polarization (see Supplementary Note 3 for the mathematical definition of polarization) also rapidly returned to a high consensus level $\phi \approx 1$. In contrast, in the group with random-based interaction (Fig. 4b, d), only a few individuals responded to the informed individual, resulting in a low response accuracy, especially when $\theta_{info} = \pi$. This further caused a decline in group polarization, preventing it from returning to $\phi \approx 1$. Additionally, the group with BOC-based interaction not only achieved the collective turn with the optimal accuracy (Fig. 4e) but also

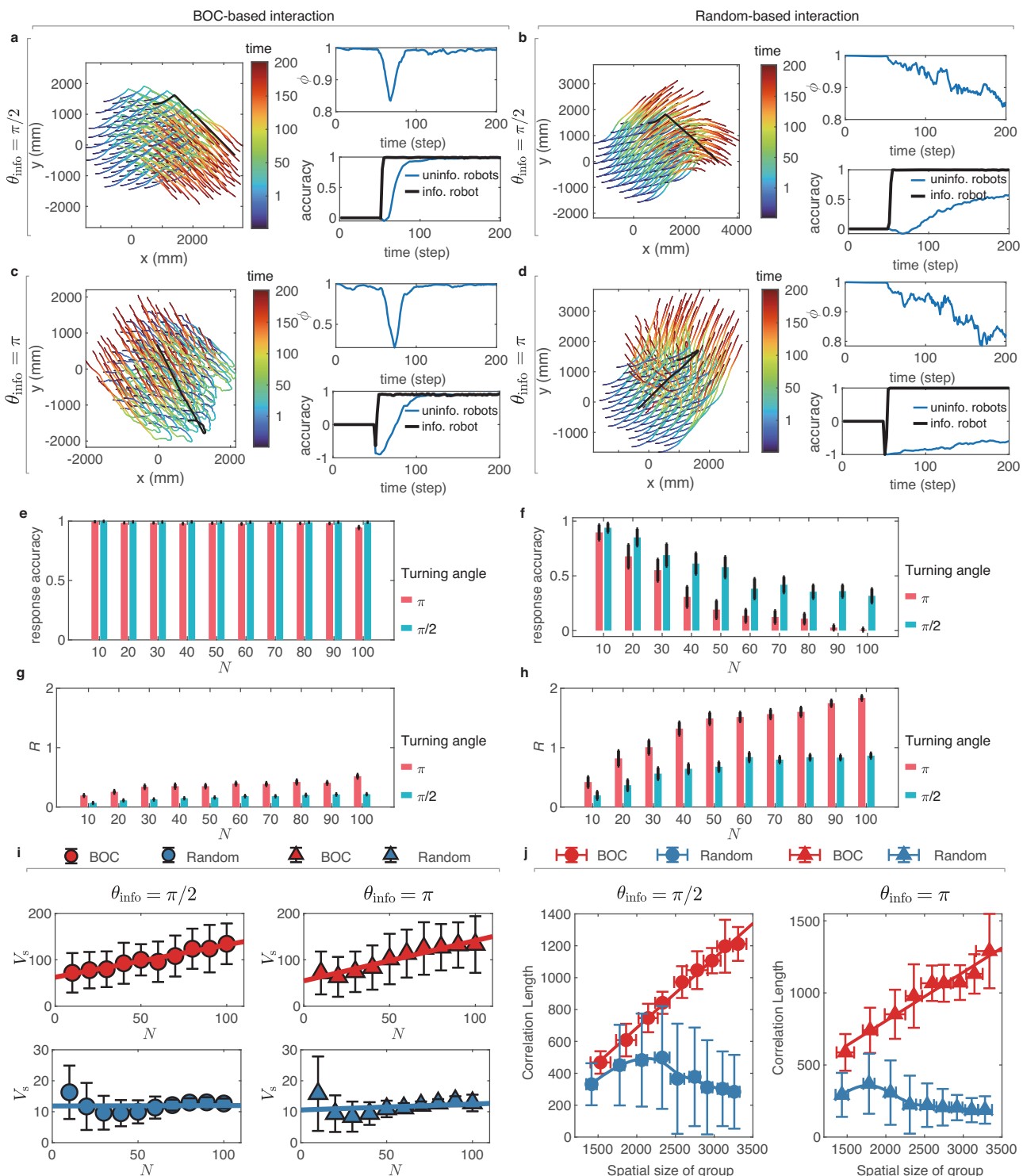

**Fig. 4 | Simulation results for the collective turn experiments. a−d** Snapshots of collective turn for the turning angles of $\frac{\pi}{2}$ and $\pi$ with $N = 100$ for groups with BOC and random-based interaction. Black lines represent the trajectories of the informed individual in panels (**a−d**). **e−h** Response accuracy and Responsiveness $R$ as a function of group size $N$ for groups with BOC and random-based interaction, respectively. **i** Information transfer speed $V_s$ as a function of $N$ for groups with BOC and random-based interaction. **j** The correlation length as a function of the spatial size of the group for BOC and random-based interaction. The spatial size of the group is calculated as the maximal relative distance among individuals. In panels **e−j**, error bars show the standard deviation from 50 independent simulations. The unit of the $V_s$ is mm s⁻¹ and both the spatial size of the group and correlation length are mm.

showed a strong advantage in keeping $R$ stable at a low level regardless of the group size (Fig. 4g), suggesting the high responsiveness of BOC-based interaction. In contrast, in the group with random-based interaction, $R$ rose sharply with the increasing group size for both turning angles (Fig. 4h), and the response

accuracy decreased (Fig. 4f) in the meanwhile, indicating the low responsiveness of the group with random-based interaction.

Moreover, we calculated the information transfer speed $V_s$ and analyzed the emergence of scale-free correlation in the groups with BOC-based and random-based interaction. Particularly, in the context

of collective turn, $V_s$ evaluates how fast the direction of one informed individual spreads within the group (see Methods for the detailed definition of $V_s$ in simulation experiments of a collective turn). As shown in Fig. 4i, the $V_s$ of the group with random-based interaction is consistently lower than that of the group with BOC-based interaction and does not increase with group size. For the group with BOC-based interaction, we found a consistent linear increase in both information transfer speed (Fig. 4i) and correlation length (Fig. 4j) with the growth of the spatial size of the group, which indicated the emergence of scale-free correlation. Nevertheless, the group with random-based interaction failed to generate scale-free correlation in the collective turn (Fig. 4j). The correlation function $C(r)$ as it varied as the increasing distance $r$ for the groups with BOC-based and random-based interaction in simulation experiments of collective turn is shown in Supplementary Fig. 29.

Furthermore, we have also analyzed the impact of the informed individual's position on BOC-based interaction in both simulation experiments of collective spin and collective turn. The simulation results showed that BOC-based interaction facilitates information transfer within the group and leads to non-trivial scale-free correlation, regardless of whether the informed individual is positioned at the center, middle, or border of the group (see Supplementary Note 9 for detailed simulation results). These simulation results provide more compelling evidence that the BOC serves as an efficient visual cue for achieving collective responses and is a critical factor influencing the emergence of scale-free correlation in the group.

### Application to swarm robotics

To further validate the feasibility of BOC-based interaction in swarm robotics, we adopted the BOC-based interaction in a swarm consisting of 50 SwarmBang robots[46–48] (see Methods and Supplementary Note 6 for a brief overview of the SwarmBang system). While the simulation results revealed the advantage of BOC-based interaction for information transfer, it is crucial to validate the effectiveness of such bio-inspired mechanisms under realistic and non-ideal conditions, including motion errors and randomness. These realistic factors may lead to incorrect responses or actions of robots and further hinder the emergence of the collective response in robotic swarms.

In contrast to the simulation experiments, on the one hand, we have the robotic swarm respond to successive abrupt turns of informed individuals, posing a higher demand on the responsiveness of BOC-based interaction. On the other hand, we adopt the pinhole camera model to simulate the vision-based sensing of robots, which is a simplified representation of how a camera projects a 3D scene onto a 2D image plane (see Methods for detailed definition of pinhole camera model). Based on the pinhole camera model, we could estimate the BOC of neighbors via the simulated camera images perceived by the focal individual (see Methods for more detailed information), which provides an opportunity to validate BOC-based interaction using simulated visual perception from the view of robots. Due to constraints of the experimental arena size, we manipulated the informed individual to initiate a turn every 50 s, for a total of two turns throughout real robot experiments. For each turn, the informed robot is picked at the front of the group and turns $\frac{\pi}{2}$ with respect to the group's moving direction. The parameters selected for the robotic experiments can be found in Supplementary Table 3.

The experimental results showed that using BOC-based interaction, the swarm with 50 robots effectively responds to the successive heading changes of the informed robot despite the challenges introduced by motion errors and randomness (Fig. 5a and see Supplementary Movie 9). As a comparison, the robotic swarm using the random-based interaction (see Methods for the detailed description of the baseline swarm model with random-based interaction) cannot follow the successive turning of the informed robots (Fig. 5b and see Supplementary Movie 10). Additionally, the response accuracy of the

group with random-based interaction slowly increased but remained at $\delta_{resp} < 0.5$, and the polarization also consistently declined, failing to return back to the high-velocity consensus. Moreover, Fig. 5c–e presented compelling statistical results that highlight the advantages of BOC-based interaction in terms of response accuracy and responsiveness compared with the random-based interaction. In addition, we also compared the BOC-based interaction with the classical Vicsek model[9], i.e., a simplified local interaction of averaging-velocity alignment. The experimental results revealed that the BOC-based interaction is better than the Vicsek model in responding to successive abrupt turns (see Supplementary Note 10 and Supplementary Movies 9, 11 for detailed results).

Furthermore, we compared the information transfer speed $V_s$ of the BOC-based and random-based interaction in robotic swarms, which shared the same definition in simulation experiments of collective turn (see Methods for detailed definition of $V_s$ in the context of a collective turn). As shown in Fig. 5f, the $V_s$ of the group with BOC-based interaction is much faster than that of the group with random-based interaction, which accounts for the strong ability of BOC-based interaction in directional information dissemination under non-ideal conditions. Importantly, we also conducted the swarm robotic experiments with smaller group sizes of 20 and 30 robots to explore the emergence of scale-free correlation in real robotic swarms. As shown in Fig. 5g, for the group with BOC-based interaction, the correlation length increases with the spatial size of the robotic swarm, indicating the emergence of non-trivial scale-free correlation in swarm robotics. However, a similar macro-level phenomenon is not observed in the group with random-based interaction. Overall, the robotic experiments validated the feasibility and applicability of the BOC-based interaction under realistic and non-ideal conditions and further supported the significant role of body orientation change in the emergence of scale-free correlation.

## Discussion

This study aimed to present a visual cue from the first-person perspective that could serve as sensory input to facilitate the emergence of collective responses in vision-based swarm models. It was hypothesized that body orientation change of neighbors might be a crucial visual cue in enhancing the efficiency of information transfer within groups. Based on a comprehensive research chain from biological data analysis to bionic mechanism modeling and further to swarm robotics application, we confirmed that BOC is not only associated with the emergence of leadership during U-turn behaviors in fish schools but also enables rapid information transfer in swarm robotic systems and gives rise to the non-trivial phenomenon of scale-free correlation.

Our work represents an effort to model collective behaviors based on visual observations rather than explicit physical measurements, which is crucial for moving beyond swarm models derived from computational theories and handcrafted designs[8,9,12–14]. In particular, we characterized each individual as a non-transparent ellipse to investigate BOC in both fish schools and artificial swarms. This further enabled us to involve several important realistic factors in local interactions, such as the aspect ratio of the individuals[19] and visual occlusions[49,50]. Additionally, we highlighted the significance of conducting a comprehensive research chain in extracting interaction mechanisms from animal groups, as it ensures such bio-inspired mechanisms (e.g., BOC-based interaction) are not only theoretically sound with biological plausibility but also applicable in real-world contexts.

Moreover, our work sheds new light on the emergence of scale-free correlation, which is a macro-level phenomenon observed in various biological and artificial systems[31,32,51,52]. The scale-free correlation suggests that the velocity fluctuations among individuals are directionally aligned over longer distances in larger groups, which typically occurs in the critical regime of a self-organized system.

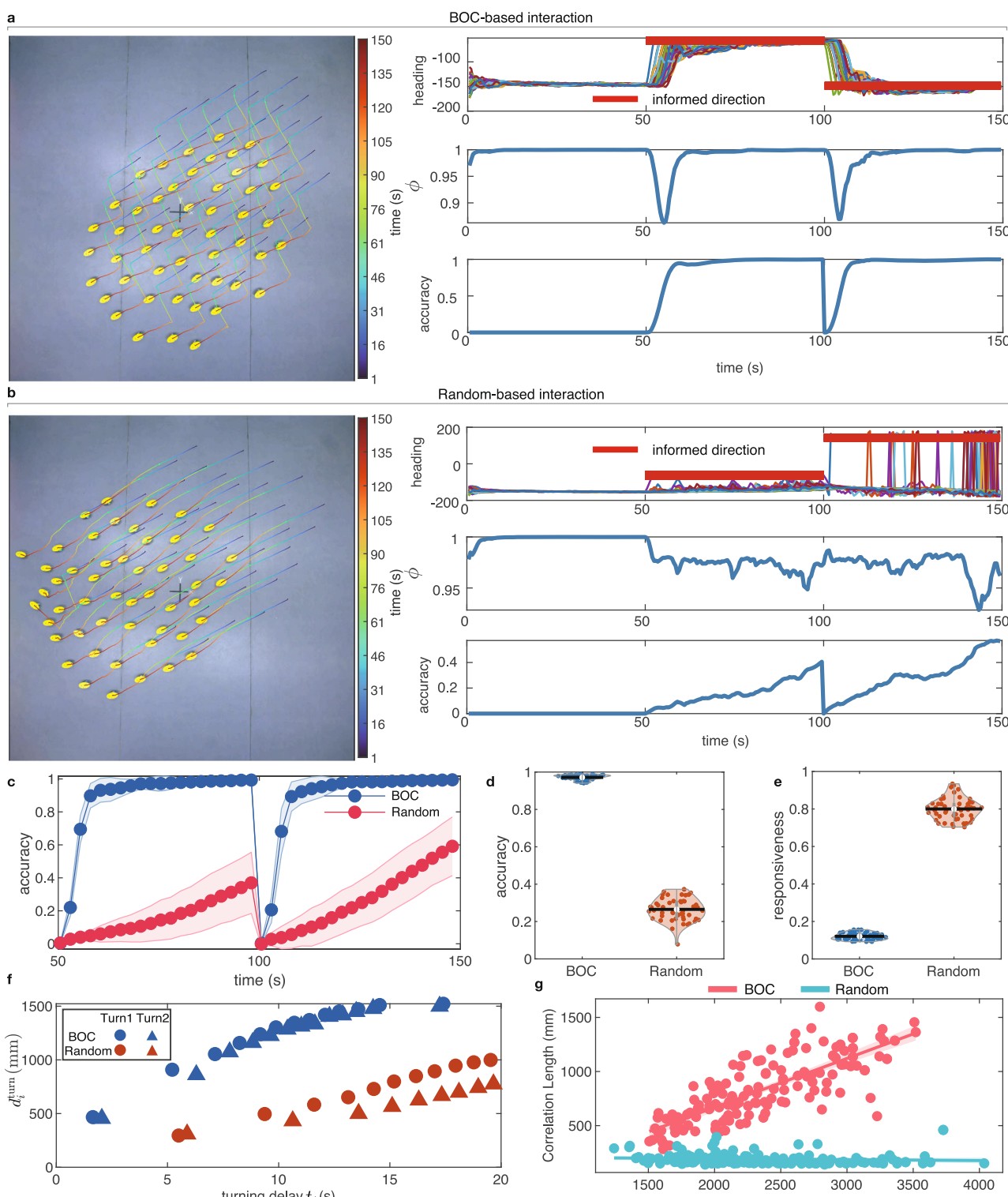

**Fig. 5 | Swarm robotic experiment results. a** The swarm consisted of 50 robots that used the BOC-based interaction to achieve the successive collective turn. **b** The swarm using random-based interaction could not respond to heading changes of the informed robot in time and failed to finish the successive collective turn. **c** Comparison of response accuracy as a function of time for groups with BOC-based and random-based interaction. **d, e** Statistical comparison of the response accuracy and responsiveness for groups with BOC-based and random-based interaction. **f** Information transfer distance $d_i^{turn}$ as a function of turning delay $t_i$ for groups with BOC-based and random-based interaction, respectively. The slopes in panel (**f**) represent the information transfer speed $V_s$ within the group. **g** The correlation length as a function of spatial size of group for BOC-based and random-based interaction, respectively. The spatial size of the group is calculated as the maximal relative distance among individuals. In panel **c**, the shaded error bar represents the standard deviation. In panels **d, e**, black lines represent the means of the data distributions. The robot experiments for each interaction type are repeated 50 times independently.

However, it is important to note that BOC-based interaction exhibits scale-free correlation with generic parameters that are qualitatively consistent with those observed in biological swarms[31] and facilitates rapid information transfer within the swarm. This further suggests the presence of scale-free correlation may be essential to the enhancement of collective responses within groups. In addition to our findings, researchers have also found the emergence of scale-free correlation in self-organized systems operating outside the critical regime. For example, Huepe et al. provided an alternative mechanism for generating this macro-level phenomenon based on positional interaction with the absence of critical dynamics[53].

Furthermore, we believe that BOC stands out as a significant visual cue in the context of swarm robotics due to its sole dependence on the orientation change-induced maneuver, which provides immediate and explicit visual feedback on the movements of neighboring robots. In contrast to other cues, such as the neighbors' bearing change, which requires additional time to accumulate velocity into displacement before identifying the neighbor's salient movement. Notably, as swarm robotics expect to make decisions and actions based on raw and immediate observations through simple local sensing devices[54,55], this advantage becomes particularly valuable because BOC can be effectively estimated via onboard cameras through standard computer vision techniques[56–59]. For example, robots can effectively estimate the magnitude of the BOC by calculating changes in the bounding box area over consecutive frames obtained with onboard optical cameras (see Supplementary Note 8 and Supplementary Fig. 21 for detailed information).

Although the BOC-based interaction is a novel mechanism with considerable potential in swarm robotics, it still remains much challenging work from the perspective of control theory and engineering, such as addressing the consensus and stability issues of BOC-based interaction[60,61], mitigating effects of noise and time delays in real-world applications[62,63], etc. Additionally, we openly admitted that elucidating the role of BOC in collective response addresses only one aspect of collective behavior modeling in this work, i.e., determining which visual cues individuals should focus on during collective response. Another fundamental issue that warrants further investigation is how biological individuals integrate BOC from visual neurobiological circuits[2] and convert them into motion to achieve collective responses[64,65].

Overall, this study offers an essential interaction mechanism for swarm robotics to perform complex and sophisticated collective tasks[5,66] that demand high maneuverability. Moreover, our results provide valuable insights not only for biologists and researchers in complex systems but also for computer scientists and engineers specializing in swarm robotics. In particular, our comprehensive research chain in this work encourages interdisciplinary collaboration across biology, physics, and engineering for the development of high-performance and cost-effective swarm robotic systems.

## Methods

### Simulation experiments of collective spin and collective turn

In this work, we conduct simulation experiments of collective spin and collective turn in a pybullet simulator[43], where each individual is a simulated two-wheeled differential robot. The pybullet simulator provides a powerful physics engine that accurately simulates the kinematics of robot motion and physical collisions, which facilitates the application of BOC-based interaction with real robots. In particular, the occluded individuals are ignored in the neighbor selection (see detailed approaches for identifying the occluded neighbor in Supplementary Note 4 and Supplementary Fig. 15).

For the collective spin simulation experiments, we assumed that each simulated robot has two motion states (or behavior): stationary and spinning. In the stationary state, both the linear and angular speeds of the simulated robots are set to 0. In the spinning state, the

angular speed is set to the maximum $\omega_{\max}^{\mathrm{sim}} = 0.83\,\mathrm{rad}\,\mathrm{s}^{-1}$, whereas the linear speed remains 0, indicating the in-place rotational movement of the simulated robots. The propagation of the spinning state (or behavior) is triggered by a spin initiator selected at the front of the group, who starts spinning at a preset activation time (the 25th step in our simulation) and stops spinning after completing a $2\pi$ rotation. The simulated robots are required to swiftly transfer the spinning state throughout the entire group within a limited time, i.e., before the spin initiator completes its $2\pi$ rotation. The reaction rule of an individual is to switch between the stationary and spinning states based on the state of the selected neighbor. For example, in the context of BOC-based interaction, each simulated robot determines the BOC of its neighbors within a perception range $R_{\mathrm{visual}}$ and selects the neighbor with the maximum BOC to react. If the selected neighbor is in the spinning state, the focal individual also enters the spinning state and starts to spin with the maximal angular speed; if the selected neighbor is in the stationary state, the focal individual switches to the stationary state and stops spinning. The simulation process of the collective spin can be found in Supplementary Movies 3, 4. Unless otherwise specified, the parameters selected for the BOC-based and random-based interaction can be found in Supplementary Table 1. An illustration of the collective spin simulation procedure is shown in Supplementary Fig. 26.

For the collective turn simulation experiments, the simulated robots are required to change their direction of movement in response to a sudden turn initiated by one informed individual (or turn initiator). The informed individual, positioned at the forefront of the group, turns with an angle $\theta_{\mathrm{info}}$ relative to the group's movement direction at a preset moment (the 50th step in our simulation). The new direction of the informed individuals then triggers the group to start a collective turn. The reaction rule of an individual in the collective turn simulation experiments is a commonly used velocity alignment rule[46], i.e., average the headings of the selected neighbor and the robot's own heading (Eq. 7). For example, in the context of BOC-based interaction, each robot adjusts its movement to align with the neighbor that exhibits the largest BOC. In particular, due to the challenges in acquiring the velocities of the neighbors from the first-person view[19], we assumed that the focal individual could access the information of neighbors' velocity. The simulation process of collective turn can be found in Supplementary Movies 5–8. Unless otherwise specified, the parameters selected for the simulation experiments of collective turn for the BOC-based and random-based interaction can be found in Supplementary Table 2. An illustration of the collective turn simulation procedure is shown in Supplementary Fig. 26.

### Construction of the BOC matrix and derivation of BOC-based motion salience

To obtain the BOC-based motion salience of each individual, we first calculated the BOC of neighbors from each individual's view according to the definition of Eq. (1) over a period of $[T - \tau, T]$, which forms a BOC matrix denoted as $\mathbf{G}(T, \tau) = [g_{ij}(T, \tau)]_{N \times N}$ (see Fig. 2b and Supplementary Fig. 31 for detailed information). In a BOC matrix $\mathbf{G}(T, \tau)$, the rows correspond to the focal individuals (or viewers), whereas the columns represent their respective neighbors. Consequently, we can derive the motion salience of each individual $G_i(T, \tau)$ by averaging each column in BOC matrix $\mathbf{G}(T, \tau)$ (Fig. 2d), i.e., the BOC-based motion salience. From a mathematical standpoint, the BOC-based motion salience is a vector $[G_1(T, \tau), \ldots, G_i(T, \tau)]_{i \in [1, \ldots, N]}$, which contains how noticeable the movements of the focal individual $i$ are perceived by other individuals in the group based on the BOC. Individuals with larger $G_i(T, \tau)$ exhibit more salient directional changes in their movements during $[T - \tau, T]$. Conversely, an individual with lower $G_i(T, \tau)$ exhibits more stable movements with fewer directional changes in the group. In particular, the BOC-based motion salience is normalized by the maximum $G_i(T, \tau)$ value in the group to ensure that the Spearman correlation analysis

results are not skewed by differences in the data scales (see Supplementary Fig. 31 for detailed information).

## Reconstructing the leader-follower network to quantify the leadership during collective U-turns

To quantitatively evaluate the leadership of each individual during collective U-turns, we first reconstructed the leader-follower network in fish schools, which is derived from each pair of leader-follower relationships within the group. By adopting the directional alignment function analysis[30], we could obtain the leader-follower relationships of any pair of two individuals in the group.

Specifically, given a pair of individuals $i$ and $j$ and a U-turn trajectory period spanning from $T_b$ (beginning timestamp) to $T_f$ (final timestamp), the directional alignment function $\xi_{ij}(\lambda_t)$ represents the degree of transient velocity alignment of these two individuals as a function of the time lag $\lambda_t$. The directional alignment function $\xi_{ij}(\lambda_t)$ is defined as follows:

$$\xi_{ij}(\lambda_t) = \left\langle \hat{\mathbf{v}}_i(t) \cdot \hat{\mathbf{v}}_j(t+\lambda_t) \right\rangle \quad (2)$$

where the $t$ is the time index that varies between $T_b$ and $T_f$. $\lambda_t$ is the time lag, which ranges from the $[-(T_f - T_b), (T_f - T_b)]$. The operation $\langle \cdot \rangle$ denotes the average function over all the time index $t$. As the time lag $\lambda_t$ increases, we can obtain the curve of $\xi_{ij}$ as a function of $\lambda_t$. In particular, we denote $\lambda_{ij}^*$ as the value at which the curve of $\xi_{ij}(\lambda_t)$ reaches its maximum value, indicating that individual $j$ achieves the highest degree of directional consensus with individual $i$ with $|\lambda_{ij}^*|$ seconds lag or ahead. The sign of $\lambda_{ij}^*$ determines the leader or follower role of individual $j$. $\lambda_{ij}^* > 0$ (or $\lambda_{ij}^* < 0$) means that the moving direction of individual $j$ is $|\lambda_{ij}^*|$ seconds behind (or ahead) of individual $i$, which suggests that individual $j$ is the follower (or leader) to individual $i$. For example, as shown in Fig. 2c, with the $\lambda_{13}^* = 0.46$s, fish-3 lags by 0.46 s to achieve the maximal alignment with fish-1, which suggests that fish-3 takes on the role of follower while the fish-1 is the leader (see Supplementary Fig. 32 for the curves of $\xi_{1,3}(\lambda_t)$ and the corresponding trajectory between individuals 1 and 3). As a result, we can identify the leader-follower relationships among each pair of individuals and construct the leader-follower network within the group from $T_b$ to $T_f$. For example, the leader-follower network derived from 2.22 s to 2.82 s is shown in Fig. 2c.

After obtaining the leader-follower network, we use the normalized out-distance[67] of each node (or individual) in the network to characterize each fish's leadership. The normalized out-distance signifies the hierarchical layer of each node (or individual) in the leader-follower network. In such a hierarchical network, some individuals act as leaders, whereas others act as followers, with directed connections between individuals, i.e., leaders point to their followers. We leverage the number of connections with other nodes, namely, the out-distance (or the out-degree) of nodes in the directed network, to represent the leadership of the focal individual. A higher out-degree indicates that the individual has stronger leadership within the group, as it has more connections with followers. For consistency, we use the same variable notations in Eq. (1). Given the time $T$ and a time period $\tau$, the leadership of individual $i$ from $T_b = T - \tau$ to $T_f = T$ is calculated as follows:

$$L_i(T, \tau) = \frac{d_i^{out}}{N-1} \quad (3)$$

where $N$ is the size of the group and $d_i^{out}$ is the out-distance (or out-degree) of the focal individual $i$ in the leader-follower network. $L_i(T, \tau)$ ranges from 0 to 1. A higher $L_i(T, \tau)$ indicates a stronger influence and a leadership role in the group. Conversely, an individual with a lower $L_i(T, \tau)$ is at a lower hierarchy in the leader-follower network. For

example, as depicted in Fig. 2c, e, fish-1 occupies the top level in the leader-follower network, and thus its leadership is denoted $L_1 = 1$. The validation of the effectiveness of $L_i(T, \tau)$ can be found in Supplementary Note 2.

## Swarm model with BOC-based interaction

To gain insights into information transfer governed by the BOC-based interaction, we integrated this bio-inspired mechanism into a self-propelled model[9,12,42] to conduct the simulation experiments of collective spin and collective turn. The self-propelled model is a widely used theoretical framework in swarm dynamics to simulate the autonomous movement of individuals, which enables the simulation of collective spin and collective turn arising from BOC-based interaction.

In the BOC-based interaction, we assume that each individual interacts with only one neighbor[6,14,16] within the visual range $R_{visual}$ to highlight the effects of BOC. As a result, the neighbor is selected only if it has the largest magnitude of body orientation change from the focal individual's perspective within the $R_{visual}$. Additionally, to streamline both the simulations and robot experiments, we ignore the blind area of BOC perception and thus set $\alpha = 0$ in Eq. (1). The magnitude of the BOC from $T - \tau$ to $T$ in the swarm model is calculated as follows:

$$g_{ij}(T, \tau) = \sum_{t=T-\tau+1}^{T} \frac{|\beta_j(t) - \beta_j(t-1)|}{\Delta t} \quad (4)$$

where the $\Delta t$ is the time interval in the simulation. $\tau$ represents the duration over which individual $i$ considers the past movements of neighbor $j$.

Here, we consider a group composed of $N$ elliptical non-transparent individuals instead of simple particles moving in a two-dimensional plane, which is characterized by the lengths of the major axis $a$ and the minor axis $b$ (Fig. 1a). The aspect ratio is set to $a/b = 0.3$ as it has a negligible impact on the simulation results. Compared with the self-propelled particle models, the elliptical model provides a more practical basis for considering realistic factors such as the aspect ratio of the neighbors and visual occlusions among individuals. Let the $\hat{\mathbf{x}}_i$ and $\hat{\mathbf{v}}_i$ be the position and velocity of individual $i$. The position of individual $i$ is updated according to:

$$\hat{\mathbf{x}}_i(t+1) = \hat{\mathbf{x}}_i(t) + v_0 \hat{\mathbf{v}}_i(t) \Delta t \quad (5)$$

where $v_0$ is a constant moving speed. In particular, we have the $\hat{\mathbf{x}}_i(t+1) = \hat{\mathbf{x}}_i(t)$ in the collective spin simulation, as the $v_0$ is always set to 0.

For the collective spin simulation experiments, the velocity of the individual $i$ is calculated as follows:

$$\hat{\mathbf{v}}_i(t+1) = \mathbf{R}(\theta_i^{rot})\hat{\mathbf{v}}_i(t) \quad (6)$$

where $\mathbf{R}(\theta_i^{rot})$ is the rotation matrix that characterizes the rotational motion of individuals, which is defined as the $\mathbf{R}(\theta_i^{rot}) = \begin{pmatrix} \cos(\theta_i^{rot}) & -\sin(\theta_i^{rot}) \\ \sin(\theta_i^{rot}) & \cos(\theta_i^{rot}) \end{pmatrix}$. $\theta_i^{rot} = \omega_i \Delta t$ is the rotation angle of individual $i$ during a time interval. $\omega_i$ represents the angular speed of the simulated robots, which is determined by the state (or behavior) of the selected neighbor. If the selected neighbor is in the stationary state, then $\omega_i = 0$ and $\theta_i^{rot} = 0$, indicating that the focal individual $i$ stops spinning. Conversely, if the selected neighbor is in the spinning state, we set the $\omega_i = \omega_{max}^{sim}$ (see Supplementary Table 1 for the parameters selection in the collective spin simulations), signifying the rotational movement of individual $i$.

For the collective turn simulation experiments, the velocity of individual $i$ is calculated as follows:

$$\hat{\mathbf{v}}_i(t+1) = \hat{\mathbf{v}}_i(t) + k_a \cdot \sum_{j \in S_i} \Theta\left(g_{ij} = \max(\mathbf{M}_i)\right) \cdot \hat{\mathbf{v}}_j(t) \qquad (7)$$

where $S_i$ denotes the neighbor set of individual $i$, which includes all neighbors within the visual range $R_{visual}$ that are not obscured by other individuals. $k_a$ is the gain of velocity alignment. $\mathbf{M}_i = [g_{i1}(T, \tau), \ldots, g_{ij}(T, \tau)]_{j \in S_i}$ contains the magnitudes of the BOC of the neighbors observed by the focal individual $i$. $\Theta$ is the Heaviside function, which serves as a binary switch to select the neighbor with the largest BOC. The Heaviside function takes a value of 1 when its argument $g_{ij} = \max(\mathbf{M}_i)$. Otherwise, it has a value of 0.

## Baseline swarm model with random-based interaction

To show the significant role of BOC in facilitating the emergence of scale-free correlation, we involved the random-based interaction as a baseline mechanism in both simulation experiments and robotic experiments. Specifically, in the random-based interaction, the focal individual selects one neighbor to react based on a uniform random distribution within the perceptual range $R_{visual}$, which is a fair and unbiased baseline interaction mechanism. By comparing with the random-based interaction, on the one hand, we can show the effectiveness of BOC-based interaction in information transfer. On the other hand, we can explore whether the BOC is a critical factor responsible for the emergence of scale-free correlation.

Similar to the swarm model with BOC-based interaction, we incorporated the random-based interaction into a self-propelled model to conduct the collective spin and collective turn simulation experiments. For the collective spin simulation experiments, the motion state (or behavior) of the focal individual $i$ depends on the current state (or behavior) of the randomly selected neighbor. For the collective turn in simulation and robotic experiments, individual $i$ updates its velocity by aligning with the randomly selected neighbor, calculated as $\hat{\mathbf{v}}_i(t+1) = \hat{\mathbf{v}}_i(t) + k_a \cdot \hat{\mathbf{v}}_j(t)$, where the neighbor $j$ is randomly selected from the distribution Uniform $(1, N)$. Unless otherwise specified, the parameter selection in the random-based interaction is the same as the swarm model with BOC-based interaction, which can be found in Supplementary Tables 1–3 for different experiments.

## Correlation function and correlation length

To obtain the correlation length within a group, we first calculated the correlation function $C(r)$ following the computational method provided by refs. 31,32. Specifically, the correlation function $C(r)$ is used to evaluate the average degree of directional alignment among individuals' velocity fluctuations at a distance $r$. The correlation function can be calculated as follows:

$$C(r) = \frac{\sum_{i,j=1}^{N} \hat{\mathbf{u}}_i \cdot \hat{\mathbf{u}}_j \delta(r - r_{ij})}{\sum_{i,j=1}^{N} \delta(r - r_{ij})}, \qquad (8)$$

where the $\hat{\mathbf{u}}_i = \hat{\mathbf{v}}_i - \frac{1}{N}\sum_{k=1}^{N} \hat{\mathbf{v}}_k$ is the velocity fluctuation of individual $i$. $\delta(r - r_{ij})$ is a smoothed Dirac $\delta$ function that is used to filter pairs of individuals located at the same distance $r$. $r_{ij}$ is the distance between two individuals. If the relative distance $r_{ij}$ between two individuals is equal to $r$, we have $\delta(r - r_{ij}) = 1$. Otherwise, $\delta(r - r_{ij}) = 0$. High values of $C(r) > 0$ indicate the strong correlation in the velocity fluctuations among all individuals at a certain distance $r$. As $r$ increases, the value of $C(r)$ steadily decreases and eventually crosses the x-axis ($C(r) < 0$), indicating that correlation in the velocity fluctuations among individuals exhibits the decaying transition from a strong correlation to a weak correlation (see Supplementary Fig. 33 for the typical curves of $C(r)$ or Fig. 3d, h). In particular, individuals who failed to respond to the spin/turn initiator were excluded from the calculations.

The correlation length is then defined as the distance $r_0$ at which the correlation function becomes zero $C(r_0) = 0$. At a distance $r_0$, the dot product of the velocity fluctuations among individuals $\hat{\mathbf{u}}_i \cdot \hat{\mathbf{u}}_j = 0$, indicating that individuals no longer interact with each other beyond the distance $r_0$ from the macro-level perspective. The definition of the correlation function and a diagram of the correlation length can be found in Supplementary Fig. 33.

## Estimation of the information transfer speed

The information transfer speed $V_s$ measures how quickly behavioral changes or information, such as the spinning state (or behavior) and direction of an informed individual, spreads within a group, reflecting the efficiency of information transfer among individuals. Due to differences in the reaction rules governing collective spin and collective turn, the estimation method of information transfer speed $V_s$ is adapted for each of these simulation experiments.

In the collective spin simulation experiments, the estimation of $V_s$ begins with defining the spinning lag and spinning rank $s_i$. The spinning lag is defined as the time delay by which an individual's start of spinning lags behind that of the spin initiator. The spinning rank $s_i$ is determined based on the value of spinning lag, where a longer lag corresponds to a higher spinning rank. For example, the spin initiator has $s_i = 1$, whereas the last individual to start spinning has $s_i = N$. Then, we defined the information transfer distance $d_i^{spin}$ as the distance over which information has traveled to reach the individual with spinning rank $s_i$, which is estimated as the relative Euclidean distance between individual $i$ and the spin initiator, i.e., $d_i^{spin} = ||\hat{\mathbf{x}}_i - \hat{\mathbf{x}}_{spin}^{ini}||$, where the $\hat{\mathbf{x}}_{spin}^{ini}$ is the position vector of a spin initiator. Finally, based on the computational method adopted in ref. 68, information transfer speed $V_s$ in the collective spin experiment is estimated as the slope of the curve formed by the spinning lag and the corresponding $d_i^{spin}$ (see Supplementary Fig. 27 for the curves in collective spin simulation experiments).

For the collective turn simulation experiments, first, we followed the method proposed in ref. 68 to calculate the turning rank $\kappa_i$ of each individual based on the leader-follower relationship among all pairs of individuals, which are determined by the sign of time lag $\lambda_{ij}^*$ (see detailed calculation of $\lambda_{ij}^*$ in Eq. (2) in Methods). Specifically, to obtain the turning rank $\kappa_i$, we assigned scores $w_{ij}$ to individual $i$ based on the $\lambda_{ij}^*$ of its neighbor $j$. In particular, we set $w_{ij} = 1$ if $\lambda_{ij}^* < 0$, indicating that individual $i$ is the follower of individual $j$, whereas we set $w_{ij} = -1$ if $\lambda_{ij}^* > 0$, suggesting that individual $i$ is the leader of individual $j$. Then, the total score of individual $i$ is calculated as $W_i = \sum_{i \neq j} w_{ij}$. Consequently, a smaller $W_i$ means that individual $i$ turns earlier than a larger number of other individuals in the group. As a result, each individual's turning rank $\kappa_i$ is determined in ascending order of their $W_i$ within the group. For example, the first individual to turn has the smallest $W_i$, which is assigned $\kappa_i = 1$. Conversely, the last turning individual is assigned $\kappa_i = N$, as it has the largest $W_i$. Once the turning rank $\kappa_i$ is obtained for each individual, the corresponding turning delay $t_i$ is calculated as $t_i = \sum_{\kappa_j < \kappa_i, \kappa_i > 1} (t_j + |\lambda_{ij}^*|)/(\kappa_i - 1)$, which represents the time delay at which the individual begins to turn relative to the turn initiator. Particularly, for the individual with turning rank $\kappa_i = 1$, we set its turning delay as $t_i = 0$.

Next, we defined $d_i^{turn}$ as the distance that information has traveled to the individual with turning rank $\kappa_i$ (or over the time within $t_i$). Following the computational approach proposed in ref. 69, we estimated the information transfer distance $d_i^{turn}$ as being proportional to the radius of the subgroup containing the individual with a turning delay less than $t_i$. The information transfer distance in the collective turn simulation experiments is estimated as:

$$d_i^{turn} = l_r(t_i)/\rho, \qquad (9)$$

where $l_r(t_i) = \frac{\max_{(j_1, j_2 \in \mathcal{H}_i)} ||\hat{\mathbf{x}}_{j_1} - \hat{\mathbf{x}}_{j_2}||}{2}$ is the estimated radius of the subgroup $\mathcal{H}_i = \{j | t_j \leq t_i, j \in [1, ..., N]\}$, which is calculated as half of the maximum relative distance within the subgroup $\mathcal{H}_i$. $\mathcal{H}_i$ contains individuals whose turning delay is less than $t_i$. $\hat{\mathbf{x}}_{j_1}$ and $\hat{\mathbf{x}}_{j_2}$ represent the position vectors of individual $j_1$ and individual $j_2$, respectively. $\rho$ is the group density. Given the uniform initial distribution of individuals in simulation experiments and no abrupt positional changes among simulated robots, we set $\rho = 1$ to facilitate a straightforward understanding of the $d_i^{\text{turn}}$. Finally, the $V_s$ in the collective turn simulation experiments is estimated by fitting the slope of the curve formed by the turning delay $t_i$ and the corresponding information transfer distance $d_i^{\text{turn}}$ (see Supplementary Fig. 28 for the curves in collective turn simulation experiments).

### Response accuracy and responsiveness

To evaluate the quality of the collective response in both the simulations and the robotic experiments of collective turn, we defined the response accuracy $\delta_{\text{resp}}(t)$ as follows:

$$\delta_{\text{resp}}(t) = \frac{1}{N} \sum_{i=1}^{N} (\hat{\mathbf{v}}_i(t) \cdot \hat{\mathbf{n}}) \in [-1, 1], \tag{10}$$

where $\hat{\mathbf{v}}_i(t)$ is the velocity of individual $i$ at time $t$. $\hat{\mathbf{n}}$ represents the direction of the informed individual, which also corresponds to the new direction that triggers the collective turn. $\delta_{\text{resp}}(t) = 1$ indicates that the group successfully responds to the informed individual and moves in the same direction as the informed individual does. Conversely, when the group moves in a direction opposite to the informed individual, $\delta_{\text{resp}}(t) = -1$.

To evaluate the efficiency of the group response to the informed individual, we followed the similar indicator used in ref. [4], i.e., the responsiveness. The responsiveness $R$ is defined as follows:

$$R = \frac{1}{t_1 - t_0} \int_{t_0}^{t_1} (1 - \hat{\mathbf{V}}(t) \cdot \hat{\mathbf{n}}) \, dt, \tag{11}$$

where the duration of the collective response is from the $t_0$ to $t_1$. The group moving direction is $\hat{\mathbf{V}}(t) = \frac{1}{N} \sum_{i=1}^{N} \hat{\mathbf{v}}_i(t)$. Notably, $R$ is the cumulative evaluation of the group response. The lower the value of $R$, the higher the group's responsiveness. The value of $R$ ranges from 0 to 2. $R = 0$ means that all individuals follow the new direction $\hat{\mathbf{n}}$ without any delay during the response process, whereas $R = 2$ suggests the worst response to the informed individual, i.e., the group moves in the direction opposite to $\hat{\mathbf{n}}$.

### Swarm robotics system

To show the advantages of BOC-based interaction in facilitating the emergence of collective responses, we developed a swarm robotic system (i.e., the SwarmBang system). The SwarmBang system comprises three primary components: a server computer, a motion capture system, and a large number of swarm robots. The server computer, equipped with a radio transmitter, is tasked with model computing, simulating local visual perception, and transmitting real-time control commands to the robots. The motion capture system is used to locate the position of each robot. A detailed description of the robot's components and swarm robotic system can be found in Supplementary Figs. 16, 19, respectively.

The workflow of our validation system begins with the measurement of each SwarmBang robot's position and heading via the NOKOV motion capture system processed on the server computer (see Supplementary Note 6 and Supplementary Fig. 17 for detailed information). Next, the server computer simulates the local vision-based sensing of the robots through the pinhole camera model (see Supplementary Note 6 for detailed information). Then, the server computer determines the desired heading $\theta_d$ for each robot based on the

swarm model and the simulated local perception. After the desired heading of each robot is obtained, the desired angular speed $\omega$ is calculated as the $\omega = \min(\frac{\Delta\theta}{\Delta t}, \omega_{\max}^{\text{robot}})$, where the $\Delta\theta = \theta_d(t+1) - \theta(t)$ and $\omega_{\max}^{\text{robot}}$ is the maximum angular speed in the robotic experiments (see Supplementary Table 3 for the parameters used in the real robot experiments). Following that, the server computer calculates the desired linear and angular speed of each robot and broadcasts to all the robots via a wireless communication module through a customized communication protocol with a fixed time interval (the details of this protocol can be found in Supplementary Note 6 and Supplementary Fig. 20). Upon receiving these desired velocities, each robot then calculates the speed of its left and right wheels based on the kinematic model of the differential-drive platform in the robots (the motion control of the SwarmBang robot can be found in Supplementary Note 6).

### Simulation of vision-based sensing of robots via the pinhole camera model

The pinhole camera model is a commonly used and effective technique for simulating visual perception[2,70,71]. The pinhole camera model describes the mathematical relationship between the coordinates of a point $p = (p_x, p_y, p_z)$ in three-dimensional space and its projection onto the image plane. The projection of this point on the image plane is calculated as $(u_x = f \cdot \frac{p_x}{p_z}, u_y = f \cdot \frac{p_y}{p_z})$, where $f$ represents the focal length of the camera. In our swarm robotic validation system, each robot is regarded as an ellipsoid with the same aspect ratio as the real SwarmBang robot. Using the pinhole camera model, we can obtain the projections of the robots (ellipsoids) onto the image plane of the simulated camera, where the shape of the projection varies as the adjustments of the robot's movement. To approximate the magnitude of BOC from the simulated images, we used the rectangular formed by the maximum $X$ and $Y$ ranges of the robot's projection to generate the simulated bounding box and then estimated the BOC by the variation in the area of these bounding boxes. More detailed information is provided in Supplementary Note 6 and Supplementary Fig. 18.

## Data availability

The datasets of spontaneous collective U-turn observed in rummy-nose tetra groups (*Hemigrammus rhodostomus*) referred to ref. [29] are available at: https://doi.org/10.5061/dryad.9m6d2.

## Code availability

All source codes and minimal datasets[72] related to this work can be found at the link: https://github.com/DerekZhengEvosil/Body_orientation_change_of_neighbors_leads_to_scale_free_correlation.

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

## Acknowledgements

We thank Professor Yandong Xiao for the insightful discussions. X.P. is supported in part by the National Natural Science Foundation of China under Grant 62076203, and in part by the Fundamental Research Funds for the Central Universities under Grant G2024KY0601.

## Author contributions

X.P. designed and managed the whole project. Z.Z. performed all the real data analysis and performed all the analytical/numerical calculations and simulations. X.P., X.L., Z.Z., and Y.X. built the swarm robotic system. Y.X. and Z.Z. performed swarm robotic experiments. Z.Z. wrote the manuscript. X.P., X.L., and Y.T. revised the manuscript.

## Competing interests

The authors declare no competing interests.

## Additional information

**Zhicheng Zheng**[1], **Yuan Tao**[1], **Yalun Xiang** ⑩[1], **Xiaokang Lei** ⑩[2] **& Xingguang Peng** ⑩[1]✉

[1]School of Marine Science and Technology, Northwestern Polytechnical University, Xi'an, Shaanxi 710072, P. R. China. [2]School of Information and Control Engineering, Xi'an University of Architecture and Technology, Xi'an, Shaanxi 710055, P. R. China. ✉e-mail: pxg@nwpu.edu.cn

