## [Peer Review File · Nature Communications]

Body Orientation Change of Neighbors Leads to Scale-Free Correlation in Collective Motion

Corresponding Author: Professor Xingguang Peng

Version 0:

Reviewer comments:

Reviewer #1

(Remarks to the Author)

If I understand correctly, the authors claim that models of collective motion assume that all group members can perceive the location and motion direction of all other individuals in a neighbourhood, which is not realistic due to occlusions. This is not entirely true (see DOI:10.1109/ACCESS.2022.3158758, e.g.), but I agree that it is worth testing model results with more realistic perceptual information. They also claim that other "contextual" information should be taken into account to properly model collective motion and suggest what follows: "Inspired by the theory success of body orientation (or attitude) coordination investigated in macro-level swarm models [24, 25], we hypothesized that the Body Orientation Change (BOC) might be a potential motion characteristic to facilitate the emergence of the collective response." They then perform analysis of fish collective motion where they identify correlations between "leadership" and BOC magnitude. They then validate the findings in simulations and with 50 wheeled robots. They claim that the data analysis and experimental results "demonstrate" BOC leads to scale-free fast re-orientation of individuals in collective motion.

Some sentences seem to have a circular logic, such as:

"For the empirical analysis, we aim to reveal the relationship between the BOC and leadership in the context of the U-turn. This exploration contributes to elucidating the motion characteristic needed by an individual to become influential and provides insights into the role of BOC in shaping the emergence of collective response within the fish group."

- What is the "empirical analysis"?

- Do you assume that it is necessary to have a leader in U-turn behaviours?

- Do you make the hypothesis that BOC is related to leadership?

- The second sentence is a rephrasing of the first sentence, but it assumes that BOC is indeed the important cue that gives leadership.

Several important statements are formulated in technical terms that require reading detailed method procedures and formalisms. For example:

"Moreover, we quantified the leadership of each individual by calculating the normalized out-degree of each node in the leader-follower network derived from the segments $[T - \tau, T]$, labeled as the $L_i(T, \tau)$ (see Methods)."

- I recommend that the authors rephrase important assumptions in simpler terms that can be readily understood and give details in the Methods section.

The assumptions of the experiments with simulated agents and robots must be clearly set in the main text. For example, in lines 136-146 it is not clear what information each agent receives, what actions it can take, and if there is any viscosity in the medium. From what the authors describe, occlusions and other optical factors seem to be neglected, despite the problem statement of the introduction.

Also, it is unclear if the experiments on collective turning with simulated agents are modelled upon the U-turn data of real fish.

Finally, I did not understand to what extent the experiments with real robots are related to the previous questions because. From the text:

"Similar to the simulation experiments of collective turn, the robotic experiment is to have the robot swarm respond to a single informed robot that exhibits successive sudden turns."

it seems that the behaviour is entirely pre-programmed, meaning that a robot plays the leader role and the others simply react to its BOC.

Therefore, I do not understand what is the purpose of these experiments and what additional information they provide.

I found the manuscript very difficult to read and understand. The problem, solution, and method should be presented more concisely. Important methods and assumptions for the results of the paper should be explicitly explained instead of citing references. It is not clear what precisely BOC means from the perspective of an individual's sensory system and how this relates to the problem statement of the introduction.

Reviewer #2

(Remarks to the Author)

This work presents a study of a novel mechanism to achieve collective response within the context of collective motion. The mechanism is based on body orientation change (BOC), which is a criteria used by the individual to select which neighbor to interact, namely the neighbor whose body orientation changed the most. The analysis is comprehensive and include analysis of data of real fish, two simulated models, and real robots experiments. The main result of the paper is that scale-free interactions emerge when BOC is used.

The work is significant because it helps in shedding light on the mechanisms behind collective response in animals like fish, and also suggest a mechanism to design analogous mechanisms for robot swarms.

I encourage the eventual publication of this article in Nature Communication. However, before doing so, I believe the article needs significant revisions to provide a more convincing argument in the results and also to make it reproducible by other researchers. Please find below the revisions I recommend:

- The principle behind BOC should be clarified already in the abstract and in the introduction. In general, when modeling collective motion, the building blocks that needs to be designed should answer to the following questions: 1. with which neighbor(s) the focal individual interacts? 2. What information is exchanged between neighbors? 3. How is the information received integrated? 4. How is the integrated information converted to motion? In this paper, BOC answers only to question 1. (and also only one neighbor is selected) This is completely ok, but it should be made very clear from the beginning, and also the explanation of the intuitive understanding behind BOC should be clear
- Finding another mechanism yielding to scale-free interaction is a significant contribution. However, the work currently suffers from the critical lack of a baseline implemented in simulation and with real robots. More precisely, we cannot be sure that BOC is the ultimate responsible ingredient yielding to scale-free correlations. Therefore, I suggest the authors to implement a baseline/control experiments in which everything is kept the same, but the single interaction neighbor is selected with another criteria, e.g. a random neighbor within a perception radius, or the closest neighbor. Would this mechanism produce scale-free interactions? If the answer is yes, then BOC is not what yields to this result, but there must be something else. Note that the swarm robotics experiment seem to contain partially this analysis, however it should be performed also or mainly for in pybullet, and results should belong in the main text and not in the supplementary materials.
- The article should review more related work, e.g. other papers in which mechanisms yielding to scale-free correlations was observed (other than criticality). For example this paper: <https://link.springer.com/article/10.1007/s10955-014-1114-8>
- The claim saying that mechanisms other than those based on field of view are "omniscient" is not entirely correct. For example, metric-based mechanism are not omniscient if a 360 degrees vision exist, even if not found in animals it can be implemented on robots with a purely local (non omniscient) mechanism. Biologically-plausible is not the same as omniscient, so a distinction should be drawn when reviewing other mechanisms
- There are lack of details concerning the real robot implementation, which hinders reproducibility. Importantly, it is not clear what computation and sensing is done onboard of the robots, and what is done instead by a central computer after data collection. If most of the computation and sensing is done externally, this reinforces further my previous comment: authors cannot claim the mechanism considered is not omniscient if their implementation on real robot (the main opportunity to prove locality) is itself omniscient. It should also be discussed exactly how BOC can be implemented on robots using purely local sensing and computation (for example, can vision and bounding boxes be used to calculate BOC over consecutive frames?). The latter is important because it may suggest what is the perceptual mechanism used by animals to do the same
- The definition of alpha is not entirely clear. Does it only change the "frontal preference" or is it also used to tune the wideness of the field of view? If the field of view is fixed, how can alpha model the full 360 degrees spectrum of frontal preference? (neighbors in the back would be anyway ignored). Please explain
- I recommend to re-order the arrangement or labeling of the panels in Fig 2 in a more logical way
- It would be interesting to know if the same results would hold in case the initiators (e.g. in the collective motion with turn simulation) are not positioned at the border
- In the methods, it would be great to give some intuitive explanation of what "normalized out-distance [44]" is.
- Page 8 line 368: prioritize  select (prioritize means giving some weight also to other elements)

Reviewer #3

(Remarks to the Author)

In this paper, first body oriented change (BOC) from fish's eye perspective is analyzed. For this purpose, fish data is analyzed and reported in paper. A flocking method is proposed based on BOC-based interactions. The performance of this new method is compared with the classical Vicsek model using accuracy and responsiveness. In addition to simulation based experiments, the method is also validated using a swarm of 50 physical robots.

The paper is written clearly and relatively easy to understand. Here are my comments:

* I would definitely change the title of the paper, "Body Orientation Change Emerges Scale-Free Correlation in Collective Motion" that was really hard for me to understand before reading the paper.

* Additionally, as shown in Fig.2-(b3), it is evident that the larger groups exhibited a more stable direction of information transfer with the BOC-based interaction.  How did you come up with that result?

* In several places of the paper, you criticize use of velocity in swarm models. "Secondly, interaction mechanisms adopted in mathematical models rely on physical measurements that might not be able to be acquired through local perception. For example, velocity [14], bearing change [37]," but in your model you also use velocity as described by: "the reaction rule involves copying the neighbor's velocity in the simulation experiments of collective turn". Can you please elaborate on that?

* I like the fact that there are real robot experiments but i think that the real robot experiments are oversimplified. As you describe in the supplementary material: "the experimental set-up is in the pseudo-distributed manner, which means the positions and headings of each robot are obtained from the motion capture system. Besides that, the server computer yields the desired angular and linear velocities for each robot based on the interaction rules. Then, the desired angular and linear speeds are transmitted to each robot through 2.4g wireless communication. Finally, each robot executes the received motion commands". As far as I understood, the computation is not even performed by the robots, all is centralized. I think the major problem is not that. In the paper, in several places you mention about vision-based sensing and collective motion. I know that you cannot add a camera to your robots but it would be great if you can simulate vision-based sensing in your robots and redo the experiments. In this way, you can justify the use of your model.

Version 1:

Reviewer comments:

Reviewer #1

(Remarks to the Author)

The revised version of the manuscript substantially improved in clarity and strength thanks to the additional explanations and control experiments. The results from behavioural analysis of fish, of simulated agents driven by the hypothesised control laws, and of physical robots provide strong evidence that Body Orientation Change could be a strong visual cue in supporting scale-free collective motion with faster signal propagation speed than other perceptual cues. As far as I could understand (see more about this below), the method and data are sound and the claims are supported by the experimental evidence. The revised manuscript and additional code in principle provide sufficient information to reproduce the results, although I haven't tested the code myself. The results and the proposed algorithms could be relevant not only for biologists and complex system scientists, but also for computer scientists in swarm robotics.

However, in my opinion the manuscript still requires substantial editing before publication. I still had a hard time to properly understand or interpret the meaning of some sentences because of writing style and syntax infelicities. For example, the article "the" is used excessively and improperly, which often led me to incorrectly think that the authors referred to a new or a specific method; for example, at line 82: "we conduct the empirical data analysis" made me believe that by "empirical data analysis" they refer to a specific method developed by themselves, whereas they probably meant to say "we analyse experimental data from collection motion of fish". Analogously, the improper use of "a, an" and "the" made me believe that they assume the presence of only one factor, such as in line 52 "we aimed to reveal the visual cue..." instead of "we aimed at revealing what visual cue...". Also, at line 145: "inspire us to develop the pairwise interaction mechanism" induced me to understand that they propose a new algorithm or principle, whereas they probably meant to say "led us to hypothesise that pairwise interactions..." or something similar.

Other misleading writing errors include the use of the verb "emerge" as a transitive verb, such as in the title and at line 144, which makes it hard to understand what the claim is. I guess that the authors meant to say something like "displays, leads to, supports", or something similar. Other infelicities include "verify an hypothesis" (hypotheses can only be tested or at best falsified, certainly not verified), "our experiments with robots demonstrate" (demonstrations are best reserved to mathematical proofs; robotic results can show, support, or at best validate something).

Even after reading this second version of the manuscript, I still do not understand what the authors precisely mean by "scale-free correlation": do they mean scale-free behavioural correlation, they way Cavagna et al used this term in their paper on bird flock analysis? It is important to define precisely what correlations they refer to (what is correlated to what) before making statements of scale-free correlation. The authors do make some sort of definition of correlation 187-190, but I failed

to understand what they mean by "we calculated the correlation length (what is the correlation length?) of different flock (flock refers to birds, not to fish) sizes. The point where the correlation function (what is the correlation function?) reaches zero, defines the correlation length (so now we have two undefined terms that define each other). The correlation function C_r represents the velocity fluctuations (does C represent a correlation or a fluctuation?) as a functions of distance r in the group (distance between what?)" The text in brackets is my own text to explain how I got rapidly lost by the explanation.

There are several other cases where the authors discuss important factors without first defining them or without defining them at all. For example, the use of the correlation matrix is described before the definition of how the matrix is constructed (it is possible to understand it from Figure 1, but that figure is referred to only later in the text. Another example: they refer to spin behaviour and spin results, but never define what such behaviour is (at line 160 they point to Figure 2a, but I could not spot any spinning behaviour --whatever that is supposed to be). Other examples: line 117 "absence of frontal preference ($\alpha = 0$)" without defining what frontal preference and α are; line 163: "Additionally the polarization sharply decreased" without explaining what the polarization is; line 175 "the curve formed by the transfer distance d_{max} " without ever explaining what that is.

Finally, the Discussion section needs to be drastically reduced to a concise discussion of the results (significance, limitations, possible developments etc.). As it stands it includes a recapitulations of some results, state of the art, and repetition of claims and considerations already made elsewhere in the manuscript.

Reviewer #2

(Remarks to the Author)

Dear authors,

after reviewing the changes you made to the manuscript, I am exceptionally impressed and satisfied by the extensiveness of the revision. In my opinion, the article is now greatly improved, also in response to the other reviewer's comments. I therefore I have no more objections to publication.

Reviewer #3

(Remarks to the Author)

In this paper, authors propose an algorithm based on body orientation change of neighbors. They implemented this algorithm both numerically and using simulated robots. This is the second round of the review process. I had several concerns about the paper and implementation with simulated robots. All my concerns are satisfied and I think that the paper is good enough to be published.

Version 2:

Reviewer comments:

Reviewer #1

(Remarks to the Author)

The authors have clarified all my remaining questions and substantially improved the clarity of the manuscript.

Response to Reviewer #1

Point 1.0: If I understand correctly, the authors claim that models of collective motion assume that all group members can perceive the location and motion direction of all other individuals in a neighbourhood, which is not realistic due to occlusions. This is not entirely true (see DOI:10.1109/ACCESS.2022.3158758, e.g.), but I agree that it is worth testing model results with more realistic perceptual information. They also claim that other "contextual" information should be taken into account to properly model collective motion and suggest what follows: "Inspired by the theory success of body orientation (or attitude) coordination investigated in macro-level swarm models [24, 25], we hypothesized that the Body Orientation Change (BOC) might be a potential motion characteristic to facilitate the emergence of the collective response." They then perform analysis of fish collective motion where they identify correlations between "leadership" and BOC magnitude. They then validate the findings in simulations and with 50 wheeled robots. They claim that the data analysis and experimental results "demonstrate" BOC leads to scale-free fast re-orientation of individuals in collective motion.

Response: We thank Reviewer #1 very much for reviewing our work and her/his insightful assessment of our work. First, we would like to express our gratitude for providing the important reference (DOI:10.1109/ACCESS.2022.3158758) to our attention, which help us gain more comprehensive understanding of the recent research status of vision-based swarm models. We have rephrased the relevant sentences and cited this important reference in the revised manuscript (see **main text: page 7, lines 299-301**):

“For example, recent research by Schilling *et al.* proposed the perceptually realistic visual neighbor selection model, revealing the significant impacts of occlusions on inter-agent distances and velocity alignment⁴⁴.”

Point 1.1: Some sentences seem to have a circular logic, such as:

"For the empirical analysis, we aim to reveal the relationship between the BOC and leadership in the context of the U-turn. This exploration contributes to elucidating the motion characteristic needed by an individual to become influential and provides insights into the role of BOC in shaping the emergence of collective response within the fish group."

- What is the "empirical analysis"?

- Do you assume that it is necessary to have a leader in U-turn behaviours?

- Do you make the hypothesis that BOC is related to leadership?

- The second sentence is a rephrasing of the first sentence, but it assumes that BOC is indeed the important cue that gives leadership.

Response: We thank Reviewer #1 for this critical comment and apologize for the circular logic in the previous manuscript. In the subsection titled as the “**empirical analysis**” of the previous manuscript, we reviewed the fundamental information about the real fish data, such as the source of the U-turn data and the phenomenon of U-turn event, with the intention of facilitating the understanding of the empirical (data) analysis, which may overlap with the information in the following subsection and further yield the circular logic. Hence, in the new version of manuscript, we have removed the sentences with circular logic and the subsection titled as the “**empirical analysis**”. Instead, we have included these basic information of real fish data into the subsection titled as the “**Individuals with larger Body Orientation Change often take on the leading role during the collective U-turn**” and refined sentences to enhance the clarity of the empirical data analysis (see **main text: pages 2-3, lines 81-96**):

“To reveal the significant role of Body Orientation Change (BOC) in the biological swarm, we conduct the empirical data analysis based on spontaneous collective U-turn observed in rummy-nose tetra groups (*Hemigrammus rhodostomus*) recorded in a circular tank^{16,28,29}. In this controlled environment, the fish groups exhibit movement predominantly in two directions: clockwise and counterclockwise. A U-turn event is initiated when a single fish, acting as the leader, abruptly changes its swimming direction. This sudden maneuver serves as a signal for the rest of the group, leading to rapid and cohesive directional change across the entire fish school.

In this work, we collected a total of 44 and 400 tracks of collective U-turns from groups of 10 and 8 fish, respectively. (Detailed methods for U-turn detection are provided in Supplementary Sec. 1, and the overview of U-turn trajectories can be found in Supplementary Fig. 3 to Fig. 8). The empirical data analysis seeks to uncover the relationship between BOC and leadership during U-turns, shedding light on the significant role of BOC in achieving the collective response. Notably, the empirical data analysis of this work is based on two key assumptions: first, a leader is present during U-turn events, typically initiating the abrupt maneuver in the group (see Supplementary videos 1-2); Second, leadership within the group is associated with certain visual or motion cues of neighbors, as it has been elucidated in many other biological swarms^{15,30-34}.”

Below, we addressed and revised the reviewers' comments of **Point 1.1** in detail.

Point 1.1.1: - What is the "empirical analysis"?

Response: We thank Reviewer #1 for this critical comment and apologize for not making this point clearly in the previous version of manuscript. In fact, the empirical (data) analysis is a common approach to understand the spatiotemporal pattern and interaction mechanism behind biological swarms by statistically analyzing real-world observational or experimental trajectory data. In this work, the term “**empirical analysis**” refers to the correlation analysis between the BOC and the leadership based on real fish data of collective U-turn behavior observed in rummy-nose tetra groups (*Hemigrammus rhodostomus*). The empirical data analysis of this work comprises three primary steps: Firstly, we determined the leadership of each individual during U-turn events by leveraging a state-of-the-art leader-follower relationship analysis method. Secondly, we introduced a quantitative method to characterize the body orientation change by computationally reconstructing the scene from the fish's perspective. Thirdly, we revealed the relationship between the leadership and the BOC by conducting the Spearman correlation analysis. The empirical data analysis provides insights into the role of BOC in shaping the emergence of collective response within the fish group. To clarify the meaning of empirical data analysis, we have added the following sentences in the new version of manuscript (see **main text: page 2, lines 91-92**):

“The empirical data analysis seeks to uncover the relationship between BOC and leadership during U-turns, shedding light on the significant role of BOC in achieving the collective response.”

Point 1.1.2: - Do you assume that it is necessary to have a leader in U-turn behaviours

Response: We thank Reviewer #1 for this critical comment. Yes, we assumed that there is a leader in U-turn behaviors, as each event of the U-turn is often initiated by one single fish. Specifically, the U-turn behavior in rummy-nose tetra groups (*Hemigrammus rhodostomus*) is recorded in a circular tank, the fish exhibit movement predominantly in two directions: clockwise and anticlockwise. A U-turn is initiated when a single fish, typically assumed as a leader, abruptly changes its swimming direction. This abrupt turn acts as a signal for the rest of the group, triggering a rapid and cohesive change in direction for the entire fish group. To

clarify the process of U-turn behavior, we have added the videos of typical U-turn behavior in the group size of 8 and 10 in the supplementary information (see **Supplementary videos 1-2** for the detailed process of U-turn behavior). Additionally, we have refined the sentences in the new version of the manuscript to clarify the process of U-turn behavior and this important assumptions:

- see **main text: page 2, lines 83-87:**

“In this controlled environment, the fish groups exhibit movement predominantly in two directions: clockwise and counterclockwise. A U-turn event is initiated when a single fish, acting as the leader, abruptly changes its swimming direction. This sudden maneuver serves as a signal for the rest of the group, leading to rapid and cohesive directional change across the entire fish school.”

- see **main text: page 2, lines 93-95:**

“Notably, the empirical data analysis of this work is based on two key assumptions: first, a leader is present during U-turn events, typically initiating the abrupt maneuver in the group (see **Supplementary videos 1-2**);”

Point 1.1.3: - Do you make the hypothesis that BOC is related to leadership?

Response: We thank Reviewer #1 for this critical comment. In fact, we hypothesized that the leadership within the group is associated with certain visual or motion cues of neighbors, which has been elucidated in many other biological swarms, for example:

- Herbert-Read J E, Perna A, Mann R P, et al. Inferring the rules of interaction of shoaling fish, *Proceedings of the National Academy of Sciences*, 2011, 108(46): 18726-18731.
- Lukeman R, Li Y X, Edelstein-Keshet L. Inferring individual rules from collective behavior, *Proceedings of the National Academy of Sciences*, 2010, 107(28): 12576-12580.
- Mann R P, Herbert-Read J E, Ma Q, et al. A model comparison reveals dynamic social information drives the movements of humbug damselfish (*Dascyllus aruanus*), *Journal of the Royal Society Interface*, 2014, 11(90): 20130794.
- Nagy M, Ákos Z, Biro D, et al. Hierarchical group dynamics in pigeon flocks, *Nature*, 2010, 464(7290): 890-893.
- Puy A, Gimeno E, Torrents J, et al. Selective social interactions and speed-induced leadership in schooling fish, *Proceedings of the National Academy of Sciences*, 2024, 121(18): e2309733121.

Hence, in the empirical data analysis, we hypothesized the potential relationship between BOC and leadership, while also assumed that two additional motion cues, i.e., the bearing change and distance might be related to the emergence of leadership during U-turn events. However, through the same analysis procedure, we found that only the BOC has the strong correlation with leadership during the U-turn events, whereas the other two motion cues, i.e., bearing change and the distance have little impact on the emergence of leadership in real fish group. To clarify this important assumption in the empirical data analysis, we have added the following sentences in the new version of manuscript (see **main text: pages 2-3, lines 95-96**):

“Second, leadership within the group is associated with certain visual or motion cues of neighbors, as it has been elucidated in many other biological swarms^{15,30-34}.”

Point 1.1.4- The second sentence is a rephrasing of the first sentence, but is assumes that BOC is indeed the important cue that gives leadership.

Response: We thank Reviewer #1 for this critical comment and apologize for the circular logic of basic information of empirical data analysis in the previous version of manuscript. In the new version of manuscript, we have removed the sentences with circular logic and the subsection titled as the “**empirical analysis**”. Instead, we have included these basic information of real fish data into the subsection titled as the “**Individuals with larger Body Orientation Change often take on the leading role during the collective U-turn**” (see **main text: page 2, lines 81-90**). Additionally, we have clarified the assumptions between the BOC and leadership in the new version of manuscript (see **main text: page 2, lines 90-96**).

In summary, as the concerns raised in **Point 1.1.1-1.1.4**, we have added the following sentences to clarify the important assumptions of empirical data analysis in the new version of manuscript (see **main text: page 2, lines 81-96**):

“To reveal the significant role of Body Orientation Change (BOC) in the biological swarm, we conduct the empirical data analysis based on spontaneous collective U-turn observed in rummy-nose tetra groups (*Hemigrammus rhodostomus*) recorded in a circular tank^{16,28,29}. In this controlled environment, the fish groups exhibit movement predominantly in two directions: clockwise and counterclockwise. A U-turn event is initiated when a single fish, acting as the leader, abruptly changes its swimming direction. This sudden maneuver serves as a signal for the rest of the group, leading to rapid and cohesive directional change across the entire fish school.

In this work, we collected a total of 44 and 400 tracks of collective U-turns from groups of 10 and 8 fish, respectively. (Detailed methods for U-turn detection are provided in Supplementary Sec. 1, and the overview of U-turn trajectories can be found in Supplementary Fig. 3 to Fig. 8). The empirical data analysis seeks to uncover the relationship between BOC and leadership during U-turns, shedding light on the significant role of BOC in achieving the collective response. Notably, the empirical data analysis of this work is based on two key assumptions: first, a leader is present during U-turn events, typically initiating the abrupt maneuver in the group (see Supplementary videos 1-2); Second, leadership within the group is associated with certain visual or motion cues of neighbors, as it has been elucidated in many other biological swarms^{15,30-34}.”

Point 1.2: Several important statements are formulated in technical terms that require reading detailed method procedures and formalisms. For example:

“Moreover, we quantified the leadership of each individual by calculating the normalized out-degree of each node in the leader-follower network derived from the segments $[T - \tau, T]$, labeled as the $L_i(T, \tau)$ (see Methods).”

- I recommend that the authors rephrase important assumptions in simpler terms that can be readily understood and give details in the Methods section.

Response: We thank Reviewer #1 for this valuable suggestion. Based on her/his suggestion, we have rephrased the important statements or assumptions in the simpler and intuitive way throughout the manuscript. For examples:

For the important statements and assumptions in the empirical data analysis, we have rephrased the process of biological data analysis and included the relevant technical formalisms in the **Methods**, the major revisions are shown below:

- see the **Results** in **main text: page 3, lines 100-108** (the example noted by the reviewer):

“For each fragment trajectory of U-turns, we could obtain the BOC matrix by calculating the BOC of neighbors from the perspective of each individual (see the geometric definition of BOC in Fig. 5 and detailed calculation process in Methods). Then, we could estimate how noticeable the movements of the focal individual are from the other’s eyes by averaging each column of the BOC matrix, which represents the BOC-based motion salience of each individual (see Methods). Next, we quantified the leadership of each individual based on the leader-follower network derived from the segments of the U-turn trajectory (see Methods). By investigating the correlation between the BOC-based motion salience and leadership, we could evaluate the role of BOC in guiding the group to switch the moving direction.”

- see the **Methods** in main text: page 10, lines 452-459:

“Furthermore, to quantify the motion salience of each individual based on the BOC, we could construct the BOC matrix $G(T, \tau) = [g_{ij}(T, \tau)]_{N \times N}$ from each individual’s view according to the definition of Eq. (1) and derive the motion salience of each individual $G_i(T, \tau)$ by averaging each column of $G(T, \tau)$ (Fig. 1d). The $G_i(T, \tau)$ represents how noticeable the movements of the focal individual are perceived from the other’s eyes based on the BOC. Individuals with larger $G_i(T, \tau)$ exhibit more salient directional changes in their movements during the period of $[T - \tau, T]$. Conversely, lower $G_i(T, \tau)$ suggests more stable movements with fewer directional changes within the $[T - \tau, T]$.”

- see the **Methods** in main text: page 11, lines 489-496:

“The normalized out-distance signifies the hierarchical layer of each node (individual) in the leader-follower network, which is essentially an unweighted directed network. In such a hierarchical network, some individuals act as leaders while others are followers, with directed connections between them, i.e., leaders point to their followers. We leverage the number of connections pointing to other nodes to represent the leadership of the focal individual, which refers to the out-distance (or the out-degree) of nodes in the directed graph. The node (individual) with higher out-degree denotes stronger leadership, as they have more connections pointing to other nodes (followers).”

- see the **Methods** in main text: page 11, lines 502-503:

“For example, as depicted in Fig. 1c,e, fish-1 occupies the top level in the leader-follower network, thus its leadership $L_1 = 1$.”

For the important statements and assumptions in the simulation experiments, we have simplified statement of simulation setup in the **Results** section and included the detailed descriptions with technical formalisms in the **Methods** section or the supplementary information, the major revisions are shown below:

- see the **Results** in the main text: page 4, lines 145-159:

“The empirical findings of BOC inspire us to develop the pairwise interaction mechanism, wherein the focal individual selects the neighbor with the largest magnitude of BOC to interact with. To explore the dynamic of information transfer with BOC-based interaction, we built the swarm model with BOC-based interaction (see Methods for detailed definition) and conducted simulation experiments of collective spin and collective turn in a real-time physics simulation environment (pybullet simulator³⁵). On the one hand, the collective spin is an idealized and simplified scenario

for studying information transfer of BOC-based interaction, which requires individuals to rapidly propagate the spinning state initiated by one informed individual (spinning initiator) throughout the group. On the other hand, the collective turn requires individuals to respond to the sudden directional change initiated by one informed individual (turn initiator), posing the complex and realistic information propagation scenario to BOC-based interaction. Detailed descriptions and assumptions of these two simulation experiments can be found in the Methods. Particularly, we involved the random-based interaction as the baseline model to demonstrate the advantage of BOC-based interaction (see Supplementary Sec.5 for the detailed definition).”

- see the **Methods** in main text: pages 8-9, lines 375-412:

“In this work, we conduct the simulation experiments of collective spin and collective turn in the pybullet simulator, where each individual is a simulated two-wheel differential robot in the pybullet environment. The pybullet simulator provides a powerful physics engine that accurately simulates the kinematics of robot motion and physical collisions, which facilitates the application of BOC-based interaction in swarm robotics. Particularly, the occluded individuals are ignored in the neighbor selection (see detailed approaches for identifying the occluded neighbor in Supplementary Sec. 4 and Supplementary Fig. 15).

For the simulation experiments of collective spin, we assumed that each simulated robot has two motion states: stationary and spinning. In the stationary state, both linear and angular speeds of simulated robots are set to 0. In the spinning state, the angular speed is set to the maximum ω_{max}^{sim} (see Supplementary Table 1 for parameters selection), while the linear speed remains 0, yielding the rotational movement of robots. The simulated robots are required to swiftly transfer the spinning state triggered by a single informed individual (spinning initiator) throughout the entire group. The informed individual starts spinning at a pre-set activation time (the 25th step in our simulation) and stops spinning after completing the 2π rotation, which is selected at the forefront of the group. Each simulated robot determines the BOC of its neighbors within a perception range R_{visual} and selects the neighbor with the maximum BOC to react. The reaction rule is to switch between stationary and spinning states based on the state of the selected neighbor. Specifically, if the selected neighbor is in the spinning state, the focal individual also enters the spinning state and starts to spin with the maximal angular speed; if the selected neighbor is in the stationary state, the focal individual switches to the stationary state and stops spinning. The simulation videos of collective spin can be found in Supplementary videos 3-4. Unless otherwise specified, the parameter selections for the BOC and random-based interaction can be found in Supplementary Table 1. The illustration of the simulation procedure of collective spin can be found in Supplementary Fig.26a.

For the simulation experiment of collective turn, the simulated robots are required to respond to a sudden turn initiated by one informed individual (turn initiator). The informed individual, positioned at the forefront of the group, turns with an angle θ info relative to the group’s movement direction at a pre-set moment (the 50th step in our simulation). Each robot calculates the BOC of its neighbors within a perception range R_{visual} and reacts to the neighbor with the maximum BOC. Due to the challenges in acquiring neighbors’ velocity from the first-person view¹⁹, we assumed that the focal individual could access the neighbors’ velocity. The reaction rule in the simulation experiments of collective turn is the commonly used velocity alignment⁴⁰, i.e., average the headings of the selected neighbor and the robot’s own heading (Eq. (7)). The simulation videos of collective turn can be found in Supplementary videos 5-8. Unless otherwise specified, parameter selections in the simulation experiments of collective turn for BOC and random-based interaction

can be found in Supplementary Table 2. The illustration of the simulation procedure of collective turn can be found in Supplementary Fig.26b.”

- see main text: page 4, lines 174-177:

“Furthermore, we analyzed the information transfer speed V_s in the group with BOC-based interaction. By fitting the slope of the curve formed by information transfer distance d_{trans} and the spinning lag (see Supplementary Sec.3 and Supplementary Fig.27 for the detailed definition), we could estimate the information transfer speed V_s in the collective spin.”

- see main text: page 4, lines 188-193:

“The point where the correlation function first reaches zero ($C(r_0) = 0$) defines the correlation length. The correlation function $C(r)$ represents the velocity fluctuations as a function of distance r in the group. As the increase of r , $C(r)$ gradually exhibit the decaying transition from strong correlation to weak correlation in velocity among individuals and eventually crossed the x-axis ($C(r) < 0$) (see Supplementary Sec.3.8 for the detailed definition).”

- see main text: page 5, lines 230-232:

“The information transfer speed is estimated by the slope of the curve formed by the d_{trans} and the turning delay in the collective turn (see Supplementary Sec.3 and Supplementary Fig.28 for detailed information).”

Point 1.3: The assumptions of the experiments with simulated agents and robots must be clearly set in the main text. For example, in lines 136-146 it is not clear what information each agent receives, what actions it can take, and if there is any viscosity in the medium.

Response: We thank Reviewer #1 for this critical comment and apologize for not clearly presenting the assumptions and setups of the simulation experiments (content in lines 136-146 of the previous manuscript). In this work, we conducted two kinds of simulation experiments: **1) collective spin; 2) collective turn** to investigate the information transfer of BOC-based interaction. The simulation experiments are conducted in the real-time physics simulation environment (pybullet simulator), as it allows for the simulation of kinematic motion control of real robots and physical collisions among individuals. Each individual in the pybullet is a two-wheeled differential simulated robot (**Fig.R7a**), which has the same size, shape and kinematics characteristics as the real robot used in our swarm robotic experiments. In the simulation of these two experiments, the simulated robots reconstruct the perception of BOC from the first-person perspective based on the positions and headings of neighbors. The action of the simulated robots is conformed to the two-wheel differential kinematic model, which is controlled by the linear and angular speed v and ω . Additionally, the effect of viscosity in the medium are not involved in the simulation experiments, as the influence of viscosity on wheeled robot motion is relatively minor in the terrestrial environment.

Moreover, to better clarify the assumptions and the information each simulated robot receives and the actions it takes in simulations, we presented the diagrams of the simulation experiments for the collective spin and collective turn in **Fig.R1**, and provide detailed descriptions of both simulation experiments as follows:

i. Simulation experiments of collective spin

In the simulation experiments of collective spin, we assume that each simulated robot has two motion states: one is stationary, and the other is spinning. The stationary state means that both the linear and angular speed of the simulated robot are set to 0. The spinning state means that the angular speed is set to the maximum ω_{max}^{sim} (see **Supplementary Table 1** for parameters selection) while the linear speed is set to 0, indicating the rotational movement of robots. The simulated robots are required to swiftly transfer the spinning state triggered by a single informed individual (spinning initiator) throughout the entire group. The informed individual starts spinning at a pre-set activation time and stops spinning after completing a 2π rotation, which is selected at the forefront of the group. Each simulated robot calculates the BOC of its neighbors within a perception range R_{visual} and selects the neighbor with the maximum BOC to react, i.e., the BOC-based interaction. The reaction rule in the collective spin has the simplest manner for the focal individual, which is to switch between stationary and spinning states based on the state of the selected neighbor. Specifically, if the selected neighbor is in a spinning state, the focal individual also enters the spinning state and starts to spin with the maximal angular speed; if the selected neighbor is stationary, the focal individual switches to the stationary state and stops spinning. The simulation videos of collective spin can be found in **Supplementary videos 3-4**. Unless otherwise specified, the parameters selection in simulation experiments of collective spin can be found in **Supplementary Table 1**. To sum up, the simulation of collective spin involves two key steps (**Fig.R1a**):

1. We randomly initialize the positions of the individuals while setting their headings to be the same and all individuals' states to be the stationary at the beginning.
2. The informed individual starts spinning at a pre-set activation time and stops spinning after completing a 2π rotation. Then, we proceed to analyze the simulation results of the collective spin.

ii. Simulation experiments of collective turn

Collective turn refers to the coordinated directional changes within the group, where simulated robots are required to respond promptly to a sudden turn initiated by one informed individual (turn initiator). The local interaction is the same as the experiments of collective spin, that each simulated robot calculates the BOC of its neighbors within a perception range R_{visual} and selects the neighbor with the maximum BOC to react. The reaction rule involves the commonly used velocity alignment, i.e., averaging the headings of the selected neighbors and one's own heading (Eq. (7) in the main text). Several assumptions are made to streamline the study of information propagation of BOC-based interaction in the simulation experiments of the collective turn. On the one hand, we assume the informed individual, positioned at the forefront of the group, turns with an angle θ_{info} relative to the group's movement direction at the pre-set moments. On the other hand, due to the challenges in acquiring neighbors' velocity from the first-person perspective, we assume that the focal individual could access the neighbors' velocity information. The simulation videos of collective turn can be found in **Supplementary videos 5-8**. Unless otherwise specified, the parameters selection in simulation experiments of collective turn can be found in **Supplementary Table 2**. In conclusion, the simulation of collective turn comprised two key steps (**Fig.R1b**):

1. We randomly initialize the positions of the individuals while setting the same headings for all individuals.

2. The informed individual abruptly turns with an angle θ_{info} relative to the group's movement direction at a pre-set moment to trigger the collective turn of the group.

In the new version of manuscript, we have included the brief descriptions of the simulation experiments setup to facilitate the understanding of simulation results (see **main text: page 4, lines 145-159**):

“The empirical findings of BOC inspire us to develop the pairwise interaction mechanism, wherein the focal individual selects the neighbor with the largest magnitude of BOC to interact with. To explore the dynamic of information transfer with BOC-based interaction, we built the swarm model with BOC-based interaction (see Methods for detailed definition) and conducted simulation experiments of collective spin and collective turn in a real-time physics simulation environment (pybullet simulator³⁵). On the one hand, the collective spin is an idealized and simplified scenario for studying information transfer of BOC-based interaction, which requires individuals to rapidly propagate the spinning state initiated by one informed individual (spinning initiator) throughout the group. On the other hand, the collective turn requires individuals to respond to the sudden directional change initiated by one informed individual (turn initiator), posing the complex and realistic information propagation scenario to BOC-based interaction. Detailed descriptions and assumptions of these two simulation experiments can be found in the Methods. Particularly, we involved the random-based interaction as the baseline model to demonstrate the advantage of BOC-based interaction (see Supplementary Sec.5 for the detailed definition).”

Additionally, we have included the schematic diagram of experiments of collective spin and collective turn in **Supplementary Fig.26** and added the detailed descriptions of simulation setup in the **Methods** section (see **main text: pages 8-9, lines 375-412**):

“In this work, we conduct the simulation experiments of collective spin and collective turn in the pybullet simulator, where each individual is a simulated two-wheel differential robot in the pybullet environment. The pybullet simulator provides a powerful physics engine that accurately simulates the kinematics of robot motion and physical collisions, which facilitates the application of BOC-based interaction in swarm robotics. Particularly, the occluded individuals are ignored in the neighbor selection (see detailed approaches for identifying the occluded neighbor in Supplementary Sec. 4 and Supplementary Fig. 15).

For the simulation experiments of collective spin, we assumed that each simulated robot has two motion states: stationary and spinning. In the stationary state, both linear and angular speeds of simulated robots are set to 0. In the spinning state, the angular speed is set to the maximum ω_{max}^{sim} (see Supplementary Table 1 for parameters selection), while the linear speed remains 0, indicating the rotational movement of robots. The simulated robots are required to swiftly transfer the spinning state triggered by a single informed individual (spinning initiator) throughout the entire group. The informed individual starts spinning at a pre-set activation time (the 25th step in our simulation) and stops spinning after completing the 2π rotation, which is selected at the forefront of the group. Each simulated robot determines the BOC of its neighbors within a perception range R_{visual} and selects the neighbor with the maximum BOC to react. The reaction rule is to switch between stationary and spinning states based on the state of the selected neighbor. Specifically, if the selected neighbor is in the spinning state, the focal individual also enters the spinning state and starts to spin with the maximal angular speed; if the selected neighbor is in the stationary state, the focal individual switches to the stationary state and stops

spinning. The simulation videos of collective spin can be found in Supplementary videos 3-4. Unless otherwise specified, the parameter selections for the BOC and random-based interaction can be found in Supplementary Table 1. The illustration of the simulation procedure of collective spin can be found in Supplementary Fig.26a.

For the simulation experiment of collective turn, the simulated robots are required to respond to a sudden turn initiated by one informed individual (turn initiator). The informed individual, positioned at the forefront of the group, turns with an angle θ_{info} relative to the group's movement direction at a pre-set moment (the 50th step in our simulation). Each robot calculates the BOC of its neighbors within a perception range R_{visual} and reacts to the neighbor with the maximum BOC. Due to the challenges in acquiring neighbors' velocity from the first-person view¹⁹, we assumed that the focal individual could access the neighbors' velocity. The reaction rule in the simulation experiments of collective turn is the commonly used velocity alignment⁴⁰, i.e., average the headings of the selected neighbor and the robot's own heading (Eq. (7)). The simulation videos of collective turn can be found in Supplementary videos 5-8. Unless otherwise specified, parameter selections in the simulation experiments of collective turn for BOC and random-based interaction can be found in Supplementary Table 2. The illustration of the simulation procedure of collective turn can be found in Supplementary Fig.26b.”

Moreover, to better distinguish the difference between these two experiments, we have added the following sentences and velocity updating equations of collective spin (the Eq. (6) compared with the Eq. (7) in main text). Please see the **page 12, lines 523-532** in **Methods** section:

“For the simulation experiments of collective spin, the velocity of the individual i is calculated as follows:

$$\hat{v}_i(t + 1) = \mathbf{R}(\theta_i^{rot})\hat{v}_i(t)$$

where the $\mathbf{R}(\theta_i^{rot})$ is the rotation matrix that characterizes the rotational motion of individuals, which is defined as the $\mathbf{R}(\theta_i^{rot}) = \begin{pmatrix} \cos(\theta_i^{rot}) & -\sin(\theta_i^{rot}) \\ \sin(\theta_i^{rot}) & \cos(\theta_i^{rot}) \end{pmatrix}$. $\theta_i^{rot} = \omega_i \Delta t$ is the rotation angle of individual i during the time interval Δt . ω_i represents the angular speed of the simulated robots, which is determined by the state of the selected neighbor. If the selected neighbor is in the stationary state, then $\omega_i = 0$ and $\theta_i^{rot} = 0$, indicating the focal individual i stops spinning. Conversely, if the selected neighbor is in the spinning state, the $\omega_i = \omega_{max}^{sim}$ (see Supplementary Table 1 for parameters selection in the collective spin), signifying the rotational movement of individual i .”

Point 1.4: From what the authors describe, occlusions and other optical factors seem to be neglected, despite the problem statement of the introduction.

Response: We thank Reviewer #1 for this critical comment and apologize for not making this point clear in the previous version of our manuscript. In fact, we have involved the effect of visual occlusion from the first-person perspective of individuals in this work. For example, the occluded individuals are discarded in calculating the BOC matrix for empirical data analysis and the neighbor selection process in the simulation experiments of BOC-based interaction. To clarify the involvement of visual occlusion in this work, we detailed the approaches of identifying the occluded neighbor from the first-person perspective of individuals as follows:

- **Involvement of visual occlusion from the first-person perspective**

In this work, we assume that individuals are non-transparent ellipses, meaning that neighbors in close proximity to the focal individual may occlude neighbors that are further away. This occlusion prevents the focal individual from perceiving the distant neighbors within the visual perception range.

To do that, we adopted the algorithm provided in the research (*Spatial Structure and Information Transfer in Visual Networks*- <https://doi.org/10.3389/fphy.2021.716576>, code can be found in <https://zenodo.org/records/4983257>). Specifically, to determine the visible and occluded neighbors in our simulation, the computation comprised four steps: Firstly, we identify the positions and angles of the tangent points (black points shown in **Fig.R2a**) where the rays originating from the focal individual's eye (red point shown in **Fig.R2a**) touch on the neighboring ellipses. Secondly, we calculate the intersection point of rays originating from the eye of the focal individual i through the tangent points on j_1 with the outlines of ellipses k (the red cross shown in **Fig.R2a**). If the intersection point exists, it means the occlusion occurs and we then remove the tangent point closest to the intersection point. Otherwise, the absence of intersections implies no occluded individual is in this certain direction. Thirdly, we numerically sort the angular positions of left tangent points for the focal individual in increasing order. Finally, for each segment (i.e., visual field) delimited by two ordered tangent points (light blue regions in **Fig.R2a**), we projected the bisectors (purple lines in **Fig.R2a**) of these segments to ascertain which ellipse is closest in this segment to find out the belongings. Through computations of these four steps, we can distinguish the distant neighbors that are occluded by nearby ones from the first-person perspective of the focal individual.

As shown in **Fig.R2b-c**, we compared the impact of the presence or absence of visual occlusion on the process of neighbor selection. Specifically, with the involvement of visual occlusion, distant individuals (colored by the dark grey) are obscured by closer individuals, thus precluding being selected by the focal individual (**Fig.R2b**). However, with the absence of the visual occlusion, the focal individual can select any neighbors within its perception range (**Fig.R2c**), which is the common perception setup in swarm models, e.g., the metric interaction.

To clarify the involvement of visual occlusion in this work, we have included the above descriptions in **Supplementary Sec.4** and **Supplementary Fig.15**. Also, we have added the following sentences in the revised manuscript:

- see main text: page 7, lines 315-318:

“Departing from the status quo, our work reveals the significance of an explicit and context-related visual cue, Body Orientation Change (BOC), attempting to explore the more fundamental mechanisms underlying the emergence of the collective response from the first-person perspective of individuals with realistic perception factors such as the aspect ratio and visual occlusions.”

- see main text: page 9, lines 416-419:

“Particularly, we assume that individuals are non-transparent ellipses, meaning that neighbors in close proximity to the focal individual may occlude neighbors that are further away (see detailed approaches for identifying the occluded neighbor in Supplementary Sec.4 and Supplementary Fig. 15).”

- see **main text: page 11, line 507**:

“Here, we consider a group composed of N elliptical **non-transparent** individuals (characterized by the length of the major axis a and the minor axis b) instead of simple particles moving in a two-dimensional plane.”

Point 1.5: Also, it is unclear if the experiments on collective turning with simulated agents are modelled upon the U-turn data of real fish.

Response: We thank Reviewer #1 for this critical comment and apologize for not making this point clear in the previous version of manuscript. Indeed, the simulation experiments of collective turn share similarities with the U-turn behavior observed in real fish, especially when the turning angle θ_{info} of the informed individual is set to 180° . However, it is important to note that the simulation experiment of collective turn is not modeled based on U-turn data of real fish. The reason for setting the turning angle to 180° is to increase the difficulty for the group in responding to the abrupt turn triggered by the informed individual, aiming to further demonstrate the effectiveness of BOC-based interaction. Additionally, the simulation experiments are conducted in the pybullet simulator, with the dynamics and kinematics of individuals conforming to the two-wheel differential mobile robot, instead of being modeled based on the movement kinetics of individual fish.

To clarify the relationship between the simulation experiments of collective turn and the U-turn behavior in real fish, we have added the detail description in the **Results** section of the revised manuscript (see **main text: page 5, lines 205-209**):

“To increase the challenge of the group in responding to the abrupt turn, we set two typical turning angles θ_{info} of the informed individual, i.e., $\theta_{\text{info}} = \frac{\pi}{2}$ and π . While the experiment setup of collective turn shares similarities with the U-turn behavior observed in real fish schools, it is important to note that the experiments are not modeled upon the U-turn data of real fish.”

Point 1.6: Finally, I did not understand to what extent the experiments with real robots are related to the previous questions because. From the text: "Similar to the simulation experiments of collective turn, the robotic experiment is to have the robot swarm respond to a single informed robot that exhibits successive sudden turns." it seems that the behaviour is entirely pre-programmed, meaning that a robot plays the leader role and the others simply react to its BOC. Therefore, I do not understand what is the purpose of these experiments and what additional information they provide.

Response: We thank Reviewer #1 for these valuable comments and apologize for not making this point clear in the previous version of our manuscript. In fact, the real robot experiments are directly related to the previous question as the crucial part of the comprehensive research chain in collective motion, which aims to validate the feasibility of BOC-based interaction under non-ideal and realistic conditions and further demonstrate its advantage of information transfer in real-world applications. Additionally, while the behavior of the leader robot (or the informed robot) is pre-programmed to make sudden turns, the real robot experiments are designed to investigate the emergent collective responses of the other robots with the challenge of motion errors and randomness, which are yielded by the real-time and adaptive motion decisions of robots. In the following, we detailed the purpose of the real robot experiments and their relevance to previous questions, as well as summarized the additional information the real robot experiments provided.

i. Purpose of real robot experiments for the BOC-based interaction

Firstly, incorporating experiments with real robots aims to validate the advantage of BOC-based interaction in real-world contexts. While the simulation results revealed the advantage of BOC-based interaction for information transfer, it is crucial to validate such bio-inspired mechanism under realistic and non-ideal conditions, including motion errors and randomness. These factors can cause incorrect responses or actions of robots, further hindering the emergence of the collective response in swarm robotics. Testing the effectiveness of BOC-based interaction under these non-ideal conditions is essential to demonstrate its robustness and applicability in real-world application of swarm robotics.

Secondly, the real robot experiments are different with the simulation experiments of collective turn. On the one hand, we have the robotic swarm responds to successive abrupt turns of informed robots, posing higher demands on the responsiveness of the local interaction mechanism. On the other hand, in the revised robotic experiments, to reinforce the validation of BOC-based interaction with local visual information, we involved the pinhole camera model to simulate the vision sensing of robots and estimated the BOC based on the simulated bounding boxes (see the **Supplementary Sec.6.2** for detailed information or **Fig.R8**). The new results of robotic experiments can be found in **main text: page 6, lines 248-292** or the **Fig.R13**). Real robotic experiments demonstrated that the BOC-based interaction is not only a theoretical and computationally swarm model but also practically feasible and effective in swarm robotic systems.

To clarify the purpose of the robotic experiments, we have added the following sentences in the new version of manuscript (see **main text: page 6, lines 249-261**):

“To further demonstrate the feasibility of BOC-based interaction in swarm robotics, we adopt the BOC-based interaction into a swarm consisting of 50 SwarmBang robots^{41,42} (see Methods and Supplementary Sec.6 for the brief of the SwarmBang system). While the simulation results revealed the advantage of BOC-based interaction for information transfer, it is crucial to validate such bio-inspired mechanisms under realistic and non-ideal conditions, including motion errors and randomness. These realistic factors may lead to incorrect responses or actions of robots and further hinder the emergence of the collective response in swarm robotics. Different from the simulation experiments, on the one hand, we have the robotic swarm respond to successive abrupt turns of informed individuals, posing the higher demand on the responsiveness of the BOC-based interaction. On the other hand, we simulated the vision-based sensing by the pinhole camera model in the swarm robotic validation system and estimated the BOC based on the simulated camera image (see Methods), which provides the opportunity to validate the BOC-based interaction using simulated visual perception from the view of robots.”

ii. The relationship with the previous questions of real robot experiments

Firstly, in this work, we attempted to establish the comprehensive research chain that extends from biological data to bio-inspired mechanism and further to swarm robotics. Specifically, to bridge the gap between vision-based models and the collective response, we revealed the visual cue that adopted by the individual in biological swarm, i.e., the BOC. The BOC serves as the efficient sensory input for vision-based models to facilitate the capture of neighbor's motion salience from the first-person perspective, which is

crucial to the collective response. Then, we translated it into the bio-inspired swarm model and further applied it into the swarm consisting of 50 real robots to achieve the collective response in real-world contexts. The comprehensive research chain is significant for modeling the collective motion, it facilitates interdisciplinary integration, combining knowledge and techniques from biology, computational modeling, and engineering to develop the high-performance and cost-effective swarm robotics. As a part of the comprehensive research chain, conducting the swarm robotic experiment ensures such bio-inspired mechanisms not only theoretically sound but also applicable in real-world environments. To clarify the important role of real robot experiments in the comprehensive research chain, we have added the following sentences in the revised manuscript:

- see main text: page 2, lines 75-77:

“The robotic experiment results demonstrate the advantage and feasibility of BOC-based interaction in swarm robotics, highlighting the enhancement of BOC on the collective response in real-world applications.”

- see main text: page 6, lines 289-292:

“The robotic experiments validated the feasibility and applicability of BOC-based interaction under realistic and non-ideal conditions and further demonstrated its significant role in the emergence of the scale-free correlation.”

- see main text: page 7, lines 329-333:

“Furthermore, we adopted this bio-inspired mechanism to a swarm consisting of 50 two-wheel differential mobile robots, which effectively achieved the successive collective turns based on the simulated vision sensing. As a part of the comprehensive research chain, conducting the swarm robotic experiment ensures the BOC-based interaction is not only theoretically sound but also applicable in real-world contexts.”

Secondly, in exploring the reason behind advantage of BOC-based interaction for information transfer, we found that the BOC-based interaction emerges the non-trivial scale-free correlation in the simulation experiments. In the previous version of manuscript, we overlooked the validation of whether the non-trivial scale-free correlation also exists in real-world applications of swarm robotics, which is important for understanding this macro-level phenomenon in the more complex and realistic settings.

Hence, in the new version of manuscript, to enhance the link between robotic experiments and the emergence of scale-free correlation, we conducted the robotic experiments with smaller group size of 20, 30 and 50 robots to investigate the emergence of scale-free correlation in swarm robotics. As shown in **Fig.R13g**, we found that for the BOC-based interaction, the correlation length increases with the size of the robotic swarm, indicating the emergence of the non-trivial scale-free correlation in real robot swarms. However, the random-based interaction (baseline model) cannot exhibit the similar macro-level phenomena, as the correlation length remained constant with the increasing flock size. The results of both the simulation and robotic experiments are consistent, further demonstrating the significant role of BOC-based interaction in generating scale-free correlation. We have included the above results in the new version of manuscript (see main text: page 6, lines 284-289):

“Importantly, we also conducted the swarm robotic experiments with smaller group sizes of 20 and 30 robots to analyze the scale-free correlation in real robotic swarms. As shown in Fig. 4g, we found that for the BOC-based interaction, the correlation length increases with the size of the robotic swarm, indicating the emergence of the non-trivial scale-free correlation in swarm robotics. However, the group with random-based interaction cannot exhibit similar macro-level phenomena.”

iii. Additional information provided by the real robots experiments for BOC-based interaction

In conclusion, the additional information offered by the real robot experiments are summarized as follows:

- **Feasibility and applicability of BOC-based interaction under realistic and non-ideal conditions**

Through the swarm robotic experiments, we confirmed that BOC serves as an effective visual cue for real robots to achieve collective response in real-world contexts. Moreover, these robotic experiments demonstrated the feasibility and applicability of the BOC-based interaction in information transfer under non-ideal conditions, e.g., the influence of motion errors and randomness.

- **Emergence of the scale-free correlation in swarm robotics with BOC-based interaction**

The robotic experiments revealed the emergence of scale-free correlation in swarm robotics using BOC-based interaction, indicating that the BOC-based interaction not only endows robots to rapidly propagate information throughout the group, but also reproduces this macro-level phenomenon in swarm robotic systems, which facilitates the understanding of scale-free correlation in real-world context.

- **Complete the research chain from biological data to bio-inspired mechanisms and further to swarm robotics**

In this work, we attempted to establish the comprehensive research chain that extends from biological data to bio-inspired mechanism and further to swarm robotics. As a part of the comprehensive research chain, conducting the swarm robotic experiment ensures the BOC-based interaction is not only theoretically sound but also applicable in real-world contexts.

Point 1.7: I found the manuscript very difficult to read and understand. The problem, solution, and method should be presented more concisely. Important methods and assumptions for the results of the paper should be explicitly explained instead of citing references.

Response: We thank Reviewer #1 for this critical comment and apologize for the lack of clarity in the previous version of our manuscript. In the revised manuscript, we have rewritten the results of both simulation and robotic experiments to improve the readability of the manuscript in the **Results** section (see **main text: pages 4-6, lines 144-292**). Additionally, we have refined the sentences in the **Abstract, Introduction and Methods** to clarify the problem and enhance the clarity of important methods and assumptions. For example,

i. Revisions in Abstract and Introduction

- see main text: page 1, lines 9-15:

“However, previous studies often characterize the visual input as a transient boolean-like sensory stream, which makes it challenging to capture the salient movements of neighbors. This further hinders the onset of collective response for the vision-based mechanisms and poses great demands on the visual sensing device in swarm robotics. An explicit and context-related visual cue is still lacking as the sensory input for decision-making in vision-based mechanisms. Here, we hypothesize that the Body Orientation Change (BOC) might be a significant visual cue characterizing the motion salience of neighbors, facilitating the emergence of the collective response.”

- see main text: page 1, lines 17-18:

“To further explore this empirical finding, we develop the pairwise interaction mechanism based on the BOC.”

- see main text: pages 1-2, lines 42-51:

“However, previous studies often simplified the intricate process of visual perception to the transient binary visual field for decision-making^{20,22,23,25}. On the one hand, such simplified visual information often makes it difficult to extract meaningful visual cues and hinders the ability to capture the motion salience of neighbors. This further poses a gap between vision-based models and the collective response which is essential for many collective tasks such as collective anti-predation⁷ and collective avoidance²⁶. On the other hand, relying on the binary visual field presents practical difficulties for real-world applications in swarm robotics, as it is challenging to achieve with onboard visual sensing devices. There is a pressing need to bridge this gap by uncovering an explicit and context-related visual cue as the sensory input to facilitate the emergence of collective response in vision-based swarm models.”

- see main text: page 2, lines 52-53:

“In this work, we aimed to reveal the visual cue that individuals adopt to select which neighbor to interact with from the first-person perspective to achieve the collective response.”

ii. Clarification of simulation setup of collective spin and collective turn (see main text: pages 8-9, lines 375-412):

“In this work, we conduct the simulation experiments of collective spin and collective turn in the pybullet simulator, where each individual is a simulated two-wheel differential robot in the pybullet environment. The pybullet simulator provides a powerful physics engine that accurately simulates the kinematics of robot motion and physical collisions, which facilitates the application of BOC-based interaction in swarm robotics. Particularly, the occluded individuals are ignored in the neighbor selection (see detailed approaches for identifying the occluded neighbor in Supplementary Sec. 4 and Supplementary Fig. 15).

For the simulation experiments of collective spin, we assumed that each simulated robot has two motion states: stationary and spinning. In the stationary state, both linear and angular speeds of simulated robots are set to 0. In the spinning state, the angular speed is set to the maximum ω_{max}^{sim} (see Supplementary Table 1 for parameters selection), while the linear speed remains 0,

yielding the rotational movement of robots. The simulated robots are required to swiftly transfer the spinning state triggered by a single informed individual (spinning initiator) throughout the entire group. The informed individual starts spinning at a pre-set activation time (the 25th step in our simulation) and stops spinning after completing the 2π rotation, which is selected at the forefront of the group. Each simulated robot determines the BOC of its neighbors within a perception range R_{visual} and selects the neighbor with the maximum BOC to react. The reaction rule is to switch between stationary and spinning states based on the state of the selected neighbor. Specifically, if the selected neighbor is in the spinning state, the focal individual also enters the spinning state and starts to spin with the maximal angular speed; if the selected neighbor is in the stationary state, the focal individual switches to the stationary state and stops spinning. The simulation videos of collective spin can be found in Supplementary videos 3-4. Unless otherwise specified, the parameter selections for the BOC and random-based interaction can be found in Supplementary Table 1. The illustration of the simulation procedure of collective spin can be found in Supplementary Fig.26a.

For the simulation experiment of collective turn, the simulated robots are required to respond to a sudden turn initiated by one informed individual (turn initiator). The informed individual, positioned at the forefront of the group, turns with an angle θ_{info} relative to the group's movement direction at a pre-set moment (the 50th step in our simulation). Each robot calculates the BOC of its neighbors within a perception range R_{visual} and reacts to the neighbor with the maximum BOC. Due to the challenges in acquiring neighbors' velocity from the first-person view¹⁹, we assumed that the focal individual could access the neighbors' velocity. The reaction rule in the simulation experiments of collective turn is the commonly used velocity alignment⁴⁰, i.e., average the headings of the selected neighbor and the robot's own heading (Eq. (7)). The simulation videos of collective turn can be found in Supplementary videos 5-8. Unless otherwise specified, parameter selections in the simulation experiments of collective turn for BOC and random-based interaction can be found in Supplementary Table 2. The illustration of the simulation procedure of collective turn can be found in Supplementary Fig.26b.”

iii. Clarification of important methods

- The clarification of calculation of correlation length (see **main text: page 4, lines 188-193**):

“The point where the correlation function first reaches zero ($C(r_0) = 0$) defines the correlation length. The correlation function $C(r)$ represents the velocity fluctuations as a function of distance r in the group. As the increase of r , $C(r)$ gradually exhibit the decaying transition from strong correlation to weak correlation in velocity among individuals and eventually crossed the x-axis (see Supplementary Sec.3.8 for the detailed definition).

- The clarification of the definition of frontal preference (see **main text: page 10, lines 437-451** and **Fig. 5b-c in main text**):

“The term $\text{fp}(t) = \left(\frac{1 + \hat{\mathbf{v}}_i \cdot \hat{\mathbf{x}}_{ij}}{2}\right)$ quantifies the frontal preference on perceiving neighbor- j 's movement from the first-person perspective of the focal individual i , ranging in $[0, 1]$. The dot product of these two vectors $\hat{\mathbf{v}}_i \cdot \hat{\mathbf{x}}_{ij}$ measures the relative bearing of neighbor- j . For example, for the neighbor- j_1 shown in Fig. 5b, the $\angle(\hat{\mathbf{v}}_i, \hat{\mathbf{x}}_{ij}) = 0$, it signifies that neighbor- j_1 locates directly in front of the focal individual- i , and the focal individual could perceive the neighbor- j_1 since the $(1 + \hat{\mathbf{v}}_i \cdot \hat{\mathbf{x}}_{ij})/2$ is 1. Once the neighbor j moves from the front to back (neighbor-

j_2, j_3, j_4 shown in Fig. 5b), the perception of the focal individual i to the neighbor j diminishes as the $fp(t)$ decreases from 1 to 0. The α is the anisotropic factor that tune the effect of forward-oriented preference on the perception of BOC. When $\alpha = 0$, the focal individual- i ignores the blind area of perception as the $fp(t)$ always equal to 1, which mean that the perception of BOC is not relevant to the relative position of neighbors. In particular, increasing α make Eq.(1) enhance the anisotropic effect of motion perception, which causes individuals to gradually narrow their ability to perceive movements of nearby neighbors toward their frontal vision. For example, as shown in Fig. 5c, if $\alpha = 10$, it means that the perception of BOC $g_{ij}(T, \tau) \approx 0$ when the relative positions of neighbors are out of the visual sight $(-\pi/2, \pi/2)$.”

- The clarification of definition of the normalized out-distance (see **main text: page 11, lines 489-496**):

“The normalized out-distance signifies the hierarchical layer of each node (individual) in the leader-follower network, which is essentially an unweighted directed network. In such a hierarchical network, some individuals act as leaders while others are followers, with directed connections between them, i.e., leaders point to their followers. We leverage the number of connections pointing to other nodes to represent the leadership of the focal individual, which refers to the out-distance (or the out-degree) of nodes in the directed graph. The node (individual) with higher out-degree denotes stronger leadership, as they have more connections pointing to other nodes (followers).”

iv. Clarification of important assumptions

- The assumptions in the empirical data analysis of real fish data (see **main text: pages 2-3, lines 93-96**):

“Notably, the empirical data analysis of this work is based on two key assumptions: first, a leader is present during U-turn events, typically initiating the abrupt maneuver in the group (see Supplementary videos 1-2); Second, leadership within the group is associated with certain visual or motion cues of neighbors, as it has been elucidated in many other biological swarms^{14,30-34}.”

- The assumptions in the simulation experiments (see **main text: page 8, lines 375-379**):

“In this work, we conduct the simulation experiments of collective spin and collective turn in the pybullet simulator, where each individual is a simulated two-wheel differential robot in the pybullet environment. The pybullet simulator provides a powerful physics engine that accurately simulates the kinematics of robot motion and physical collisions, which facilitates the application of BOC-based interaction in swarm robotics.”

- The assumptions in the swarm robotic experiments (see **main text: page 6, lines 255-266**):

“Different from the simulation experiments, on the one hand, we have the robotic swarm respond to successive abrupt turns of informed individuals, posing the higher demand on the responsiveness of the BOC-based interaction. On the other hand, we simulated the vision-based sensing by the pinhole camera model in the swarm robotic validation system and estimated the BOC based on the simulated camera image (see Methods), which provides the opportunity to validate the BOC-based interaction using simulated visual perception from the view of robots.

Due to constraints in the size of the experimental arena, we manipulated the informed individual to initiate a turn every 50 seconds, for a total of two turns throughout robot experiments. For each turn, the informed robot is picked at the front of the group and turns $\frac{\pi}{2}$ with respect to the group's moving direction. Parameters selection in robotic experiments can be found in Supplementary Table 3.”

- The assumptions in the swarm robotic validation system (see **main text: page 12, lines 543-550**):

“The swarm robotic validation system (i.e., SwarmBang system) comprises three primary components: firstly, a server computer; secondly, a motion capture system; and thirdly, a large number of swarm robots. The server computer, equipped with a radio transmitter, is tasked with model computing, simulating local visual perception, and transmitting real-time control commands to the robots. The motion capture system is used to locate the position of each robot. A detailed description of the robot's components and swarm robotic system can be found in Supplementary Fig.16 and Supplementary Fig.19, respectively.”

Point 1.8: It is not clear what precisely BOC means from the perspective of an individual's sensory system and how this relates to the problem statement of the introduction.

Response: We thank Reviewer #1 for this critical comment and apologize for not making this point clear in the previous version of manuscript. From the perspective of an individual's sensory system, changes in the neighbors' body orientation actually lead to variations in their projection of body shape on the focal individual's retina, reflecting real-time adjustments in the neighbors' movement. BOC quantitatively measures the change in these visual projections from the first-person perspective, serving as an explicit and context-related visual cue that characterizes the neighbor's motion salience. To clarify the principle of BOC from the perspective of an individual's sensory system, we have added the following sentences in the **Introduction** section (see **main text: page 2, lines 53-59**):

“Inspired by the theory success of body orientation (or attitude) coordination investigated in macro-level swarm models^{26,27}, we hypothesized that the Body Orientation Change (BOC) might be a significant visual cue that could effectively capture the swift maneuvers made by neighbors in response to the external stimuli, i.e., the motion salience of neighbors. Since the body orientation changes of neighbors directly lead to variations in the projections of their body shape on the focal individual's retina, reflecting the real-time adjustments in the movement of neighbors.”

Additionally, we fully agree with the reviewer's concerns regarding the lack of clarity in the relationship between the BOC and the problem statement in the introduction. In fact, despite significant progress in understanding the emergence of collective motion, it is important to explore this issue from an individual's own perspective with realistic perceptual information. However, previous studies often simplified the intricate process of visual perception to the transient binary visual field for decision-making. For example,

- Bastien R, Romanczuk P. A model of collective behavior based purely on vision. *Science advances*, 2020, 6(6): eaay0792;
- Castro D, Ruffier F, Eloy C. Modeling collective behaviors from optic flow and retinal cues. *Physical Review Research*, 2024, 6(2): 023016;
- Qi J, Bai L, Wei Y, et al. Emergence of adaptation of collective behavior based on visual perception. *IEEE Internet of Things Journal*, 2023.

On the one hand, such simplified visual information is insufficient to extract meaningful visual cues and hinders the ability to capture the motion salience of neighbors. This further poses a gap between vision-based models and the collective response which is essential for many collective tasks such as collective anti-predation and collective avoidance. On the other hand, relying on the binary visual field presents practical difficulties for real-world applications of swarm robotics, as it is challenging to achieve with onboard visual sensing devices. In summary, we detailed on how BOC relates to the problems stated in the introduction in three parts:

i. Capturing the salient movements of neighbors from the first-person perspective

Previous studies often simplify visual perception as the transient binary visual field, making it difficult to extract meaningful visual cues. This further hinders the ability to capture the maneuvering movements of neighbors. However, the BOC is an explicit and context-related visual cue to characterize the motion salience of neighbors. Since changes in the neighbors' body orientation lead to variations in their projection of body shape on the focal individual's retina, reflecting real-time adjustments in the neighbors' movement. BOC quantitatively measures the change of these visual projections from the first-person perspective, which effectively captures the contextual motion information of neighbors, e.g., the swift maneuvers of neighbors.

ii. Facilitating the emergence of collective response within the group

The key process of the collective response is the indirect information propagation governed by combinations of the direct local interaction, which depends on how an individual selects its neighbors according to certain visual or motion cues. BOC provides immediate and explicit feedback on the changes in neighbors' motion adjustments, which is crucial for the emergence of the collective response. By using BOC as a visual cue in neighbor selection, individuals can adjust their actions swiftly and lead to more effective collective response within the group.

iii. Practicability in real-world application of swarm robotics

Relying on the binary visual field for decision-making presents challenges in real-world applications of swarm robotics, as it is demanding to achieve with onboard visual sensing devices. In contrast, BOC is a practical and detectable visual cue that can be captured using standard cameras available on most robotic platforms, reducing the need for specialized sensing devices and sophisticated computer vision technology (see **Supplementary Sec.8** and **Supplementary Fig.21**). The low barrier to real-world implementation makes the BOC-based interaction more feasible and cost-effective, enabling the swarm robotic system to operate with simpler, more affordable sensing devices while maintaining high performance in achieving the collective response.

To clarify the relationship with the problem statement of this work, we have rewritten the problem statement in the **Introduction** section (see **main text page 1-2, lines 34-51**):

“Despite the explanatory success at the emergence of collective motion, the proposed metric, topological, or selective interaction mechanism primarily reproduces the collective motion in a phenomenological manner. These interaction mechanisms often rely on the physical measurements that may not be directly available from the first-person perspective of animals^{8,9,14}.”

Beyond these phenomenological interaction mechanisms, the vision-based modeling paradigm has become popular in current research^{2,17-20}, which offers more biological plausibility by mimicking how animals perceive and react to their neighbors. Plenty of studies have achieved complex collective motion by simply relating the visual inputs and movement decisions²⁰⁻²⁴. However, previous studies often simplified the intricate process of visual perception to the transient binary visual field for decision-making^{20,22,24}. On the one hand, such simplified visual information often makes it difficult to extract meaningful visual cues and hinders the ability to capture the motion salience of neighbors. This further poses a gap between vision-based models and the collective response which is essential for many collective tasks such as collective anti-predation⁷ and collective avoidance²⁵. On the other hand, relying on the binary visual field presents practical difficulties for real-world applications in swarm robotics, as it is challenging to achieve with onboard visual sensing devices. There is a pressing need to bridge this gap by uncovering an explicit and context-related visual cue as the sensory input to facilitate the emergence of collective response in vision-based swarm models.”

Moreover, we have also refined the sentences in the section of **Discussion** to clarify how the BOC relates to the problem statements in the **Introduction** (see **main text pages 7-8, lines 334-352**):

“Recently, several studies have reproduced complex collective motion by relating the boolean visual input with motion decisions, yet challenges persist in achieving the collective response^{20,21,24}. On the one hand, the simplified visual information limits the recognition of salient movements of neighbors and further hinders the emergence of the collective response. In contrast, the BOC provides immediate and explicit feedback on the maneuverable movement of neighbors, reflecting real-time adjustments in the neighbors’ movement. By using BOC as a visual cue to select the interacting neighbors, individuals can adjust their actions swiftly leading to the effective collective response in the group. Additionally, the compelling evidence from simulation and robotic experiments demonstrates that BOC-based interaction not only enables the swarm to follow the informed direction but also emerges the scale-free correlation in the group.

On the other hand, obtaining a binary visual field using onboard visual sensing devices is challenging, which constrains the practical application of such vision-based swarm models. Conversely, it is pertinent to highlight that the BOC also stands out as an effective visual cue for achieving the collective response in swarm robotics. Compared to the binary visual field, BOC is a practical and detectable visual cue that can be estimated using the camera on robots through handy computer vision techniques⁴⁸⁻⁵¹. For example, the robot can effectively estimate the magnitude of the BOC by calculating changes in the bounding box area over consecutive frames obtained from the onboard optical camera (see Supplementary Sec.8 and Supplementary Fig.21 for detailed information).”

Finally, we thank Reviewer #1 again for her/his very insightful and constructive comments. We hope our responses above have addressed those very legitimate issues/concerns in a satisfactory manner.

Response to Reviewer #2

Point 2.0: This work presents a study of a novel mechanism to achieve collective response within the context of collective motion. The mechanism is based on body orientation change (BOC), which is a criteria used by the individual to select which neighbor to interact, namely the neighbor whose body orientation changed the most. The analysis is comprehensive and include analysis of data of real fish, two simulated models, and real robots experiments. The main result of the paper is that scale-free interactions emerge when BOC is used.

The work is significant because it helps in shedding light on the mechanisms behind collective response in animals like fish, and also suggest a mechanism to design analogous mechanisms for robot swarms.

I encourage the eventual publication of this article in Nature Communication. However, before doing so, I believe the article needs significant revisions to provide a more convincing argument in the results and also to make it reproducible by other researchers.

Response: We thank Reviewer #2 very much for reviewing our work and her/his positive assessment of the significant role of BOC in the emergence of scale-free correlation. We next address each of the reviewer's constructive comments/suggestions in order.

Please find below the revisions I recommend:

Point 2.1: The principle behind BOC should be clarified already in the abstract and in the introduction. In general, when modeling collective motion, the building blocks that needs to be designed should answer to the following questions: 1. with which neighbor(s) the focal individual interacts? 2. What information is exchanged between neighbors? 3. How is the information received integrated? 4. How is the integrated information converted to motion? In this paper, BOC answers only to question 1. (and also only one neighbor is selected) This is completely ok, but it should be made very clear from the beginning, and also the explanation of the intuitive understanding behind BOC should be clear

Response: We thank Reviewer #2 for these valuable suggestions and apologize for not making this point clear in the previous version of the manuscript. Firstly, we fully agreed the reviewer's insightful comments noted "In general, when modeling collective motion, the building blocks that needs to be designed should answer to the following questions: 1. with which neighbor(s) the focal individual interacts? 2. What information is exchanged between neighbors? 3. How is the information received integrated? 4. How is the integrated information converted to motion?". We acknowledged that the BOC only answers to the question 1 in these building blocks, i.e., with which neighbor the focal individual interacts during the U-turn events. It is because we believed that neighbor selection is fundamental to the emergence of collective motion and has great impact on the interaction topology of the group, which further affects the information transfer in the group and the shaping of the collective response. Hence, we started with the exploration of how the focal individual selects its neighbor for the collective response. To clarify the specific question BOC addresses and the assumption of pairwise interaction from the beginning, we have added the following sentences in the **Abstract** and **Introduction**:

- see **main text: page 1, lines 17-18:**

“To further explore this empirical finding, we develop the pairwise interaction mechanism based on the BOC.”

- see main text: page 2, lines 52-53:

“In this work, we aimed to reveal the visual cue that individuals adopt to select which neighbor to interact with from the first-person perspective to achieve the collective response.”

- see main text: page 4, lines 145-147:

“The empirical findings of BOC inspire us to develop the pairwise interaction mechanism, wherein the focal individual selects the neighbor with the largest magnitude of BOC to interact with.”

Secondly, for the principle behind the body orientation change (BOC), the BOC is one of the explicit and context-related visual cues that characterize the neighbor’s motion salience. Since changes in the neighbors’ body orientation lead to variations in their projection of body shape on the focal individual’s retina, reflecting real-time adjustments in the neighbors’ movement. BOC quantitatively measures the change of these visual projections from the first-person perspective, which effectively captures the contextual motion information of neighbors, e.g., the swift maneuvers of neighbors. In the new version of manuscript, we have refined the following sentences in the **Abstract** and **Introduction** to better clarify the principle and the intuitive understanding behind BOC:

- see main text: page 1, lines 11-15:

“An explicit and context-related visual cue is still lacking as the sensory input for decision-making in vision-based mechanisms. Here, we hypothesize that the Body Orientation Change (BOC) might be a significant visual cue characterizing the motion salience of neighbors, facilitating the emergence of the collective response.”

- see main text: page 2, lines 55-59:

“Inspired by the theory success of body orientation (or attitude) coordination investigated in macro-level swarm models^{26,27}, we hypothesized that the Body Orientation Change (BOC) might be a significant visual cue that could effectively capture the swift maneuvers made by neighbors in response to the external stimuli, i.e., the motion salience of neighbors. Since the body orientation changes of neighbors directly lead to variations in the projections of their body shape on the focal individual’s retina, reflecting the real-time adjustments in the movement of neighbors.”

Point 2.2: Finding another mechanism yielding to scale-free interaction is a significant contribution. However, the work currently suffers from the critical lack of a baseline implemented in simulation and with real robots. More precisely, we cannot be sure that BOC is the ultimate responsible ingredient yielding to scale-free correlations. Therefore, I suggest the authors to implement a baseline/control experiments in which everything is kept the same, but the single interaction neighbor is selected with another criteria, e.g. a random neighbor within a perception radius, or the closest neighbor. Would this mechanism produce scale-free interactions? If the answer is yes, then BOC is not what yields to this result, but there must be something else. Note that the swarm robotics experiment seem to contain partially this analysis, however it should be performed also or mainly for in pybullet, and results should belong in the main text and not in the supplementary materials.

Response: We thank Reviewer #2 for these very constructive comments and excellent suggestions. Following the suggestion of Reviewer #2, we have incorporated two baseline

models to demonstrate the significant role of BOC in the emergence of scale-free correlation, respectively. The two baseline models are introduced as follows:

- **Random-based interaction:** The random-based interaction means that an individual randomly selects one neighbor to react within a perception radius R_{visual} . The random neighbor selection process obeys the Uniform distribution (i.e., the id of the selected neighbor $\sim Uniform(1, N)$).
- **Nearest neighbor-based interaction:** The nearest neighbor-based interaction means that the individual only reacts to the neighbor with the nearest relative distance.

In particular, to ensure the fairness of the comparative experiments, we set the same parameters of these two baseline models with the BOC-based interaction (see **Supplementary Tables 1-3**). Through extensive simulations and real robot experiments, we found that only BOC-based interaction emerges the scale-free correlation in the group, while the other two interaction mechanisms, i.e., random-based and nearest neighbor-based interaction cannot reproduce this macro-level phenomenon. The above results confirmed that **BOC is a primary factor responsible for producing the non-trivial scale-free correlation**. The detailed simulation and real robot experimental results of collective spin and collective turn are shown in the following:

i. Comparing the emergence of scale-free correlation with Random-based and Nearest neighbor-based interaction in the simulation experiments of collective spin

To begin with, we compare the simulation results between the BOC and random-based interaction. By analyzing snapshots in the collective spin, we could tell the difference in the ability of information transfer between these two models. Specifically, it is evident that BOC-based interaction facilitates the rapid transfer of the spin initiator's state, leading to collective spinning across the entire group (**Fig.R3a** and **Fig.R3b**). In contrast, only a few of individuals near the spin initiator entered into the spinning state in the group with random-based interaction, leading to the failure of the collective spinning. Even after the spin initiator completes the rotation of 2π , a significant portion of individuals remain stationary, highlighting the shortcoming of random-based interaction in information transfer (**Fig.R3e** and **Fig.R3f**). We have also provided the simulation videos of BOC-based interaction and random-based interaction in the collective spin, please see **Supplementary videos 3-4**. For the emergence of the scale-free correlation, in the case of the BOC-based interaction, we found the linear increase in correlation length with the grow of flock size (**Fig.R3c**), implying the emergence of scale-free correlation. However, in the case of the random-based interaction, we found that correlation length did not rise linearly with the increasing flock size, indicating the absence of scale-free correlation within the group (**Fig.R3g**). Additionally, we found that the information transfer speed V_s in the random-based interaction remained consistently slower than that in BOC-based interaction, also without continually increasing with group size (**Fig.R3d** and **Fig.R3h**).

Next, we compare the simulation results between the BOC-based interaction with the nearest neighbor-based interaction. Similar to the random-based interaction, the group with the nearest neighbor-based interaction fails to achieve the collective spin. **Fig.R4e-f** illustrates a worst-case where only one individual responds to the spin initiator (colored by red) in the nearest neighbor-based interaction. This is because these two individuals are the nearest neighbor to each other, impeding the spread of information within the group. Moreover, the nearest neighbor-based interaction also fails to exhibit scale-free

correlation (**Fig.R4g**). Additionally, the information transfer speed V_s also remains constant with increasing group size (**Fig.R4h**).

To sum up, in the simulation experiments of collective spin, the BOC-based interaction facilitates rapid information propagation in the group, while also exhibiting the scale-free correlation. However, the random-based interaction and nearest neighbor-based interaction not only fail to demonstrate scale-free correlation but also exhibit poor ability of information transfer within the group.

ii. Comparing the emergence of scale-free correlation with Random-based and Nearest neighbor-based interaction in the simulation experiments of collective turn

From the view of response accuracy and responsiveness, we found that both random-based interaction and nearest neighbor-based interaction failed to achieve the collective turn, exhibiting very low response accuracy and responsiveness, especially in larger group size (**Fig.R5f-h** and **Fig.R6f-h**). It is worth noting that for random-based interaction, the group can only achieve the collective turn when the group size is small ($N = 10$), but both response accuracy and responsiveness keep decreasing with the increase of group size. In contrast, BOC-based interaction not only rapidly responds to sudden turns but also demonstrates high response accuracy, regardless of the group size (**Fig.R5a-c** or **Fig.R6a-c**).

Moreover, for the emergence of scale-free correlation, as shown in **Fig.R5i** and **Fig.R6i**, we found that the correlation length for both random-based and nearest neighbor-based interactions did not exhibit a trend of linear growth with the increase of flock size. In addition, the information transfer speed V_s also remained nearly constant with the increase of group size in both random-based and nearest neighbor-based interactions (**Fig.R5j** and **Fig.R6j**). However, it is worth noting that BOC-based interaction exhibits a linear growing trend of correlation length with the increasing flock size, indicating the emergence of the scale-free correlation. The information transfer speed V_s of BOC-based interaction also keeps growing with the increase of group size (**Fig.R5d-e** or **Fig.R6d-e**).

In fact, the nearest neighbor-based interaction restricts an individual's perceptual radius to the relative distance of its nearest neighbor. This further introduces an inherent bias in the actual range of neighbor selection when compared to the BOC-based and random-based interactions. Hence, we decided to use the **random-based interaction as the baseline model** for comparison with the BOC-based model in the new version of manuscript.

iii. Comparing the emergence of scale-free with the Random-based interaction in the swarm robotic experiments

Furthermore, we conducted the robotic experiments of BOC-based interaction and random-based interaction with the group size of 20, 30 and 50 to investigate the emergence of scale-free correlation in swarm robotics. The detailed robotic experimental results are shown as follows:

As shown in **Fig.R13g**, we found that for the BOC-based interaction, the correlation length increases with the size of the robotic swarm, indicating the emergence of the non-trivial scale-free correlation in real robotic swarms. However, the random-based

interaction cannot exhibit the similar macro-level phenomena, as the correlation length remained nearly constant with the increasing flock size. The results of both the simulation and robotic experiments are consistent, further demonstrating the significant role of BOC-based interaction in generating scale-free correlation. More robotic experiments results can be found in **main text page 6, lines 249-292** or the **Fig.R13**.

In the revised manuscript, we have rewritten the **Results** of simulation and robotic experiments to provide a clearer understanding of the role of BOC in producing the scale-free correlation (see **main text: pages 4-6, lines 145-292**).

Point 2.3: The article should review more related work, e.g. other papers in which mechanisms yielding to scale-free correlations was observed (other than criticality). For example this paper: <https://link.springer.com/article/10.1007/s10955-014-1114-8>.

Response: We thank Reviewer #2 for the valuable suggestion and for bringing this paper (<https://link.springer.com/article/10.1007/s10955-014-1114-8>) to our attention, which provides us with a comprehensive understanding of the emergence of scale-free correlation. We have included this important reference in the revised manuscript (see **main text: page 8, lines 361-365**):

“Apart from our work, researchers have also observed the scale-free correlation in the self-organized system with a non-critical regime. For example, Huepe *et al.* achieved the emergence of the scale-free correlation in a self-propelled model with position-based interaction, which provides an alternative mechanism for generating this macro-level phenomenon without the presence of critical dynamics⁵⁵.”

Point 2.4: The claim saying that mechanisms other than those based on field of view are "omniscient" is not entirely correct. For example, metric-based mechanism are not omniscient if a 360 degrees vision exist, even if not found in animals it can be implemented on robots with a purely local (non omniscient) mechanism. Biologically-plausible is not the same as omniscient, so a distinction should be drawn when reviewing other mechanisms

Response: We thank Reviewer #2 for providing these insightful comments and critical suggestions. We fully agreed the reviewer noted, “Biologically-plausible is not the same as omniscient, so a distinction should be drawn when reviewing other mechanisms”. We acknowledged that it is not appropriate to distinguish the vision-based swarm models from previous studies based on whether they have omniscient perception. In fact, our goal is to highlight the biological plausibility of the vision-based swarm models, as these models aim to mimic how animals perceive and react to their neighbors. Previous swarm models often reproduce the collective motion in a phenomenological manner and rely on the physical measurements that may not be directly available from the first-person perspective of individuals. Instead, vision-based models not only move beyond phenomenologically reproducing the similar trajectory of collective motion, but also provide the deeper understanding and exploration on the emergence of collective motion from the first-person perspective. In the new version of manuscript, we have removed the inappropriate expression and rewritten sentences in the **Introduction** when reviewing other mechanisms (see **main text: page 1, lines 34-40**):

“Despite the explanatory success at the emergence of collective motion, the proposed metric, topological, or selective interaction mechanism primarily reproduces the collective motion in a phenomenological manner. These interaction mechanisms often rely on the physical

measurements that may not be directly available from the first-person perspective of animals^{8,9,14}. Beyond these phenomenological interaction mechanisms, the vision-based modeling paradigm has become popular in current research^{2, 17-20}, which offers more biological plausibility by mimicking how animals perceive and react to their neighbors.”

Point 2.5: There are lack of details concerning the real robot implementation, which hinders reproducibility. Importantly, it is not clear what computation and sensing is done onboard of the robots, and what is done instead by a central computer after data collection. If most of the computation and sensing is done externally, this reinforces further my previous comment: authors cannot claim the mechanism considered is not omniscient if their implementation on real robot (the main opportunity to prove locality) is itself omniscient.

Response: We thank Reviewer #2 for the insightful comments and appreciate the reviewer for pointing out the issue of insufficient description of the robotic experiments. We fully agreed the reviewer noted “If most of the computation and sensing is done externally, this reinforces further my previous comment: authors cannot claim the mechanism considered is not omniscient if their implementation on real robot (the main opportunity to prove locality) is itself omniscient.”. In the revised manuscript, we have removed the incorrect expressions and refined the relevant sentences (see **main text: page 1, lines 34-40**).

Indeed, we acknowledged that the swarm robotic validation system is in the pseudo-distributed manner that most of the computation and sensing is achieved by the server computer (see the clarification of workflow of swarm robotic system in below or the **Fig.R10**). The reason lies in the low wireless communication bandwidth of our robots (31 bytes, 248 bits, see **Supplementary Sec.6.4** or **Fig.R12**), transferring the global information such as the positions and headings of the large-scale robotic swarm is quite challenging. The transmission of the global information incurs significant time delay in the control of swarm robots, which further hinders the validation of swarm model in the large-scale robotic swarm. For example, considering that the robot's x position, y position, and heading are float data types, transmitting the pose information of one robot requires 12 bytes of bandwidth. Scaling this up to 50 robots would require 600 bytes, necessitating approximately 20 times transmissions to broadcast all the information based on our wireless communication module. Given the transmission distance and potential electromagnetic signal interference, completing this transmission would take at least 4 seconds (the maximum transmission interval without packet loss is 0.2 seconds in the real test). Hence, we are compelled to rely on the external computation source for the majority of sensing tasks, local perception simulation, and model computation.

Additionally, it is noteworthy that although the collection of motion states of all robots and swarm model runs on the external server computer, the outcome derived from the BOC-based interaction remain computationally consistent, regardless of whether the local interaction mechanism is computed on the robots or on a server computer. It is because both approaches rely on the global information such as the positions and headings centrally perceived from the server computer to simulate the local perception of the robots. Each robot independently makes its motion decisions based on the same simulated local information. Notably, to reinforce the validation of BOC-based interaction with local visual information in the revised robotic experiments, we have provided the more realistic local perception information for robots by simulating vision-based sensing through the pinhole camera model (see **Supplementary Sec.6.2** or **Fig.R8**) and redone the swarm robotic experiments (see **main text: page 6, lines: 249-292**).

Furthermore, to improve the reproducibility of swarm robotic experiments and clarify the computational distinctions between the server computer and robots, we provided additional details of the swarm robotic validation system in the following three parts:

i. Workflow of the swarm robotic validation system

As shown in **Fig.R9**, we illustrated the architecture of swarm robotic validation system. The swarm robotic validation system (i.e., SwarmBang system) comprises three primary components: firstly, a server (central) computer; secondly, a motion capture system; and thirdly, a large number of swarm robots (while the system has the capacity to support up to 100 robots, we limited our robotic experiments to 50 robots due to spatial constraints). The server computer, equipped with a radio transmitter, is tasked with simulating local visual perception, model computing, and transmitting real-time control commands to the robots. The motion capture system is used to locate the position of each robot. To clarify the differences in computational content between the robots and the server computer, we illustrate the workflow of the swarm robotic validation system in **Fig.R10**.

As shown in **Fig.R10**, the workflow of our validation system begins with the measurement of each SwarmBang robot's position and heading using the NOKOV motion capture system processed on the server computer (**step 1**). Next, in the revised manuscript, we involved the the pinhole camera model to simulate the vision-based sensing of robots in server computer which can improve the realism of perceptual inputs of robots and also suggested by Reviewer #3 (please see the **Point 3.4**) (**step 2**). Based on the pinhole camera model, we can obtain projections of the neighboring robots onto the simulated image plane (see **Fig.R8a**), allowing us to simulate the visual perception of robots and approximate the body orientation change (BOC) using the generated bounding boxes based on the simulated images (see **Supplementary Sec.6.2** or **Fig.R8b-e**). Then, the server computer determines the desired heading θ_d for each robot based on the corresponding swarm model and the simulated local information (**step 3**). After obtaining the desired heading of each robot, the desired angular speed ω is calculated as the $\omega = \min(\frac{\Delta\theta}{\Delta t}, \omega_{max}^{robot})$, where the $\Delta\theta = \theta_d(t + 1) - \theta(t)$ and ω_{max}^{robot} is the maximum angular speed in robotic experiments (see **Supplementary Table 3** for parameters selection in real robot experiments). Following that, the server computer calculates the desired linear and angular speed of each robots and broadcast to all robots based on a wireless communication module through the customized communication protocol with a fixed time interval Δt (details of this protocol is shown below or the **Fig.R12**) (**step 4**). Upon receiving these desired motion commands, each robot then calculates the speed of its left and right wheels based on the kinematic model of the differential-drive platform in the robots (details of the motion control of robots is shown below) (**step 5**).

In the new version of manuscript, we have included the workflow of the robotic validation system in the **Supplementary Sec.6.1** and added the following sentences in the section of **Methods** to clarify the computational distinctions between the server computer and robots (see **main text: page 12, lines 543-566**):

“The swarm robotic validation system (i.e., SwarmBang system) comprises three primary components: firstly, a server computer; secondly, a motion capture system; and thirdly, a large number of swarm robots. The server computer, equipped with a radio transmitter, is tasked with model computing, simulating local visual perception, and transmitting real-time control commands to the robots. The motion capture system is used to locate the position of each robot. A detailed

description of the robot's components and swarm robotic system can be found in Supplementary Fig.16 and Supplementary Fig.19, respectively.

The workflow of our validation system begins with the measurement of each SwarmBang robot's position and heading using the NOKOV motion capture system processed on the server computer. Next, the server computer then simulates the local vision-based sensing process of each robot through the pinhole camera model. Based on the pinhole camera model, we can obtain projections of the 3D neighboring robots onto a 2D image plane, allowing us to simulate the visual perception of the robots and approximate the body orientation change (BOC) using the generated bounding boxes based on the simulated images. Then, the server computer determines the desired heading θ_d for each robot based on the corresponding swarm model and the simulated local information. After obtaining the desired heading of each robot, the desired angular speed ω is calculated as the $\omega = \min\left(\frac{\Delta\theta}{\Delta t}, \omega_{max}^{robot}\right)$, where the $\Delta\theta = \theta_d(t+1) - \theta(t)$ and ω_{max}^{robot} is the maximum angular speed in robotic experiments (see Supplementary Table 3 for parameters selection in real robot experiments). Following that, the server computer calculates the desired linear and angular velocity of each robot and broadcasts to all robots based on a wireless communication module through the customized communication protocol with a fixed time interval $\Delta t = 0.5s$ (details of this protocol can be found in Supplementary Sec.6.4 and Supplementary Fig.20). Upon receiving these desired velocities, each robot then calculates the speed of its left and right wheels based on the kinematic model of the differential-drive platform in the robots (The motion control of the SwarmBang robot can be found in Supplementary Sec.6.3).”

ii. Hardware design and motion control of the SwarmBang robot.

Moreover, we provided the detailed schematic of the robot's hardware design (**Fig.R11**). The hardware design of the SwarmBang robot is separated into two parts: 1) the PCB board for decision-making and communications and 2) the PCB board for motion control and battery management. Two 3.7V rechargeable batteries (2*800mAh) provide energy for about 1 hour in our experimental settings. Each robot is equipped with a wireless communication module (NRF24L01) to receive commands from the server computer. According to the wheeled robot's kinematic model, after receiving the motion command of the desired linear and angular speed, the velocities of the left and right wheels are calculated as follows:

$$\begin{cases} v_l = v + \frac{\omega}{2} \cdot L \\ v_r = v - \frac{\omega}{2} \cdot L \end{cases}$$

where the v and ω are the linear speed and angular speed, respectively. L is the length between the two wheels.

To control the velocity of the robots' wheels, we use two four-phase, five-wire step motors to drive the SwarmBang robots. The inherent step angle of the motor shaft is 5.625 degrees, with a subdivision ratio of 64. It takes 4096 pulse signals for the motor shaft to rotate the robot's wheel one full revolution. The rotational speed of the step motor is adjusted by changing the frequency of the pulse signals. The maximum pulse frequency that the step motor can respond to is 1kHz in our robots, resulting in a maximum motor speed of approximately 14.65 RPM ($v_{rpm}^{max} = \frac{f \times 60}{\frac{360}{5.625} \times 64} = \frac{1000 \times 60}{4096} \approx 14.6RPM \approx 0.24n/sec$). The wheels of

the SwarmBang robots have a radius of $r = 17.5 mm$, and the distance between the left and right wheels is $L = 60 mm$. Therefore, the theoretical maximum linear speed of the real robot

is approximately $v \approx 25 \text{ mm/s}$, and the theoretical maximum angular velocity is approximately $\omega_{max} \approx 0.83 \text{ rad/s}$. We have added the above details about the SwarmBang robot in **Supplementary Sec.6.3** (see **SI: page 9, lines 302-322**).

iii. Division of transmission frequency and customized communication protocol.

Transmitting motion commands for 50 robots with low communication bandwidth and tight latency requirements presents significant challenges for the robotic validation system. Hence, we developed a low-redundancy communication protocol and employed multiple communication frequencies to simultaneously transmit motion commands, with each frequency corresponding to 25 robots. As shown in **Fig.R12a**, each robot's desired linear and angular speeds are represented using 9 bits: 1 bit for indicating the turning direction (0 for left, 1 for right), 3 bits for the desired linear speed, and 5 bits for the angular speed. Each data frame contains the desired linear and angular speeds for 25 robots, totaling $9 \times 25 = 225$ bits. Additionally, we have implemented a Cyclic Redundancy Check (CRC) in each robot to verify the correctness of the received data. Upon receiving a data frame, each robot locates its desired linear and angular speeds within the data frame based on its unique robot ID. Furthermore, we employed frequency division multiplexing (2.42GHz and 2.46GHz) to separately control two groups of robots based on the customized communication protocol, each consisting of 25 robots (**Fig.R12b**), totaling 50 robots. The approach of frequency division implicitly enhances communication bandwidth and enables the control of large-scale swarm robotics with low latency for validating the feasibility of the BOC-based interaction in real-world contexts. We have added the above details about the swarm robotic validation system in **Supplementary Sec.6.4** (see **SI: page 10, lines 323-338**) and **Supplementary Fig.20**.

Point 2.6: It should also be discussed exactly how BOC can be implemented on robots using purely local sensing and computation (for example, can vision and bounding boxes be used to calculate BOC over consecutive frames?). The latter is important because it may suggest what is the perceptual mechanism used by animals to do the same

Response: We thank Reviewer #2 for the insightful comments and fully agree the reviewer noted, "The latter is important because it may suggest what is the perceptual mechanism used by animals to do the same". Regrettably, due to our limited expertise in biology and neurobiology of animals, we were unable to establish a clear connection between the intricate biological visual nervous system and the detection of BOC through the empirical data analysis. Nevertheless, we highly valued the significant suggestion provided by the reviewer on calculating BOC based on vision and consecutive frames of bounding boxes. To validate its feasibility, we intend to estimate the BOC with consecutive bounding boxes based on the simulated RGB camera images captured in the pybullet simulator. The experiment results are shown in the following:

■ Estimation of the BOC using consecutive bounding boxes based on images captured by the simulated RGB camera in pybullet simulator

First, the pybullet simulator has both OpenGL GPU visualizer and the built-in CPU renderer, which makes it easy for us to obtain the simulated RGB camera images from the view of robots. By recognizing the color and shape of robots, we could obtain the bounding box of each robot from the simulated RGB images in the pybullet. The bounding box ($\mathbf{x}, \mathbf{y}, \mathbf{w}, \mathbf{h}$) is defined as a rectangular characterized by four parameters: the \mathbf{x} and \mathbf{y}

coordinates of the top-left corner in the image, the box's width (w), and the box's height (h), which represents the position and size of an object within an image.

Second, we conducted the simulation experiments of collective turn with a group size of 10 to get the consecutive simulated RGB images from a certain robot's view (**Fig.R7a**), where the informed individual is the robot-0 and initiates the abrupt turning at the 50s. As shown in **Fig.R7b**, we demonstrated the simulated camera view of robot-2 from the $T = 51s$ to $T = 54s$ and marked the bounding boxes (red rectangular shown in **Fig.R7b**) of four neighboring robots detected from the simulated RGB images. Third, to quantify the approximated BOC based on these bounding boxes, it is intuitive to estimate the BOC by measuring the consecutive changes in the bounding box area. On the one hand, the bounding box area is closely related to the body orientation change of individuals. On the other hand, estimating BOC through bounding box areas is computationally simple and efficient for the robots.

Finally, as shown in **Fig.R7c**, we estimated the BOC based on the area of the bounding boxes over consecutive simulated RGB images. From **Fig.R7c**, we found that before the informed individual made the abrupt turn, the estimated BOC remained low and constant for all the perceived robots (robots 0, 8, 6, and 9). After the informed individual suddenly turned, the BOC of robot-0 (the informed individual) was the first to increase, followed by a rising trend in the BOC of the other three robots. These results suggested that consecutive changes in the bounding box area can be used to quantitatively reflect the trend of BOC.

In summary, it is practical to estimate the BOC using consecutive changes in bounding box area, which provided the promising route to implement the BOC on the real-world application of swarm robotics. Notably, the rapid advancement in computer vision has led to a mature and widely applied methodology for obtaining bounding boxes of specific objects from images captured by the cameras. Even researchers outside the field of computer vision can easily achieve object detection and tracking. Below are some SOTA (State-of-the-Art) methods that facilitate the straightforward implementation of bounding box detection:

- Wang C Y, Bochkovskiy A, Liao H Y M. YOLOv7: Trainable bag-of-freebies sets new state-of-the-art for real-time object detectors, Proceedings of the IEEE/CVF conference on computer vision and pattern recognition. 2023: 7464-7475.
- Wang A, Sun Y, Kortylewski A, et al. Robust object detection under occlusion with context-aware compositionalnets, Proceedings of the IEEE/CVF conference on computer vision and pattern recognition. 2020: 12645-12654.
- Redmon J, Divvala S, Girshick R, et al. You only look once: Unified, real-time object detection, Proceedings of the IEEE conference on computer vision and pattern recognition. 2016: 779-788.
- Hu P, Wang W, Zhang C, et al. Detecting salient objects via color and texture compactness hypotheses, IEEE Transactions on Image Processing, 2016, 25(10): 4653-4664

In the new version of manuscript, we have included the discussion on how exactly BOC can be implemented on robots using purely local sensing and computation in the **Discussion Section** (see **main text: page 8, lines 347-352**):

“Compared to the binary visual field, BOC is a practical and detectable visual cue that can be estimated using the camera on robots through handy computer vision techniques⁴⁸⁻⁵¹. For example, the robot can effectively estimate the magnitude of the BOC by calculating

changes in the bounding box area over consecutive frames obtained from the onboard optical camera (see Supplementary Sec.8 and Supplementary Fig.21 for detailed information).”

The detailed results of estimation of the BOC using bounding boxes over consecutive frames have been included in **Supplementary Sec. 8** and **Supplementary Fig.21**.

Point 2.7: The definition of alpha is not entirely clear. Does it only change the "frontal preference" or is it also used to tune the wideness of the field of view? If the field of view is fixed, how can alpha model the full 360 degrees spectrum of frontal preference? (neighbors in the back would be anyway ignored). Please explain.

Response: We thank Reviewer #2 for this critical comment and apologize for not making this point clearly in the previous version of manuscript. Indeed, the alpha (α) only changes the frontal preference from the perspective of the focal individual. To emulate the fact of frontal preference in biological perception, we involve the $fp(t) = \left(\frac{1+\hat{v}_i \cdot \hat{x}_{ij}}{2}\right)^\alpha$ in the main text Eq.(1) to characterize the anisotropic visual perception of individuals, where the α is the anisotropic factor that tunes the degree of frontal preference on the neighbor- j . We explain the meaning of $fp(t) = \left(\frac{1+\hat{v}_i \cdot \hat{x}_{ij}}{2}\right)^\alpha$ and the effect of anisotropic factors α in the following:

As shown in **Fig.R15a**, the $\left(\frac{1+\hat{v}_i \cdot \hat{x}_{ij}}{2}\right)$ quantifies the frontal preference on perceiving neighbor- j 's movement from the first-person perspective of the focal individual i , ranging in $[0, 1]$. The dot product of these two vectors $\hat{v}_i \cdot \hat{x}_{ij}$ measures the relative bearing of neighbor- j . It simulates that as the first-person view of neighbor- j moves from the front to the back, the visual perception of focal individual i to the neighbor- j diminishes. For example, as shown in **Fig.R15a**, if $\angle(\hat{v}_i, \hat{x}_{ij}) = 0$, i.e., the dot product of $\hat{v}_i \cdot \hat{x}_{ij} = 1$, it signifies that neighbor- j_1 locates directly in front of the focal individual- i , and the focal individual could perceive the neighbor- j_1 since the $(1 + \hat{v}_i \cdot \hat{x}_{ij})/2$ is 1 (neighbor- j_1 shown in **Fig.R15a**). Once the neighbor- j gradually moves from the front to back respect to the focal individual i , i.e., $\angle(\hat{v}_i, \hat{x}_{ij})$ goes from 0 to π or $-\pi$, the visual perception to the neighbor- j could diminish since $(1 + \hat{v}_i \cdot \hat{x}_{ij})/2$ decreases from 1 to 0 (neighbor j_2, j_3, j_4 in **Fig.R15a**). By incorporating the $\left(\frac{1+\hat{v}_i \cdot \hat{x}_{ij}}{2}\right)$ with the perception of BOC, we could emulate the anisotropic perception of neighbor's body orientation change (Eq. (1) in the main text).

To tune the effect of frontal preference of individual, we involve the anisotropic factor α in the $fp(t)$, i.e., $fp(t) = \left(\frac{1+\hat{v}_i \cdot \hat{x}_{ij}}{2}\right)^\alpha$. When $\alpha = 0$, the focal individual i ignores the forward-oriented perception as the $fp(t)$ always equal to 1 (the green line shown in **Fig.R15b**), which means that the perception of BOC is not relevant to the relative positions of neighbors. In particular, increasing α ($\alpha \geq 1$) make Eq. (1) enhance the anisotropic effect of the visual perception, which causes individuals to gradually narrow the perception of nearby neighbors toward their frontal vision. For example, as shown in **Fig.R15b**, if $\alpha = 10$, it means that the perception of BOC $g_{ij}(T, \tau) = 0$ when the relative positions of neighbors are out of the visual sight $(-\pi/2, \pi/2)$.

In the revised manuscript, we have added the detailed explanations and geometric definitions to better clarify the effects of α on the frontal preference (see the **main text: page 10, lines 437-451** and **Fig.5b-c**):

“The $\left(\frac{1 + \hat{\mathbf{v}}_i \cdot \hat{\mathbf{x}}_{ij}}{2}\right)$ quantifies the frontal preference on perceiving neighbor- j 's movement from the first-person perspective of the focal individual i , ranging in $[0, 1]$. The dot product of $\hat{\mathbf{v}}_i \cdot \hat{\mathbf{x}}_{ij}$ measures the relative bearing of neighbor j . For example, for the neighbor- j_1 shown in Fig.5b, the $\angle(\hat{\mathbf{v}}_i, \hat{\mathbf{x}}_{ij}) = 0$, it signifies that neighbor- j_1 locates directly in front of the focal individual i , and the focal individual could perceive the neighbor- j_1 since $(1 + \hat{\mathbf{v}}_i \cdot \hat{\mathbf{x}}_{ij})/2$ is 1. Once the neighbor j moves from the front to back (neighbor- j_2, j_3, j_4 shown in Fig.5b), the perception of the focal individual i to the neighbor j diminishes as the $\text{fp}(t)$ decreases from 1 to 0. The α is the anisotropic factor that tunes the effect of forward-oriented preference on the perception of BOC. When $\alpha = 0$, the focal individual i ignores forward-oriented perception as the $\text{fp}(t)$ is always equal to 1, which means that the perception of BOC is not relevant to the relative position of neighbors. In particular, increasing α ($\alpha \geq 1$) makes Eq. (1) enhance the anisotropic effect of visual perception, which causes individuals to gradually narrow the perception of nearby neighbors toward their frontal vision. For example, as shown in Fig.5c, if $\alpha = 10$, it means that the perception of BOC $g_{ij}(T, \tau) = 0$ when the relative positions of neighbors are out of the visual sight $(-\pi/2, \pi/2)$.”

Point 2.8: I recommend to re-order the arrangement or labeling of the panels in Fig 2 in a more logical way

Response: We thank Reviewer #2 for this constructive suggestion. We have revised the arrangement of Fig.2 (see **main text: page 19, Fig.2**) to better demonstrate the results in simulation experiments of collective spin. In **Fig.2**, we start with the comparison of snapshots between the BOC and random-based interaction. Specifically, in **Fig.2a**, we demonstrate the snapshot of the collective spin of $N = 200$ when the spinning initiator had just spun 2π for the BOC-based interaction. The headings and group polarization of BOC-based interaction as a function of time are shown in **Fig.2b-c**, respectively. **Fig.2d** shows the correlation function varied as the increasing distance r with different group sizes for BOC-based interaction. **Fig.2e** shows the snapshot of the collective spin of $N = 200$ when the spinning initiator had just spun 2π for the random-based interaction. Notably, the color rendering of individuals represents the length of the spinning lag, and the individuals who consistently remained stationary are colored with grey. The headings and group polarization of the random-based interaction as a function of time are shown in **Fig.2f-g**, respectively. **Fig.2h** demonstrates the correlation function varied as the increasing distance r with different group sizes for random-based interaction. In **Fig.2i-j**, we compare the BOC-based interaction and random-based interaction from the perspective of the information transfer speed and the emergence of scale-free correlation. **Fig.2i** shows the relationship between the information transfer speed and group size for BOC and random-based interaction, respectively. **Fig.2j** shows the relationship between the correlation length and flock size for BOC and random-based interaction, respectively. Particularly, as the analysis between basic properties of information transfer (e.g., information transfer direction, group polarization, max spinning lag) and group size in previous version of **Fig.2** is not directly relevant to the key finding of this work, we have moved the relevant analysis results to **Supplementary Sec.11** and **Supplementary Fig.30**.

Point 2.9: It would be interesting to know if the same results would hold in case the initiators (e.g. in the collective motion with turn simulation) are not positioned at the border

Response: We thank Reviewer #2 for the constructive comments and critical suggestions. Indeed, we overlooked the investigation about the impact of initiator's position on the simulation results of collective spin and collective turn in the previous version of manuscript. Following reviewer's excellent suggestions, we analyzed the impact of different position on the simulation experiments of collective spin and collective turn. Through extensive simulations, we found that the outcomes of both the collective spin and collective turn experiments remain consistent regardless of whether the initiator is positioned around at the center, middle, or border of the group. The detailed simulation results are shown as follows:

Firstly, we define the spatial center proximity index γ to characterize the relative position of the initiator respect to the flock center. As shown in **Fig.R16a**, the γ is calculated as the $\gamma = \frac{d_i^c}{F_{\text{radius}}}$, which is the ratio of the relative distance between the initiator and the flock center to the radius of the flock. d_i^c is the relative distance between the initiator and the flock center. F_{radius} is the radius of the flock, estimated by the half of the maximal relative distance among individuals. Specifically, when the $\gamma \leq 0.45$, the initiator is positioned around the flock center (**Fig.R16b**). When the $0.45 < \gamma < 0.75$, the initiator is positioned around the middle of the flock (**Fig.R16c**). When the $\gamma \geq 0.75$, the initiator is positioned around the border of the flock (**Fig.R16d**).

Secondly, we present the simulation results of collective spin with different positions of the initiator. We analyzed the impact of the initiator's position on information propagation in the collective spin from four perspectives: the emergence of scale-free correlation, changes in information transfer speed, the maximum spinning lag, and group polarization. As shown in **Fig.R17a-c**, we found that the initiator's position has negligible impact on the simulation outcome of the collective spin. Specifically, the correlation length consistently increases linearly with the flock size, when the initiator is positioned around the center, middle and border of the group, indicating the emergence of scale-free correlation with different initiator's positions. Moreover, both information transfer speed and the max spinning lag grow with the increasing group size, while group polarization exhibits a decreasing trend. These findings are consistent with the simulation results demonstrated in the main text **Fig.2**.

Finally, we show the simulation results of collective turn with different positions of the initiator. We compare the results from the response accuracy, responsiveness, change in information transfer speed and the emergence of scale-free correlation. As shown in **Fig.R18**, we found that the simulation outcome of collective turn is barely affected by the initiator's position. No matter whether the initiator is located at the center, middle, or border of the group, the group with BOC-based interaction not only exhibits the high response accuracy in quickly responding to the initiator's sudden turns (**Fig.R18a,b**), but also demonstrates the emergence of scale-free correlation within the group (**Fig.R18c,e**). Additionally, the information transfer speed also shows an growing trend with the increase of group size (**Fig.R18d,f**). These simulation results align with those presented in the main text **Fig.3**.

In summary, regardless of the initiator is positioned at the center, middle, or border of the group, the group with BOC-based interaction demonstrates efficient information propagation in both experiments of collective spin and collective turn, while also exhibiting the non-trivial scale-free correlation. In the revised manuscript, we have added the above detailed simulation results in **Supplementary Sec.9** and included the relevant conclusions in main text (see **main text: pages 5-6, lines 240-245**):

“Furthermore, we have also analyzed the impact of the informed individual’s position on the BOC-based interaction in both simulation experiments of collective spin and collective turn. The simulation results demonstrated that the BOC-based interaction facilitates information transfer within the group and exhibits non-trivial scale-free correlation, regardless of whether the informed individual is positioned at the center, middle, or border of the group (see Supplementary Sec.9 for detailed simulation results).”

Point 2.10: In the methods, it would be great to give some intuitive explanation of what "normalized out-distance [44]" is.

Response: We thank Reviewer #2 for this critical suggestion and apologize for not making this point clear in the previous manuscript. In fact, the normalized out-distance is calculated as the $\frac{d_i^{out}}{N-1}$, which signifies the hierarchical layer of each node (individual) in the leader-follower network. Specifically, the leader-follower network is essentially an unweighted directed graph. In such a hierarchical network, some individuals act as leaders while others are followers, with directed connections between them, i.e., the leader points to its followers. We leverage the number of connections pointing to other nodes to represent the leadership of the focal individual, which refers to the out-distance (or the out-degree) of nodes d_i^{out} in the directed graph. The node (individual) with higher out-degree denotes stronger leadership, as they have more connections pointing to other nodes (followers). Normalization through division by $N - 1$ enables the comparative and consistent assessment of leadership among different individuals.

In the new version of manuscript, we have included the intuitive explanation of normalized out-distance to enhance clarity (see **main text: page 11, lines 489-496**):

“After obtaining the leader-follower network, we use the normalized out-distance⁵⁷ of each node in the network to characterize each fish’s leadership. The normalized out-distance signifies the hierarchical layer of each node (individual) in the leader-follower network, which is essentially an unweighted directed graph. In such a hierarchical network, some individuals act as leaders while others are followers, with directed connections between them, i.e., leaders point to its followers. We leverage the number of connections pointing to other nodes to represent the leadership of the focal individual, which refers to the out-distance (or the out-degree) of nodes in the directed graph. The node (individual) with higher out-degree denotes stronger leadership, as they have more connections pointing to other nodes (followers).”

Point 2.11: Page 8 line 368: prioritize  select (prioritize means giving some weight also to other elements)

Response: We thank Reviewer #2 for this constructive suggestion. We have revised this point in the new version of manuscript (see **main text: page 12, lines 538-539**):

“ Θ is the Heaviside function, which serves as a binary switch to select the neighbor with the largest BOC.”

Reviewer #2 (Remarks on code availability):

Point 2.12: The code is insufficient for reproducibility. The code is only providing scripts that generate plots, starting from data. Although this is ok for fish experiments, it is unacceptable for

the simulations and swarm robotics setup. There should be at the very least the code to perform the two simulations: spins and collective turn, and better if also the swarm robotics code.

Response: We thank Reviewer #2 for this critical comment. In the new repository of codes, First, we additionally provided the program to reproduce the simulation experiments of collective spin and collective turn in pybullet simulator. Second, due to the technical complexity involved in developing code for microcontrollers with C++, which may pose challenges for researchers without the experience in embedded systems development when attempting to use the SwarmBang robot, we provided the similar robotic control code based on pybullet simulators with the 3D physics model of our robots. The provided code mirrors the motion control approach used in the SwarmBang robots, ensuring a close alignment of motion control between simulation and real robot. Additionally, the codes that simulates the vision-sensing by the pinhole camera model in swarm robotic experiments are also provided. All the codes and example data related to this work are placed in the GitHub repository, which can be found at:https://github.com/DerekZhengEvosil/Body_orientation_change_of_neighbors_emerges_scale_free_correlation. The detailed descriptions of codes can be found in the “readme.md” file in the GitHub repository. The GitHub link is also included in the Code availability in main text (see **main text: page 13, lines 581-582**).

Finally, we thank Reviewer #2 again for her/his very insightful and constructive comments. We hope our responses above have addressed those very legitimate issues/concerns in a satisfactory manner.

Response to Reviewer #3

Point 3.0: In this paper, first body oriented change (BOC) from fish's eye perspective is analyzed. For this purpose, fish data is analyzed and reported in paper. A flocking method is proposed based on BOC-based interactions. The performance of this new method is compared with the classical Vicsek model using accuracy and responsiveness. In addition to simulation based experiments, the method is also validated using a swarm of 50 physical robots. The paper is written clearly and relatively easy to understand.

Response: We sincerely appreciate the valuable feedback and assessment provided by Reviewer #3. Next, we addressed each of her/his valuable comments and critical suggestions in order.

Point 3.1: I would definitely change the title of the paper, "Body Orientation Change Emerges Scale-Free Correlation in Collective Motion" that was really hard for me to understand before reading the paper.

Response: We thank Reviewer #3 for this critical comment and apologize for not making this point clear in the previous version of manuscript. We acknowledged that the concept of Body Orientation Change (BOC) is confusing for the first-time readers of our work. The confusion may stem from our straightforward introduction of the BOC without clearly specifying whose BOC we are referring to—*the focal individual's or its neighbors'*. This further involves the ambiguity in understanding the body orientation change in the context of collective motion. To facilitate the understanding of the key findings from the title, we have revised the title of this paper from "*Body Orientation Change Emerges Scale-Free Correlation in Collective Motion*" to "***Body Orientation Change of Neighbors Emerges Scale-Free Correlation in Collective Motion***".

Point 3.2: Additionally, as shown in Fig.2-(b3), it is evident that the larger groups exhibited a more stable direction of information transfer with the BOC-based interaction.  How did you come up with that result?

Response: We thank Reviewer #3 for this critical comment. In Fig.2-(b3) of the previous version of manuscript, we demonstrated the distribution of the direction of information transfer θ_s with different group sizes and illustrated the average information transfer direction for each group size using black lines. Indeed, we observed a decreasing trend in the average information propagation direction as the group size increased. However, upon re-evaluation of this simulation results and the definition of θ_s , we recognized that the observed trend does not imply an increase in stability of the information propagation direction within the group. In the new version of manuscript, we have removed the incorrect interpretation of this results. Moreover, as the analysis between basic properties of information transfer (e.g., information transfer direction, group polarization, max spinning lag) and group size in the collective spin is not directly relevant to the key finding of this work, we have moved the relevant analysis results to **Supplementary Sec.11** and **Supplementary Fig.30**. In **Supplementary Sec.11**, we have provided the detailed descriptions of these results (see **SI: pages 13, lines 443-459**):

“To demonstrate the basic property of information transfer in the simulation experiments of collective spin, we investigated the impact of group size on BOC-based interaction from the view of group polarization, max spinning lag, and direction of information transfer. As increase of group size, we found a decreasing trend in the group's polarization (Fig.30a) and higher maximal

spinning lag (Fig.30b) which is defined as the time that the last individual starts to spin lag behind the initiator (see Supplementary Sec.3 for the detailed definition). Besides that, we also analyzed the direction of information transfer θ_s within the group, which provides valuable insights into how information travels within the group. θ_s is approximated by the angle between the group velocity and the vector that points from the mean position of individuals who started spinning in the latter 20% to the mean position of individuals who started spinning in the first 20% (see Supplementary Sec.3 for the detailed definition). Particularly, $\theta_s = 0^\circ$ suggests that information transfers from the front-to-back, $\theta_s = 90^\circ$ means that it transfers from side-to-side, and $\theta_s = 180^\circ$ indicates the back-to-front transfer direction. As shown in Fig.30c, as the group size increases, the decreasing trend of θ_s implies that the flow of information transfer becomes more aligned with the front-to-back direction of the simulation setup.”

Point 3.3: In several places of the paper, you criticize use of velocity in swarm models. "Secondly, interaction mechanisms adopted in mathematical models rely on physical measurements that might not be able to be acquired through local perception. For example, velocity [14], bearing change [37]," but in your model you also use velocity as described by: "the reaction rule involves copying the neighbor's velocity in the simulation experiments of collective turn". Can you please elaborate on that?

Response: We thank Reviewer #3 for this critical comment. We acknowledge that the criticism of using velocity is not appropriate as it implies that using velocity in swarm models is inherently flawed or should be avoided. Indeed, the neighbors' velocity plays a significant role in the emergence of collective motion and the practical applications of swarm robotics. In our simulation experiments of collective turn, we incorporated the velocity of the selected neighbor in the stage of motion decision (as detailed in Eq. (7) in the main text). The reason lies in that: on the one hand, despite many researchers have endeavor to estimate the neighbors' velocity through local perception, there remains a significant gap in obtaining real-time and accurate velocity information from the first-person perspective. On the other hand, in this work, we focus on the issue of revealing the visual cues that individuals adopt to select which neighbor to interact with to achieve the collective response, which is the fundamental question that need addressing in modeling the collective motion. For the simulation experiments of collective turn, we intended to demonstrate the advantage of BOC-based interaction and the emergence of scale-free correlation in more complex and realistic interaction scenario. Recognizing the technical difficulties of obtaining neighbors' velocity from the first-person perspective in the status quo, we assumed that the focal individual has access to the neighbors' velocity in the BOC-based interaction. In the new version of manuscript, we have removed the inappropriate statements regarding the use of neighbors' velocity and rephrased the relevant sentences (see **main text: page 7, lines 301-311**):

“Secondly, although previous swarm models successfully reproduce the spatiotemporal patterns of biological swarms, the underlying fundamental principles governing these fascinating collective motions remain incompletely elucidated. This is because these interaction mechanisms often rely on the physical measurements derived from the computational theories and handcrafted designs, leading to the lack of biological evidence^{9,13,14}. In spite of the significant impact of these physical measurements on collective motion, it is important to explore the local interaction from the first-person perspective of individuals to gain a deeper understanding of the onset of collective motion⁴⁵. Since the macro-level phenomena emerging from the individual level, such as U-turn behavior, arise from individuals adjusting their movements based on first-person perspective visual observation.”

Moreover, we have included assumptions and explanations of using the velocity information in the new version of manuscript (see **main text: page 9, lines 405-408**):

“Due to the challenges in acquiring neighbors’ velocity from the first-person view¹⁹, we assumed that the focal individual could access the neighbors’ velocity. The reaction rule in the simulation experiments of collective turn is the commonly used velocity alignment⁴⁰, i.e., average the headings of the selected neighbor and the robot’s own heading (Eq. (7)).”

Point 3.4: I like the fact that there are real robot experiments but i think that the real robot experiments are oversimplified. As you describe in the supplementary material: "the experimental set-up is in the pseudo-distributed manner, which means the positions and headings of each robot are obtained from the motion capture system. Besides that, the server computer yields the desired angular and linear velocities for each robot based on the interaction rules. Then, the desired angular and linear speeds are transmitted to each robot through 2.4g wireless communication. Finally, each robot executes the received motion commands". As far as I understood, the computation is not even performed by the robots, all is centralized. I think the major problem is not that. In the paper, in several places you mention about vision-based sensing and collective motion. I know that you cannot add a camera to your robots but it would be great if you can simulate vision-based sensing in your robots and redo the experiments. In this way, you can justify the use of your model.

Response: We thank Reviewer #3 for this very constructive comment and excellent suggestions. As pointed out by the Reviewer #3, we acknowledged that the swarm robotic validation system is in the pseudo-distributed manner that most of the computation and sensing is achieved by the server computer. The reason lies in the low wireless communication bandwidth of our robots (31 bytes, 248 bits, see **Supplementary Sec.6.4** or **Fig.R12**), transferring the global information such as the positions and headings of the large-scale robotic swarm is quite challenging. The transmission of the global information incurs significant time delay in the control of swarm robots, which further hinders the validation of swarm model in the large-scale robotic swarm. For example, considering that the robot's x position, y position, and heading are float data types, transmitting the pose information of one robot requires 12 bytes of bandwidth. Scaling this up to 50 robots would require 600 bytes, necessitating approximately 20 times transmissions to broadcast all the information based on our wireless communication module. Given the transmission distance and potential electromagnetic signal interference, completing this transmission would take at least 4 seconds (the maximum transmission interval without packet loss is 0.2 seconds in the real test). Hence, we are compelled to rely on the external computation source for the majority of sensing tasks, local perception simulation, and model computation.

Additionally, it is noteworthy that although the collection of motion states of all robots and swarm model runs on the external server computer, the outcome derived from the BOC-based interaction remains computationally consistent, regardless of whether the local interaction mechanism is computed on the robots or on a server computer. It is because both approaches rely on the global information such as the positions and headings centrally perceived from the server computer to simulate the local perception of the robots. Each robot independently makes its motion decisions based on the same simulated local information.

In the revised robotic experiments, as suggested by Reviewer #3, to reinforce the validation of BOC-based interaction with local visual information, we have simulated the vision-based sensing of robots in swarm robotic validation system and redone the robotic experiments with 50 real robots to justify the use of BOC-based interaction. Specifically, we adopt the pinhole

camera model to simulate the vision-based sensing of robots. Based on the pinhole camera model, we can obtain projections of the neighboring robots onto the simulated image plane, allowing us to simulate the visual perception of the robots and approximate the body orientation change (BOC) using the generated bounding boxes based on the simulated images (see **Supplementary Sec.6.2** or **Fig.R8**). Detailed experimental results of these two parts are shown as follows:

■ **Robotic experiments based on the simulated vision sensing of robots through the pinhole camera model**

The pinhole camera model is a commonly used and effective technique for simulating visual sensing process, which has successfully supported the research in the following published papers:

- Li H, Qiu J, Yu K, et al. Fast safety distance warning framework for proximity detection based on oriented object detection and pinhole model. *Measurement*, 2023, 209: 112509.
- Fahimipour A K, Gil M A, Celis M R, et al. Wild animals suppress the spread of socially transmitted misinformation. *Proceedings of the National Academy of Sciences*, 2023, 120(14): e2215428120.
- Harpaz R, Nguyen M N, Bahl A, et al. Precise visuomotor transformations underlying collective behavior in larval zebrafish. *Nature communications*, 2021, 12(1): 6578.

As shown in **Fig.R8a**, the pinhole camera model describes the mathematical relationship between the coordinates of a point in three-dimensional space and its projection onto the simulated image plane. Consider a 3D point $p = (p_x, p_y, p_z)$ shown in **Fig.R8a**, the position of (u_x, u_y) on the simulated camera plane is calculates as:

$$\begin{cases} u_x = f \frac{p_x}{p_z} \\ u_y = f \frac{p_y}{p_z} \end{cases}$$

where f is the focal length of the camera.

For the revised swarm robotic experiments, each robot is regarded as an ellipsoid with the same aspect ratio as the real robot. Using the pinhole camera model, we can obtain the projections of neighboring robots (ellipsoids) onto the simulated camera's image plane, where the shape of the projection varies as the robots' orientation changes (**Fig.R8b-e**). To approximate the magnitude of BOC from the simulated images, we used the rectangular formed by the maximum X and Y ranges of the robot's projection as the simulated bounding box (red rectangular in **Fig.R8b-e**) and estimated the BOC by the variation in the area of these bounding boxes. The relationship between the robot's orientation and the corresponding area of the simulated bounding box is depicted in **Fig.R8f**. From the **Fig.R8f**, we found that as the robot's orientation continuously changes, the area of bounding box on the simulated image plane also grows steadily, which provides evidence that the variation in the area of the simulated bounding box is practical to estimate the BOC of neighboring robots. To streamline the robotic experiments, we assume the robot has omnidirectional vision, which is a common hardware setup in vision-based swarm robotics, such as the following robotic platform:

- Schilling F, Schiano F, Floreano D. Vision-based drone flocking in outdoor environments, *IEEE Robotics and Automation Letters*, 2021, 6(2): 2954-2961;
- Anoop A S, Kanakasabapathy P. Review on swarm robotics platforms, 2017 International Conference on Technological Advancements in Power and Energy (TAP Energy). IEEE, 2017: 1-6;

- Karimi M, Ahmadi A, Kavandi P, et al. WeeMiK: A low-cost omnidirectional swarm platform for outreach, research and education, 2016 4th International Conference on Robotics and Mechatronics (ICROM). IEEE, 2016: 26-31;
- Klingner J, Kanakia A, Farrow N, et al. A stick-slip omnidirectional powertrain for low-cost swarm robotics: Mechanism, calibration, and control, 2014 IEEE/RSJ International Conference on Intelligent Robots and Systems. IEEE, 2014: 846-851;

In particular, it is necessary to implement the random-based interaction as the baseline model to demonstrate advantage of BOC-based interaction and the significant role of BOC in yielding to the scale-free correlation (as suggested by Reviewer #2, see **Point 2.2**). Hence, in the new version of manuscript, we presented the comparative analysis of robotic experiments between the BOC-based interaction and random-based interaction in main text to maintain the integrity of this work. The comparison results between BOC-based interaction and Vicsek-like interaction are placed in **Supplementary Sec.10** (see **SI: page: 12, lines 431-442** or **Fig.R14**). Unless otherwise specified, the model parameters selection in the robotic experiments can be found in **Supplementary Table 3** for BOC-based interaction, random-based interaction and Vicsek model. The detailed results of robotic experiments are shown as follows:

As shown in **Fig.R13**, we presented the new results of the robotic experiments with BOC-based interaction and random-based interaction. From the **Fig.R13a-b**, we compared the trajectories of BOC-based interaction and random-based interaction with the group size of 50 robots. Notably, the trajectory of the group with BOC-based interaction exhibits the perfect zigzag motion pattern, indicating that the group with BOC-based interaction rapidly responds to the turning of informed individuals with optimal response accuracy ($\delta_{resp} \approx 1$) and quickly restores the group polarization back to the high consensus ($\phi \approx 1$). However, for the random-based interaction, the group fails to respond to the turning of informed individuals, and the response accuracy of the group remains low (δ_{resp} always below 0.5), further leading to the decreasing trend in group polarization.

In **Fig.R13c-e**, we demonstrated the statistical comparison of the response accuracy and responsiveness between the BOC-based interaction and random-based interaction. We found that the group with BOC-based interaction exhibits the high response accuracy close to $\delta_{resp} \approx 1$ (**Fig.R13c-d**) and the high responsiveness ($R \leq 0.2$) (**Fig.R13e**), whereas the response accuracy of the group with random-based interaction ranges between 0.2 and 0.4 (**Fig.R13c-d**) with lower responsiveness (**Fig.R13e**). Additionally, we also compared the information transfer speed of these two interaction mechanisms. As shown in **Fig.R13f**, we illustrated the averaged curves of the information transfer distance d_{trans} as a function of turning delay and estimated the information transfer speed by the slope of the curves. From the **Fig.R13f**, we found that the group with BOC-based interaction exhibited significantly faster information transfer speed V_s compared to the group with random-based interaction.

Moreover, we have also conducted the robotic experiments with the group size of 20 and 30 to perform the scale-free correlation analysis in swarm robotics. As shown in **Fig.R13g**, we found that for the BOC-based interaction, the correlation length increases with the size of the robotic swarm, indicating the emergence of the non-trivial scale-free correlation in real robotic swarms. However, the random-based interaction cannot exhibit the similar macro-level phenomena, as the correlation length remained constant with the increasing flock size. In the new version of main text, we have included the above robotic experiment results (see **main text: page 6, lines 249-292**) and the details of the visual sensing

simulation is added in **Methods** and **Supplementary Sec.6.2**, respectively. The robotic experiment videos of BOC-based interaction, random-based interaction, and Vicsek model can be found in **Supplementary videos 9-11**.

Reviewer #3 (Remarks on code availability):

Point 3.5: It seems that the code is available.

Response: We thank Reviewer #3 for this valuable comment. In the new repository of codes, First, we additionally provided the program to reproduce the simulation experiments of collective spin and collective turn in pybullet simulator. Second, due to the technical complexity involved in developing code for microcontrollers with C++, which may pose challenges for researchers without the experience in embedded systems development when attempting to use the SwarmBang robot, we provided the similar robotic control code based on pybullet simulators with the 3D physics model of our robots. The provided code mirrors the motion control approach used in the SwarmBang robots, ensuring a close alignment of motion control between simulation and real robot. Additionally, the codes that simulates the vision-sensing by the pinhole camera model in swarm robotic experiments are also provided. All the codes and example data related to this work are placed in the GitHub repository, which can be found at:[https://github.com/DerekZhengEvosil/Body orientation change of neighbors emerges scale free correlation](https://github.com/DerekZhengEvosil/Body_orientation_change_of_neighbors_emerges_scale_free_correlation). The detailed descriptions of codes can be found in the “readme.md” file in the GitHub repository. The GitHub link is also included in the Code availability in main text (see **main text: page 13, lines 581-582**).

Finally, we thank Reviewer #3 again for her/his very insightful and constructive comments. We hope our responses above have addressed those very legitimate issues/concerns in a satisfactory manner.

Summary of Revisions in the main text and Supplementary Information

First of all, we would like to appreciate the reviewers' meticulous review and constructive feedback, which have significantly improved the quality of our work. In this round of revision, we have addressed concerns raised by the reviewers point by point and made several significant improvements to our manuscript. Specifically, we have rewritten the results of simulation and robotic experiments for better clarity, refined the sentences for clarity of problem statements and important assumptions, enhanced the clarity of figures, added new simulation comparison experiments, redone the robotic experiments, and provided codes to enhance the reproducibility of our work. Here, we have provided the detailed account of all revisions made to the main text and supplementary information (in decreasing order of importance):

i. redo the robotic experiments with the simulated vision-sensing of robots

As stated in **Point 3.4**, we have redone the robotic experiments with the simulated vision-sensing through the pinhole camera model. Based on the pinhole camera model, we can obtain projections of the neighboring robots onto the simulated image plane, which allows us to approximate the body orientation change (BOC) using the generated bounding boxes based on the simulated images. The new results of robotic experiments demonstrate the feasibility of the BOC-based interaction using the simulated local visual information. Additionally, we also conducted the robotic experiments with the group size of 20, 30 and 50 robots to perform the analysis of scale-free correlation in swarm robotics. We have rewritten the results of robotic experiments in the new version of manuscript (see **main text: page 6, lines 249-292**). The method of simulating vision-sensing of robots have been included in the **Methods** section (see **main text: page 13, lines 568-575**) and detailed descriptions have been added in **Supplementary Sec.6.2** (see **SI: pages 8-9, lines 278-301**).

ii. incorporating the baseline model in the simulation and robotic experiments to demonstrate the significant role of BOC in the emergence of scale-free correlation

As stated in **Point 2.2**, to demonstrate the significant role of BOC in the emergence of scale-free correlation, we have compared the BOC-based interaction with random-based interaction and nearest neighbor-based interaction from the perspective of ability of information transfer and the emergence of scale-free correlation. However, due to inherent unfairness of the nearest neighbor-based interaction, i.e., an individual's perceptual radius is restricted to the relative distance of its nearest neighbor. Hence, we chose the random-based interaction as the baseline model in both simulation and real robot experiments. Also, we have included the relevant results in the new version of main text. We have rewritten the results of simulation experiments of collective spin and collective turn and the results of real robots (see **main text: pages 4-6, lines 145-292; Fig. 2; Fig. 3; Fig.4**). The detailed definition of baseline model has been included in **Supplementary Sec.5** (see **SI: pages 7-8, lines 220-243**).

iii. discussing on how BOC can be implemented on the robot using the bounding box over consecutive frames

As stated in **Point 2.6**, we validated the feasibility of calculating BOC with consecutive bounding boxes based on the simulated RGB camera images captured in the pybullet simulator. Based on the simulated RGB images, we obtained the bounding boxes of neighbors

and successfully estimated the BOC by calculating the consecutive changes in the bounding box area, which provided the promising route to implement the BOC on the real-world application of swarm robotics. We have included the discussion of this point in the new version of manuscript (see **main text: page 8, lines 344-352**) and added the relevant results in **Supplementary Sec.8** (see **SI page 11, lines 368-394**).

iv. providing the codes to reproduce the simulation experiments of collective spin, collective turn, and the robotic control code in pybullet simulator with a 3D robot physics model

As stated in **Point 2.12** and **Point 3.5**, we have provided the code to reproduce the simulation experiments of collective spin and collective turn. Additionally, the same robotic control code based on pybullet simulators with the 3D physics model of our robots is also included in the new GitHub repository. The provided code mirrors the motion control approach used in the robots, ensuring a close alignment of motion control between simulation and real robot. All the codes and example data related to this work can be found in the following link: https://github.com/DerekZhengEvosil/Body_orientation_change_of_neighbors_emerges_scale_free_correlation.

v. providing details on the swarm robotic validation system

As stated in **Point 2.5** and **Point 3.4**, we have provided additional details of swarm robotic validation system from three aspects: 1) workflow of the swarm robotic validation system; 2) hardware design and motion control of the SwarmBang robot; 3) division of transmission frequency and customized communication protocol. In the new version of manuscript, we have added the detailed descriptions on the workflow of the validation system (see **main text: pages 12-13, lines 542-566**) and included these additional information of the swarm robotic validation system in **Supplementary Sec.6** (see **SI: pages 8-10, lines 244-338**).

vi. clarifying the purpose and relationship with previous question of real robot experiments

As stated in **Point 1.6**, we have clarified the purpose of real robot experiments from three aspects: 1) validation of feasibility and applicability of BOC-based interaction under realistic and non-ideal conditions; 2) validation of emergence of the scale-free correlation in swarm robotics with BOC-based interaction; 3) The crucial part in research chain from biological data to bio-inspired mechanisms and further to swarm robotics. We have added the sentences to clarify the purpose of real robot experiments in the new version of manuscript (see **main text: page 6, lines 249-261; page 2, lines: 75-77; page 7, lines 329-333; page 6, lines 289-292**).

vii. investigating the impact of different positions of spin/turn initiators on the simulation experiments of collective spin and collective turn

As stated in **Point 2.9**, we conducted simulation experiments of collective spin and collective turn with different positions of the spin/turn initiators (informed individual). Through extensive simulations, we found that the BOC-based interaction rapidly propagates the information and emerges the scale-free correlation regardless of the position of the initiators. In the new version of manuscript, we have added the relevant conclusions in the **Results** section (see **main text: pages 5-6, lines 240-245**) and included detailed simulation results in **Supplementary Sec.9** (see **SI: pages 11-12, lines 395-430**).

viii. changing the title of the manuscript

As stated in **Point 3.1**, due to the lack of clarity in the title regarding whether the BOC refers to the focal individual or the neighbors, the title becomes difficult to understand for the first-time readers. Hence, in the new version of manuscript, we have changed the title of our work from “*Body Orientation Change Emerges Scale-Free Correlation in Collective Motion*” to “**Body Orientation Change of Neighbors Emerges Scale-Free Correlation in Collective Motion**” to facilitate the understanding of the key findings from the title.

ix. clarifying the principle and intuitive understanding behind the BOC

As stated in **Point 1.8** and **Point 2.1**, we have added the following sentences to clarify the principle behind the BOC and provided the intuitive understanding of BOC:

- see the **Introduction** in the new main text: page 2, lines 55-59;
- see the **Abstract** in the new main text: page 1, lines 17-18;
- see the **Introduction** in the new main text: page 2, lines 52-53;
- see the **Results** in the new main text: page 3, lines 141-142;
- see the **Abstract** in the new main text: page 1, lines 11-15.

x. clarifying the problem statement in the Abstract, Introduction and how the BOC addressed the problem in the Discussion

As stated in **Point 1.7**, **Point 1.8**, **Point 2.1**, **Point 2.4**, **Point 2.6**, **Point 3.3** and **Point 3.4**, we have refined most of the sentences in the **Abstract**, **Introduction**, and **Discussion** sections to clarify the motivations and contributions of our work:

- see the **Abstract** in the new main text: page 1, lines 8-15;
- see the **Abstract** in the new main text: page 1, lines 17-18;
- see the **Introduction** in the new main text: page 1, lines 34-40;
- see the **Introduction** in the new main text: page 2, lines 52-53;
- see the **Discussion** in new main text: page 8, lines 347-352.

xi. clarifying the important assumptions and methods in the empirical data analysis, simulation experiment setup, swarm robotic experiment setup

As stated in **Point 1.1**, **Point 1.3**, **Point 1.4**, **Point 1.5**, **Point 2.1**, **Point 2.5**, **Point 2.7**, **Point 2.10**, and **Point 3.3**, we have revised the relevant sentences in the **Abstract**, **Introduction**, **Results** and **Methods** sections:

- see the **Results** in the new main text: pages 2-3, lines 81-96;
- see the **Results** in the new main text: page 4, lines 145-159;
- see the **Methods** in the new main text: pages 8-9, lines 375-412;
- see the **Methods** in the new main text: page 12, lines: 523-532;
- see the **Methods** in the new main text: page 9, lines 416-419;
- see the **Discussion** in the new main text: page 7, lines 315-318;
- see the **Results** in the new main text: page 5, lines 205-209;
- see the **Abstract** in the new main text: page 1, lines 17-18;
- see the **Introduction** in the new main text: page 2, lines 52-53;
- see the **Results** in the new main text: page 4, lines 145-147;
- see the **Abstract** in the new main text: page 1, lines 11-15;
- see the **Introduction** in the new main text: page 2, lines 55-59;

- see the **Methods** in the new main text: page 12, lines 543-566;
- see the **Methods** in the new main text: page 10, lines 437-450.

xii. improving the readability of the manuscript and reviewing more related work

As stated in **Point 1.0, Point 1.2, Point 1.7, Point 2.3, Point 2.7, Point 2.11** and **Point 3.2** we have revised corresponding sentences and enhance the clarity of figure in the new version of manuscript:

- see main text: page 7, lines 299-301;
- see main text: page 3, lines 100-108;
- see main text: page 11, lines 489-496;
- see main text: page 11, lines 502-503;
- see main text: page 4, lines 145-159;
- see main text: pages 8-9, lines 375-412;
- see main text: page 4, lines 174-177;
- see main text: page 4, lines 188-193;
- see main text: page 5, lines 230-232;
- see main text: page 1, lines 9-15;
- see main text: page 1, lines 17-18;
- see main text: pages 1-2, lines 42-51;
- see main text: page 2, lines 52-53;
- see main text: page 4, lines 188-193;
- see main text: page 10, lines 437-451;
- see main text: pages 11, lines 489-496;
- see main text: pages 2-3, lines 93-96;
- see main text: page 8, lines 375-379;
- see main text: page 6, lines 255-266;
- see main text: page 12, lines 538-539;
- see the Fig.2 in the new main text: page 19.

xiii. moving some simulation results that not directly relevant to the key findings into the Supplementary information

As stated in **Point 3.2**, in the revised manuscript, we have removed the incorrect interpretation on information transfer direction in the simulation experiments of collective spin. Moreover, as these analysis between basic properties of information transfer and group size in the collective spin (e.g., information transfer direction, group polarization, max spinning lag) is not directly relevant to the key finding of this work, we have moved the relevant analysis results to **Supplementary Sec.11** and **Supplementary Fig.30** (see **SI: pages 13, lines 443-459**).

Response Figure

Fig.R1 | The illustration of simulation experiments of collective spin and collective turn. a, In the simulation experiments of collective spin, we assume that each simulated robot has two motion states: one is stationary, and the other is spinning. The stationary state means that both the linear and angular speed of the simulated robot are set to 0. The spinning state means that the angular speed is set to the maximum ω_{max}^{sim} (see **Supplementary Table 1** for parameters selection) while the linear speed is set to 0, indicating the rotational movement of robots. The simulated robots are required to swiftly transfer the spinning state triggered by a single informed individual (spinning initiator) throughout the entire group. The informed individual starts spinning at a pre-set activation time and stops spinning after completing a 2π rotation, which is selected at the forefront of the group. Each simulated robot calculates the BOC of its neighbors within a perception range R_{visual} and selects the neighbor with the maximum BOC to react, i.e., the BOC-based interaction. The reaction rule in the collective spin has the simplest manner for the focal individual, which is to switch between stationary and spinning states based on the state of the selected neighbor. Specifically, if the selected neighbor is in a spinning state, the focal individual also enters the spinning state and starts to spin with the maximal angular speed; if the selected

neighbor is stationary, the focal individual switches to the stationary state and stop spinning. **b**, In the simulation experiments of collective turn, collective turn refers to the coordinated directional changes within the group, where simulated robots are required to respond promptly to a sudden turn initiated by one informed individual (turn initiator). The local interaction is the same as the experiments of collective spin, that each simulated robot calculates the BOC of its neighbors within a perception range R_{visual} and selects the neighbor with the maximum BOC to react. The reaction rule involves the commonly used velocity alignment, i.e., averaging the headings of the selected neighbors and one's own heading (Eq. (7) in the main text). Several assumptions are made to streamline the study of information propagation of BOC-based interaction in the simulation experiments of the collective turn. On the one hand, we assume the informed individual, positioned at the forefront of the group, turns with an angle θ_{info} relative to the group's movement direction at the pre-set moments. On the other hand, due to the challenges in acquiring neighbors' velocity from the first-person perspective, we assume that the focal individual could access the neighbors' velocity information.

Fig.R2 | The illustration of the involvement of the visual occlusion. In this work, we assume that individuals are non-transparent ellipses, implying that neighbors in close proximity to the focal individual may occlude neighbors that are further away. This occlusion prevents the focal individual from perceiving the distant neighbors within the visual perception range. **a**, shows how we determine the occluded neighbor in our simulations. Firstly, we identify the positions and angles of the tangent points (black points shown in panel **a**) where the rays originating from the focal individual's eye (red point shown in panel **a**) touch on the neighboring ellipses. Secondly, we calculate the intersection point of rays originating from the eye of the focal individual i through the tangent points on j_1 with the outlines of ellipses k (the red cross shown in panel **a**). If the intersection point exists, it means the occlusion occurs and we then remove the tangent point closest to the intersection point. Otherwise, the absence of intersections implies no occluded individual is in this certain direction. Thirdly, we numerically sort the angular positions of left tangent points for the focal individual in increasing order. Finally, for each segment (i.e., visual field) delimited by two ordered tangent points (light blue regions in panel **a**), we projected the bisectors (purple lines in panel **a**) of these segments to ascertain which ellipse is closest in this segment to find out the belongings. Through computations of these four steps, we can distinguish the distant neighbors that are occluded by nearby ones from the first-person perspective of the focal individual. **b-c**, we compared the impact of the presence or absence of visual occlusion on the process of neighbor selection. Specifically, with the involvement of visual occlusion, distant individuals (colored by the dark grey) are obscured by closer individuals, thus precluding being selected by the focal individual (**b**). However, with the absence of the visual occlusion, the focal individual can select any neighbors within its perception range (**c**), which is the common perception setup in swarm models, e.g., the metric interaction.

Fig.R3 | The comparison experiments of collective spin between the BOC-based interaction and random-based interaction. According to the simulation results, we found that the group with BOC-based interaction rapidly propagate the spinning state of the informed individual (colored by red) and achieves the collective spin (a,b). However, for the random-based interaction, the group failed to achieve the collective spin (e,f). For the emergence of scale-free correlation, in the case of the BOC-based interaction, we observed a linear increase in correlation length with flock size (c), suggesting the emergence of scale-free correlation. However, for the random-based interaction, we found that correlation length did not increase linearly with flock size, indicating the absence of scale-free correlation (g). Additionally, we found that the information transfer speed in the random-based interaction remained lower than that in BOC-based interaction, without continually increasing with group size (d,h). The unit of information transfer speed is $mm s^{-1}$. The unit of flock size and correlation length are the mm , respectively. The

informed individual is activated at the 25th step in the simulation. The color rendering of individuals represents the length of the spinning lag, and the individuals who consistently remained stationary are colored with grey.

Fig.R4 | The comparison experiments of collective spin between the BOC-based interaction and Nearest neighbor-based interaction. a-d, we demonstrate the advantage of BOC-based in information transfer and the emergence of scale-free correlation. Similar to the random-based interaction, the group with the nearest neighbor-based interaction fails to achieve collective spin. We illustrate the worst case where only one individual responds to the informed individual (colored by red). This is because these two individuals are the nearest neighbor to each other, impeding the spread of information (e,f). Additionally, the nearest neighbor-based interaction also fails to exhibit scale-free correlation, as the correlation length remains nearly constant regardless of flock size (g). Additionally, the information transfer speed consistently remained low with increasing group size in the case of nearest neighbor-based interaction (h). The unit of information transfer speed is $mm s^{-1}$. The unit of flock size and correlation length are the mm , respectively. The informed individual is activated at the 25th step in the simulation. The

color rendering of individuals represents the length of the spinning lag, and the individuals who consistently remained stationary are colored with grey.

BOC-based interaction

Random-based interaction

Fig.R5 | The comparison experiments of collective turn between the BOC-based interaction and Random-based interaction. **a-c**, we demonstrate the advantage of information transfer in the collective turn with the turning angle of $\frac{\pi}{2}$ and π from the perspective of response accuracy and responsiveness, respectively. The BOC-based interaction rapidly responds to sudden turns with the high response accuracy regardless of the group size. Particularly, it is worth noting that for the random-based interaction, the group is able to achieve the collective turn when the group size is small ($N = 10$), but its response accuracy responsiveness keeps decreasing with the increase of group size (**f-h**), indicating the weak ability of information transfer. For the emergence of scale-free correlation, the BOC-based interaction exhibits a linear growing trend of correlation length with the increasing flock size (**d**), suggesting the emergence of scale-free correlation. Additionally, the information transfer speed keeps growing with the increase of group size in the case of BOC-based interaction (**e**). In contrast, we found that the correlation length for random-based interaction did not exhibit a trend of linear growth with the increase of flock size (**i**). And also, the information transfer speed also remained nearly constant with the increase of group size for the random-based interactions (**j**). The unit of information transfer speed is $mm\ s^{-1}$. The unit of flock size and correlation length are the mm , respectively.

BOC-based interaction

Nearest neighbor-based interaction

Fig.R6 | The comparison experiments of collective turn between the BOC-based interaction and Nearest neighbor-based interaction. **a-c**, we demonstrate the advantage of information transfer in the collective turn with the turning angle of $\frac{\pi}{2}$ and π from the perspective of response accuracy and responsiveness, respectively. The BOC-based interaction rapidly responds to the sudden turn with the high response accuracy regardless of the group size. However, it is worth noting that for nearest neighbor-based interaction, the group failed to achieve the collective turn with low response accuracy and responsiveness (**f-h**), indicating the shortcoming in the information transfer. For the emergence of scale-free correlation, the BOC-based interaction exhibits a linear growing trend of correlation length with the increasing flock size (**d**), suggesting the emergence of scale-free correlation. And the information transfer speed of BOC-based interaction also keeps growing with the increase of group size (**e**). In contrast, we found that the correlation length for nearest neighbor-based interaction did not exhibit a trend of linear growth with the increase of flock size (**i**). In addition, the information transfer speed also remained nearly constant with the increase of group size in the case of the nearest neighbor-based interaction (**j**). The unit of information transfer speed is $mm\ s^{-1}$. The unit of flock size and correlation length are the mm , respectively.

b Identify the bounding box from the camera view of the robot-2

c Calculate the BOC through area of bounding boxes over consecutive frames

Fig.R7 | Estimation of the BOC using bounding boxes over consecutive frames captured by the simulated RGB camera in the pybullet. **a**, we conducted the simulation experiments of collective turn with a group size of 10 to get the consecutive simulated RGB images from a certain robot's view. The informed individual is the robot-0, which is positioned at the forefront of the group. **b**, the robot-2 observed four robots in total (robots 0, 8, 6, and 9) and obtained the bounding boxes (red rectangular shown in panel **b**) of the four robots detected from the frames $T = 51s$ to $T = 54s$. **c**, we estimate the BOC based on the area of the bounding boxes over $2s$ consecutive simulated RGB image frames. We found that before the informed individual made the abrupt turn, the estimated BOC remained constant for all the perceived robots (robots 0, 8, 6, and 9). After the informed individual suddenly turned, the BOC of robot-0 (the informed individual) is the first to increase, followed by a rising trend in the BOC of the other three robots, indicating that changes in the bounding box area can be used to quantitatively reflect the BOC. The unit of magnitude of BOC calculated from the bounding box is $pixel^2$. The bounding box $(\mathbf{x}, \mathbf{y}, \mathbf{w}, \mathbf{h})$ is defined as a rectangular characterized by four parameters: the \mathbf{x} and \mathbf{y} coordinates of the top-left corner in the image, the box's width (\mathbf{w}), and the box's height (\mathbf{h}), which represents the position and size of an object within an image.

b oriented to 0° respect to x-axis

d oriented to 30° respect to x-axis

c oriented to 60° respect to x-axis

e oriented to 90° respect to x-axis

Fig.R8 | Simulate the vision sensing of robot through the pinhole camera model. **a**, the pinhole camera model describes the mathematical relationship between the coordinates of a point in three-dimensional space and its projection onto the simulated image plane. **b-e**, Using the pinhole camera model, we can obtain the projections of neighboring robots (ellipsoids) onto the simulated camera's image plane, where the size of this projection varies as the robots' orientation changes. To approximate the magnitude of BOC from the simulated images, we use the rectangle formed by the maximum X and Y ranges of these projection as the bounding box and quantify the BOC by the variation in the area of bounding box (red rectangle in **b-e**). **f**, the relationship between the robot's orientation and the corresponding area of simulated bounding box. We found that as the orientation of the robot continuously changes, the area of bounding box on the simulated image plane also grows steadily, which provides evidence that the variation in the area of the simulated bounding box is practical to estimate the BOC of neighboring robots.

Fig.R9 | The architecture of swarm robotic validation system. **a**, Overview of the miniature mobile robot (i.e., SwarmBang robot) used for swarm robotic experiments. The robot is a miniature platform ($60\text{ mm} \times 60\text{ mm} \times 60\text{ mm}$) and weighs 200g. The components of a robot are described as follows (numbers between parentheses refer to labels in **(a)**). The robot is built with two-layered structures to keep in line with the sensor-reaction loop in the swarm model, i.e., top sensor PCB board (1) and bottom actuation PCB board (2). They are directly connected by four copper bars (3) and communicate through universal serial ports. Both of them host a 32-bit, 72MHz ARM microprocessor (STM32F1 series). The top sensor board manages the overall logic computation and communicates wirelessly with the server computer using the radio module through the customized communication protocol (see **Supplementary Sec. 6.4** for detailed information). The bottom actuation board is for the robot motion control. Each robot is equipped with two 3D-printing wheels (4) actuated by the step motor (5), which can drive the robot with the maximum speed of 25 mm s^{-1} and the maximum turning angle of 0.83 rad s^{-1} (see **Supplementary Sec.6.3** for detailed information). A universal wheel (7) is mounted at the bottom to keep the robot standing upright. Two 3.7V rechargeable batteries (8) provided energy for about

1 hour in our experimental settings. The passive infrared reflective balls (9) for the NOKOV motion capture system are mounted on the elliptical deck (10). The main and minor axis of the deck are 150 *mm* and 80 *mm*, respectively. **b**, the experimental system is able to support 10^2 magnitudes of miniature robots. Due to the constraints of the experimental arena, a maximum of 50 robots were used in the collective turn experiments. **c**, the swarm robotics validation system (i.e., SwarmBang system) comprises three primary components: firstly, a server computer; secondly, a motion capture system; and thirdly, a large number of swarm robots. The server computer, equipped with a radio transmitter, is tasked with model computing, simulating local perception and transmitting real-time control commands to the robots. The motion capture system is used to locate the position of each robot.

Fig.R10 | The workflow of the revised swarm robotic validation system. The workflow of our validation system begins with the measurement of each SwarmBang robot's position and heading using the NOKOV motion capture system processed on the server computer (**step 1**). Next, in the revised manuscript, we involved the pinhole camera model to simulate the vision-based sensing of robots in server computer which can improve the realism of perceptual inputs of robots and also suggested by Reviewer #3 (please see the **Point 3.4**) (**step 2**). In the following, the server computer determines the desired heading θ_d for each robot based on the corresponding swarm model and the simulated local visual information (**step 3**). After obtaining the desired heading of each robot, the desired angular speed ω is calculated as the $\omega = \min(\frac{\Delta\theta}{\Delta t}, \omega_{max}^{robot})$, where the $\Delta\theta = \theta_d(t+1) - \theta(t)$ and ω_{max}^{robot} is the maximum angular speed in robotic experiments (see **Supplementary Table 3** for parameters selection in real robot experiments). Following that, the server computer calculates the desired linear and angular velocity of each robots and broadcast to all robots based on a wireless communication module through the customized communication protocol with a fixed time interval Δt (details of this protocol is shown below or the **Fig.R12**) (**step 4**). Upon receiving these desired motion commands, each robot then calculates the speed of its left and right wheels based on the kinematic model of the differential-drive platform in the robots (**step 5**).

Fig.R11 | The schematic of SwarmBang robot. The hardware design of the SwarmBang robot is separated into two parts: 1) the PCB board for decision-making and communications; 2) the PCB board for motion control and battery management. Two 3.7V rechargeable batteries (2*800mAh) provide energy for about 1 hour in our experimental settings. Each robot is equipped with a wireless communication module (NRF24L01) to receive commands from the server computer.

Fig.R12 | The division of transmission frequency in the wireless communication system and the customized wireless communication protocol. **a**, Transmitting motion commands for 50 robots with low communication bandwidth and tight latency requirements presents significant challenges for the robotic validation system. Hence, we developed a low-redundancy communication protocol and employed multiple communication frequencies to simultaneously transmit motion commands, with each frequency corresponding to 25 robots. Each robot's desired linear and angular speeds are represented using 9 bits: 1 bit for indicating the turning direction (0 for left, 1 for right), 3 bits for the desired linear speed, and 5 bits for the angular speed. Each data frame contains the desired linear and angular speeds for 25 robots, totaling $9 \times 25 = 225$ bits. Additionally, we have implemented a Cyclic Redundancy Check (CRC) in each robot to verify the correctness of the received data. Upon receiving a data frame, each robot locates its desired linear and angular speeds within the data frame based on its unique robot ID. **b**, In our robotic experiment, we employed frequency division multiplexing (2.42GHz and 2.46GHz) to separately control two groups of robots based on the communication protocol, each consisting of 25 robots, totaling 50 robots.

Fig.R13 | Swarm robotic experiments of BOC-based interaction and Random-based interaction. a-b, we compared the trajectories of BOC-based interaction and random-based

interaction with the group size of 50 robots. Notably, the trajectory of the group with BOC-based interaction exhibits the perfect zigzag motion pattern, indicating that the BOC-based interaction rapidly responds to the turning of informed individuals with optimal response accuracy ($\delta_{resp} \approx 1$) and quickly restores the group polarization back to the high consensus ($\phi \approx 1$). However, for the random-based interaction, the group fails to respond to the turning of informed individuals, and the response accuracy of the group remains low (δ_{resp} always below 0.5), further leading to the decreasing trend in group polarization. **c-e**, we demonstrate the statistical comparison of the response accuracy and responsiveness between the BOC-based interaction and random-based interaction. We found that the group with BOC-based interaction exhibits a high response accuracy close to 1 and the high responsiveness ($R < 0.2$), whereas the response accuracy of the group with random-based interaction is only ranging between 0.2 and 0.4 with low responsiveness. **f**, we also compare the information transfer speed of these two interaction mechanisms. We found that the group with BOC-based interaction exhibited significantly faster information transfer speed compared to the group with random-based interaction. **g**, we found that in BOC-based interaction, the correlation length increases with the size of the robot flock, highlighting the emergence of the non-trivial scale-free correlation in the real robotic swarms. However, the group with random-based interaction cannot exhibit the similar macro-level phenomena.

Fig.R14 | Swarm robotic experiments of BOC-based interaction and Vicsek model. **a**, The robotic experiment results demonstrate that using BOC-based interaction, the swarm with 50 robots not only successfully follows the informed robot to change its trajectory but also quickly responds to the heading change of the informed robot. **b**, As a comparison, the swarm using the Vicsek model is difficult to follow the informed robot. **c-d**, There are noticeable differences in

accuracy and responsiveness between BOC-based interaction and the Vicsek model. BOC-based interaction outperforms the Vicsek model in terms of response accuracy and responsiveness. **e**, The response accuracy as a function of time for the BOC-based interaction and the Vicsek model, respectively. **f**, The information transfer speed V_s of BOC is much faster than that of the Vicsek model.

Fig.R15 | The effect of the anisotropic factor α tuning the forward-oriented preference of individual. **a**, the $\left(\frac{1 + \hat{\mathbf{v}}_i \cdot \hat{\mathbf{x}}_{ij}}{2}\right)$ quantifies the frontal preference on perceiving neighbor- j 's movement from the first-person perspective of the focal individual i , ranging in $[0, 1]$. The dot product of these two vectors $\hat{\mathbf{v}}_i \cdot \hat{\mathbf{x}}_{ij}$ measures the relative bearing of neighbor- j . It simulates that as the first-person view of neighbor- j moves from the front to the back, the visual perception of focal individual i to the neighbor- j diminishes. For example, as shown in **a**, if $\angle(\hat{\mathbf{v}}_i, \hat{\mathbf{x}}_{ij}) = 0$, i.e., the dot product of $\hat{\mathbf{v}}_i \cdot \hat{\mathbf{x}}_{ij} = 1$, it signifies that neighbor- j_1 locates directly in front of the focal individual- i , and the focal individual could perceive the neighbor- j_1 as the $(1 + \hat{\mathbf{v}}_i \cdot \hat{\mathbf{x}}_{ij})/2$ is 1 (neighbor- j_1 shown in **a**). Once the neighbor- j gradually moves from the front to back respect to the focal individual i , i.e., $\angle(\hat{\mathbf{v}}_i, \hat{\mathbf{x}}_{ij})$ goes from 0 to π or $-\pi$, the visual perception to the neighbor- j could diminish since $(1 + \hat{\mathbf{v}}_i \cdot \hat{\mathbf{x}}_{ij})/2$ decreases from 1 to 0 (Neighbor j_2, j_3, j_4 in **a**). By incorporating the $\left(\frac{1 + \hat{\mathbf{v}}_i \cdot \hat{\mathbf{x}}_{ij}}{2}\right)$ with the perception of BOC, we could emulate the anisotropic perception of neighbor's body orientation change (Eq. (1) in the main text). **b**, to tune the effect of frontal preference, we involve the anisotropic factor α , i.e., $fp(t) = \left(\frac{1 + \hat{\mathbf{v}}_i \cdot \hat{\mathbf{x}}_{ij}}{2}\right)^\alpha$. When $\alpha = 0$, the focal individual i ignores the forward-oriented perception as the $fp(t)$ always equal to 1 (the green line shown in **b**), which means that the perception of BOC is not relevant to the relative positions of neighbors. In particular, increasing α ($\alpha \geq 1$) make Eq. (1) enhance the anisotropic effect of the visual perception, which causes individuals to gradually narrow the perception of nearby neighbors toward their frontal vision. For example, as shown in **b**, if $\alpha = 10$, it means that the perception of BOC $g_{ij}(T, \tau) = 0$ when the relative positions of neighbors are out of the visual sight $(-\pi/2, \pi/2)$.

Fig.R16 | The geometric definition of spatial center proximity index. **a**, the spatial center proximity index γ is the ratio of the relative distance between the initiator and the center to the radius of the flock, calculated as the $\gamma = \frac{d_i^c}{F_{radius}}$. d_i^c is the relative distance between the initiator and the center. F_{radius} is the radius of the flock, estimated by the half of the maximal relative distance among individuals. **b-d**, we demonstrate the spatial distribution of individuals with different γ . When the $\gamma \leq 0.45$, the initiator is positioned around the flock center (**b**). When the $0.45 < \gamma < 0.75$, the initiator is positioned around the middle of the flock (**c**). When the $\gamma \geq 0.75$, the initiator is positioned around the border of the flock (**d**).

Fig.R17 | The simulation experiments result of collective spin with different positions of initiators. We analyzed the impact of initiator's position on information propagation in the simulation experiments of collective spin from four perspectives: the emergence of scale-free correlation, changes in information transfer speed, the maximum spinning lag, and group polarization. As shown in **a-c**, we found that the position of initiators has negligible impact on the simulation outcome of collective spin. Specifically, the correlation length linearly increases with the flock size when the initiator is positioned around the center, middle and border of the group, indicating the emergence of scale-free correlation. In addition, with the increasing group size, the information transfer speed and the max spinning lag also grows, and the group polarization shows the decreasing tendency, which is consistent with the results demonstrated in the main text **Fig.2**. The unit of information transfer speed is $mm s^{-1}$. The unit of flock size and correlation length are the mm , respectively.

Fig.R18 | The simulation experiments result of collective turn with different positions of initiators. We compare the results of different position of initiator from the response accuracy, responsiveness, change in information transfer speed and the emergence of scale-free correlation. We found that the simulation outcome of collective turn is barely affected by the initiator's position. No matter whether the initiator is located at the center, middle, or border of the group, the group not only exhibits a high response accuracy in quickly responding to the initiator's sudden turns (a,b), but also demonstrates the emergence of scale-free correlation within the group (c,e). Additionally, the information transfer speed shows an increasing trend with the increase of group size (d, f). These results are consistent with those presented in the main text Fig.3. The unit of information transfer speed is $mm s^{-1}$. The unit of flock size and correlation length are the mm , respectively.

Response to Reviewer #1

Point 1.0: The revised version of the manuscript substantially improved in clarity and strength thanks to the additional explanations and control experiments. The results from behavioural analysis of fish, of simulated agents driven by the hypothesised control laws, and of physical robots provide strong evidence that Body Orientation Change could be a strong visual cue in supporting scale-free collective motion with faster signal propagation speed than other perceptual cues. As far as I could understand (see more about this below), the method and data are sound and the claims are supported by the experimental evidence. The revised manuscript and additional code in principle provide sufficient information to reproduce the results, although I haven't tested the code myself. The results and the proposed algorithms could be relevant not only for biologists and complex system scientists, but also for computer scientists in swarm robotics.

Response: We thank Reviewer #1 very much for reviewing our paper again and appreciate her/his positive assessment of our work on benefiting the disciplines from biology to computer science. Following her/his constructive comments and suggestions, we have carefully reviewed the entire manuscript to address not only the concerns raised by the reviewer but also revised any similar issues to improve the manuscript. In addition, based on the valuable feedback from the reviewer, we acknowledged that our previous writing style, i.e., placing many definitions and descriptions of important concepts in **Methods** section or Supplementary Information while directly referencing them in **Results section**, made it challenging for readers to understand our findings and follow the manuscript. In this round of revisions, we have revised our writing style by briefly defining each important concept before or immediately after its first mention to facilitate the readability of the manuscript, while providing detailed mathematical definitions and descriptions in **Methods** section to ensure the clarity of these concepts or methods.

There have been seven points of major revisions made to the manuscript:

- i. revise the syntax infelicities and writing errors in main text to improve the readability of the manuscript based on the reviewer's valuable suggestions and take instructions from the **Nature Research Editing Service** to ensure proper English language, grammar, punctuation, spelling, and overall style throughout the manuscript.
- ii. integrate the definitions of body orientation change (BOC), the effect of frontal preference and anisotropic parameter α into a new subsection ("**Quantifying Body Orientation Change to measure motion salience of neighbors**") at the beginning of the **Results** section to enhance the logical flow of our manuscript from "key definitions of modeling BOC – the crucial role of BOC in the biological swarm – investigation of BOC through simulation experiments – further validation of BOC with real robots".
- iii. add sentences in **Results** section to briefly clarify the definition of important factors and methods (e.g., information transfer speed, correlation function and correlation length, response accuracy, responsiveness, self-propelled model, pinhole camera model, etc.) before referring to them, while providing the detailed mathematical definitions in **Methods** section.
- iv. add five new subsections in **Methods** section to clarify the important factors and methods with rephrased descriptions of detailed computational process and mathematical

definitions, such as estimation of information transfer speed, correlation function and correlation length, and construction of BOC matrix and derivation of BOC-based motion salience, response accuracy, responsiveness, and random-based interaction, which were previously defined in Supplementary information but lacked sufficient clarity and were inconvenient to read.

- v. rewrite the **Discussion** section to clarify the significance, limitations and possible developments of our results in a concise manner.
- vi. revise the snapshots in figures of collective spin results (see **Fig.3** in main text) to better facilitate the understanding of the spinning state (or behavior) and reorder figures in main text to improve the logical flow of the manuscript.
- vii. change the title from “Body Orientation Change of Neighbors Emerges the Scale-free Correlation in Collective Motion” to “**Body Orientation Change of Neighbors Leads to Scale-free Correlation in Collective Motion**” to revise the writing errors in the title.

Point 1.1: However, in my opinion the manuscript still requires substantial editing before publication. I still had a hard time to properly understand or interpret the meaning of some sentences because of writing style and syntax infelicities. For example, the article "the" is used excessively and improperly, which often led me to incorrectly think that the authors referred to a new or a specific method; for example, at line 82: "we conduct the empirical data analysis" made me believe that by "empirical data analysis" they refer to a specific method developed by themselves, whereas they probably meant to say "we analyse experimental data from collection motion of fish". Analogously, the improper use of "a, an" and "the" made me believe that they assume the presence of only one factor, such as in line 52 "we aimed to reveal the visual cue..." instead of "we aimed at revealing what visual cue...". Also, at line 145: "inspire us to develop the pairwise interaction mechanism" induced me to understand that they propose a new algorithm or principle, whereas they probably meant to say "led us to hypothesise that pairwise interactions..." or something similar.

Response: We thank Reviewer #1 very much for carefully reviewing our manuscript and apologize for the inappropriate writing style and syntax infelicities in the previous version of the manuscript. In the revised manuscript, we have revised these syntax infelicities raised by the reviewer point by point (see below). Furthermore, recognizing our limitations in academic English writing, we have also enlisted the support of the **Nature Research Editing Service** to address any issues with article or verb usage and other writing errors in the revised manuscript.

Point 1.1.1: for example, at line 82: "we conduct the empirical data analysis" made me believe that by "empirical data analysis" they refer to a specific method developed by themselves, whereas they probably meant to say "we analyse experimental data from collection motion of fish".

Response: We thank Reviewer #1 very much for pointing that out and apologize for the inappropriate usage of the article “the” in the previous version of the manuscript. We have revised this point in the new version of manuscript to avoid misleading of results. For example,

- see **main text: page 3, lines 133-134:**

“To reveal the **importance** of body orientation change in biological swarms, we **analyzed experimental data** from spontaneous collective U-turn behaviors observed in rummy-nose tetra groups (*Hemigrammus rhodostomus*) recorded in a circular tank^{16,28,29}.”

- see **main text: page 2, lines 67-69:**

“Interestingly, **the analysis of real fish data** revealed that an individual fish with the larger BOC tends to lead other fish during the U-turn behaviors.”

- see **main text: page 4, line 145:**

“Notably, **the analysis of experimental data from the fish schools** is based on two key assumptions:”

- see **main text: page 24, caption of Fig.2:**

“**Analysis of experimental data from collective U-turn behaviors in fish schools.**”

Point 1.1.2: Analogously, the improper use of "a, an" and "the" made me believe that they assume the presence of only one factor, such as in line 52 "we aimed to reveal the visual cue..." instead of "we aimed at revealing what visual cue...".

Response: We thank Reviewer #1 very much for pointing that out and apologize for the inappropriate usage of the article “the” in previous version of the manuscript. We have revised this point in the new version of manuscript (see **main text: page 2, lines 52-53**):

“In this study, **we aimed at revealing what visual cue individuals adopt from the first-person perspective to** select their interacting neighbors during collective responses.”

Point 1.1.3: Also, at line 145: "inspire us to develop the pairwise interaction mechanism" induced me to understand that they propose a new algorithm or principle, whereas they probably meant to say "led us to hypothesise that pairwise interactions..." or something similar.

Response: We thank Reviewer #1 very much for pointing that out and apologize for not making this clear in the previous version of manuscript. Based on the reviewer’s valuable suggestion, in the new version of manuscript, we have revised relevant sentences to avoid misleading expressions (see **main text: page 5, lines 227-229**):

“The empirical findings **in fish schools led us to hypothesize a pairwise interaction mechanism based on BOC, where the focal individual selectively interacts with the neighbor exhibiting the largest BOC, referred to as BOC-based interaction.**”

In addition to addressing the incorrect usage of the article pointed out by the reviewer, we have also made revisions based on feedback from the **Nature Research Editing Service** to address any related errors. For example,

- see **main text: page 1, lines 18-21:**

“Then, we conduct experiments of collective spin and collective turn with **a** real-time physics simulator to investigate the dynamics of information transfer in BOC-based interaction and further validate its effectiveness on 50 real miniature swarm robots.”

- see main text: page 1, lines 27-28:

“Large-scale collective motion is a widely observed phenomenon in biological systems, e.g., flocks of starlings¹, schools of zebrafish², and herds of sheep³.”

- see main text: page 1, lines 32-33:

“Extensive studies have been conducted to reveal interaction mechanisms through computational theories and handcrafted designs.”

- see main text: page 4, lines 183-185:

“the peak of the distribution of the correlation coefficient ρ is close to zero, indicating neither a positive nor a negative correlation between the BOC-based motion salience”

- see main text: page 8, lines 383-385:

“To further validate the feasibility of BOC-based interaction in swarm robotics, we adopted the BOC-based interaction in a swarm consisting of 50 SwarmBang robots^{45,46} (see Methods and Supplementary Sec. 6 for a brief overview of the SwarmBang system).”

- see main text: page 17, lines 755-758:

“Following that, the server computer calculates the desired linear and angular speed of each robot and broadcasts to all the robots via a wireless communication module through a customized communication protocol with a fixed time interval (the details of this protocol can be found in Supplementary Sec. 6.4 and Supplementary Fig. 20).”

In particular, to maintain the clarity of the response letter, we have not presented all of the revisions of incorrect usage of the article “a, an, the”. All unshown revisions have been highlighted in red within the revised manuscript.

Point 1.2: Other misleading writing errors include the use of the verb “emerge” as a transitive verb, such as in the title and at line 144, which makes it hard to understand what the claim is. I guess that the authors meant to say something like “displays, leads to, supports”, or something similar.

Response: We thank Reviewer #1 very much for pointing that out and apologize for the misleading writing errors in the previous manuscript. Based on the reviewer’s valuable suggestion, we have replaced the term “emerge” with “leads to” to accurately convey our claims and rephrased each relevant sentence in the revised manuscript. For example,

- see main text: page 1, lines 1-2:

The title of this work has been changed from “Body Orientation Change of Neighbors Emerges Scale-Free Correlation in Collective Motion” to “**Body Orientation Change of Neighbors Leads to Scale-Free Correlation in Collective Motion**”.

- see main text: page 5, line 226:

The subtitle in **Results** section has been changed from “BOC-based interaction emerges the non-trivial scale-free correlation” to “**BOC-based interaction leads to the non-trivial scale-free correlation**”.

- see main text: page 1, lines 21-23:

“The experimental results show that BOC-based interaction not only facilitates the directional information transfer within the group but also leads to scale-free correlation within the swarm.”

- see main text: page 7, lines 323-325:

“On the contrary, random-based interaction do not lead to scale-free correlation, as the correlation length remains approximately constant with the increasing spatial size of group.”

Point 1.3: Other infelicities include "verify an hypothesis" (hypotheses can only be tested or at best falsified, certainly not verified), "our experiments with robots demonstrate" (demonstrations are best reserved to mathematical proofs; robotic results can show, support, or at best validate something).

Response: We thank Reviewer #1 for these valuable comments and apologize for the infelicities in the previous manuscript. We fully agreed with the reviewer’s comments regarding the distinction between “testing” and “verifying” a hypothesis, as well as the appropriate use of verbs such as “demonstrate” in the context of experimental results. Based on the reviewer’s suggestions, we have revised any incorrect usage of such verbs in the revised manuscript. For example,

- see main text: page 1, line 14:

“To test our hypothesis, we reveal the significant role of BOC during collective U-turn behaviors in fish schools by reconstructing scenes from the view of individual fish.”

- see main text: page 2, lines 60-62:

“To test this hypothesis, we analyzed experimental data from spontaneous U-turn behavior in groups of rummy-nose tetra (*Hemigrammus rhodostomus*)^{16,28,29}, aiming to uncover the significance of BOC in achieving the collective response.”

- see main text: page 5, lines 221-223:

“These additional results showed that the aspect ratio had a negligible effect on the experimental outcomes that confirmed the significant role of BOC during U-turns in fish schools (see Supplementary Fig. 14 for detailed analysis results).”

- see main text: page 7, lines 293-294:

“These results showed that the BOC-based interaction is an effective mechanism for information transfer with great efficiency and scalability.”

- see main text: page 16, lines 738-739:

“To show the advantages of BOC-based interaction in facilitating the emergence of collective responses, we developed a swarm robotic system (i.e., the SwarmBang system).”

- see main text: page 8, line 376-379:

“The simulation results showed that BOC-based interaction facilitates information transfer within the group and leads to non-trivial scale-free correlation, regardless of whether the informed individual is positioned at the center, middle, or border of the group (see Supplementary Sec. 9 for detailed simulation results).”

- see main text: page 9, lines 404-406:

“The experimental results showed that using BOC-based interaction, the swarm with 50 robots effectively responds to the successive heading changes of the informed robot despite the challenges introduced by motion errors and randomness (Fig. 5a).”

- see main text: page 9, lines 429-431:

“Overall, the robotic experiments validated the feasibility and applicability of the BOC-based interaction under realistic and non-ideal conditions and further supported the significant role of body orientation change in the emergence of scale-free correlation.”

In addition to addressing the verb usage errors pointed out by the reviewer, we have also revised unclear expressions based on instructions from the **Nature Research Editing Service** to ensure that our findings are more easily understood by readers. For example,

- see main text: page 3, lines 103-104:

“Moreover, we consider the variation between these two distances as the transient measure of the neighbor’s body orientation change,”

- see main text: pages 12-13, lines 579-581:

“We leverage the number of connections with other nodes, namely, the out-distance (or the out-degree) of nodes in the directed network, to represent the leadership of the focal individual.”

- see main text: page 1, lines 38-40:

“vision-based modeling paradigm has recently become popular in current research 2,17–20 , as these models are more biologically plausible in mimicking how animals perceive and react to their neighbors.”

- see main text: page 1, lines 5-7:

“Recently, vision-based mechanisms, which establish the relationship between visual inputs and motion decisions, have been applied to model and better understand the emergence of collective motion.”

- see main text: page 7, lines 298-299:

“To identify the occurrence of scale-free correlation within the group, we analyzed the relationship between the correlation length and spatial sizes of group.”

In particular, to maintain the clarity of the response letter, we have not presented all of the revisions of writing errors such as the usage of verb. All unshown revisions have been highlighted in red within the revised manuscript.

Point 1.4: Even after reading this second version of the manuscript, I still do not understand what the authors precisely mean by "scale-free correlation": do they mean scale-free behavioural correlation, they way Cavagna et al used this term in their paper on bird flock analysis? It is important to define precisely what correlations they refer to (what is correlated to what) before making statements of scale-free correlation.

Response: We thank Reviewer #1 for these insightful comments and apologize for not making this clear in the previous manuscript. Yes, it is correct that the scale-free correlation mentioned in our study is consistent with the way *Cavagna et al* used in their paper. Their pioneering work provided a solid foundation for understanding the onset of collective response in starling birds from the perspective of scale-free correlation. The computational methods of identifying scale-free correlation provided by *Cavagna et al*, are instrumental in guiding our study to analyzing collective response in artificial and swarm robotics systems.

Indeed, the term “correlation” in scale-free correlation refers to the correlations of velocity fluctuations among individuals in this study. It is because the signature of scale-free correlation in this work is identified by the increasing correlation length with the increasing spatial size of group. This correlation length is estimated through the correlation function of individuals’ velocity fluctuations $C(r) = \frac{\sum_{i,j=1}^N \hat{\mathbf{u}}_i \cdot \hat{\mathbf{u}}_j \delta(r-r_{ij})}{\sum_{i,j=1}^N \delta(r-r_{ij})}$, which represents the average inner product of the velocity fluctuations of all pairs of individuals with mutual distance r . In the definition of the $C(r)$, $\hat{\mathbf{u}}_i = \hat{\mathbf{v}}_i - \frac{1}{N} \sum_{k=1}^N \hat{\mathbf{v}}_k$ represents the velocity fluctuations of the individual i . $\delta(r - r_{ij})$ is a smoothed Dirac function used to select pairs of individuals at the same distance r . r_{ij} is the distance between two individuals (please see **Point 1.5** for a detailed computational process of correlation length based on $C(r)$).

Moreover, we fully agreed the reviewer noted “It is important to define precisely what correlations they refer to (what is correlated to what) before making statements of scale-free correlation”. In the revised manuscript, we have added the following sentences in the **Introduction** and **Results** section to clarify what kinds of correlation we refer to.

- see main text: page 2, lines 75-77:

“Notably, scale-free correlation in this study refers to the correlations among the velocity fluctuations of individuals, which has been widely adopted to understand the onset of collective responses in biological swarms^{31,32}.”

- see main text: page 7, lines 295-298:

“To explore the reason underlying the advantage of BOC-based interaction in information transfer, we hypothesized that it might be the consequence of scale-free correlation, which is a macro-level phenomenon characterized by the correlations of velocity fluctuations among individuals.”

Point 1.5: The authors do make some sort of definition of correlation 187-190, but I failed to understand what they mean by "we calculated the correlation length (what is the correlation length?) of different flock (flock refers to birds, not to fish) sizes. The point where the correlation function (what is the correlation function?) reaches zero, defines the correlation length (so now we have two undefined terms that define each other). The correlation function C_r represents the velocity fluctuations (does C represent a correlation or a fluctuation?) as a functions of distance r in the group (distance between what?)" The text in brackets is my own text to explain how I got rapidly lost by the explanation.

Response: We thank Reviewer #1 for these important comments and apologize for not making this point clear in the previous manuscript. In the following, we first addressed each concerns raised by the reviewer and then provided a detailed definition and computational process of correlation function and correlation length to clarify the concept of correlation in this study.

i. **addressing each concerns raised by the reviewer**

Point 1.5.1: The authors do make some sort of definition of correlation 187-190, but I failed to understand what they mean by "we calculated the correlation length (what is the correlation length?) of different flock (flock refers to birds, not to fish) sizes.

Response: We thank Reviewer #1 for pointing that out and apologize for not making this point clear in the previous manuscript. In fact, correlation length provides a good measure of the spatial span of the correlation domain, which is defined as the zero of the correlation function $C(r)$ (same as the $C(r) = \frac{\sum_{i,j=1}^N \hat{u}_i \cdot \hat{u}_j \delta(r-r_{ij})}{\sum_{i,j=1}^N \delta(r-r_{ij})}$ stated in **Point 1.4**). In brief, based on the mathematical definitions of $C(r) = \frac{\sum_{i,j=1}^N \hat{u}_i \cdot \hat{u}_j \delta(r-r_{ij})}{\sum_{i,j=1}^N \delta(r-r_{ij})}$, the distance r_0 at which the correlation function $C(r_0) = 0$ serves as an estimation of the correlation length within the group. This implies that the velocity fluctuations of the focal individual are no longer correlated with those of its neighbors beyond the distance r_0 .

In addition, we fully agreed the reviewer noted that "flock refers to birds, not to fish". We acknowledged that the "flock size" is inappropriate to represent the spatial size in this study. Hence, we have rephrased the "flock size" to "**spatial size of group**" in the revised manuscript, because the term 'group' in our simulation (or robotic) experiments refers not to bird flocks or fish schools, but to a group of simulated robots (or real robots), we have opted to use the term "spatial size of group" for simplicity expression (see the **Fig.3-5** in main text).

Point 1.5.2: The point where the correlation function (what is the correlation function?) reaches zero, defines the correlation length (so now we have two undefined terms that define each other). The correlation function C_r represents the velocity fluctuations (does C represent a correlation or a fluctuation?) as a functions of distance r in the group (distance between what?)" The text in brackets is my own text to explain how I got rapidly lost by the explanation.

Response: We thank Reviewer #1 for this critical comment and apologize for not making this point clear in the previous manuscript. In fact, correlation function $C(r)$ represents the average directional alignment of the velocity fluctuations of all pairs of individuals in

a mutual distance r , which is calculated as $C(r) = \frac{\sum_{i,j=1}^N \hat{\mathbf{u}}_i \cdot \hat{\mathbf{u}}_j \delta(r-r_{ij})}{\sum_{i,j=1}^N \delta(r-r_{ij})}$. In the correlation function $C(r)$, r serves as the independent variable, representing the distance between two individuals. The correlation function $C(r)$ describes the degree of average directional alignment of the velocity fluctuations as a function of this distance. r is a variable that we vary to investigate how the directional correlation between individual's velocity fluctuations evolves with increasing distance. Particularly, $\hat{\mathbf{u}}_i = \hat{\mathbf{v}}_i - \frac{1}{N} \sum_{k=1}^N \hat{\mathbf{v}}_k$ represents the velocity fluctuations of individual i . $\delta(r - r_{ij})$ is a smoothed Dirac function used to select pairs of individuals at the same distance r . r_{ij} is the distance between two individuals. If the relative distance r_{ij} between two individuals is equal to the variable r , $\delta(r - r_{ij}) = 1$, otherwise, $\delta(r - r_{ij}) = 0$.

ii. detailed computational process of correlation length and revisions in main text

In the following, we have provided a detailed computational process of the correlation function and the definition of correlation length:

To obtain the correlation length, we first calculated the correlation function $C(r) = \frac{\sum_{i,j=1}^N \hat{\mathbf{u}}_i \cdot \hat{\mathbf{u}}_j \delta(r-r_{ij})}{\sum_{i,j=1}^N \delta(r-r_{ij})}$ with the increasing distance r , where $\hat{\mathbf{u}}_i = \hat{\mathbf{v}}_i - \frac{1}{N} \sum_{k=1}^N \hat{\mathbf{v}}_k$ represents the 2D vectors of the velocity fluctuation of individual i . r is a variable that represents the distance between two individuals, which can be varied to investigate how the directional correlation between velocity fluctuations evolves with increasing distance. $\delta(r - r_{ij})$ is a smoothed Dirac function used to select pairs of individuals at the same distance r . r_{ij} is the distance between two individuals. If the relative distance r_{ij} between two individuals is equal to the variable r , $\delta(r - r_{ij}) = 1$, otherwise, $\delta(r - r_{ij}) = 0$.

Specifically, for a given distance r , the correlation function $C(r)$ measures the average directional alignment of velocity fluctuations among individuals at a mutual distance r . For example, as shown in **Fig.R1a**, when two vectors of the velocity fluctuation are aligned in the same direction, their dot product is positive ($\hat{\mathbf{u}}_i \cdot \hat{\mathbf{u}}_j = 1$). As the difference in the direction of velocity fluctuations increases, the dot product keeps decreasing, eventually becoming negative ($\hat{\mathbf{u}}_i \cdot \hat{\mathbf{u}}_j < 0$). Particularly, when two vectors of velocity fluctuation are orthogonal, the dot product of $\hat{\mathbf{u}}_i \cdot \hat{\mathbf{u}}_j = 0$. When two vectors of velocity fluctuation are oppositely directed, we have the $\hat{\mathbf{u}}_i \cdot \hat{\mathbf{u}}_j = -1$. As a result, as shown in **Fig.R1b**, when distance r is small, the velocity fluctuations among individuals are closely aligned, leading to a high value of $C(r) > 0$. However, as the distance r increases, the directions of these velocity fluctuations are gradually diverged from each other, causing $C(r)$ to decrease continuously until it eventually crosses the x-axis $C(r) < 0$. Finally, correlation length r_0 is defined at where the $C(r)$ just crossed the x-axis (marked by the red circle in **Fig.R1b**), i.e., $C(r_0) = 0$.

Moreover, we recognized that the previous inclusion of detailed formulas of the correlation function and correlation length in the Supplementary Information may have made it difficult for readers to fully grasp the concepts of these two important methods. Hence, in the revised manuscript, we have rephrased the descriptions of the correlation function and correlation length in **Results** section before we referred, while providing the full mathematical definitions in **Methods** section under the subsection titled "**Correlation function and correlation length**", with the intention of minimizing the use of formulas in the **Results** section to improve

the overall readability of the manuscript (also attempt to align with valuable suggestions of Reviewer #1 in the first round review).

- see main text: page 7, lines 298-315:

“To identify the occurrence of scale-free correlation within the group, we analyzed the relationship between the correlation length and spatial sizes of group. The correlation length indicates the distance at which velocity fluctuations among individuals are no longer aligned, providing an estimate of the spatial range over which the influence of one individual on another diminishes. To obtain the correlation length, we first calculated the correlation function of the velocity fluctuation $^3C(r)$, which measures the average directional alignment of velocity fluctuations among individuals at a distance r (see Methods for a detailed mathematical definition of $C(r)$). In particular, r is a variable that represents the distance between two individuals, which can be varied to investigate how the directional correlation between velocity fluctuations evolves with increasing distance. Specifically, when distance r is small, the velocity fluctuations among individuals are closely aligned, leading to a high value of $C(r) > 0$. However, as the distance r increases, the directions of these vectors gradually diverge, causing $C(r)$ to continuously decrease until it eventually crosses the x-axis $C(r) < 0$.

The correlation length is then defined as the distance r_0 at which the correlation function $C(r)$ first reaches zeros, i.e., $C(r_0) = 0$ (see Methods for the details of the correlation length calculation or Supplementary Fig. 33). For example, we illustrated the correlation function $C(r)$ varies with the increasing distance r for BOC and random-based interaction in Fig. 3d and Fig. 3h, respectively.”

- see main text: pages 14-15, lines 653-672:

“To obtain the correlation length within a group, we first calculated the correlation function $C(r)$ following the computational method provided by Cavagna et al^{31,32}. Specifically, the correlation function $C(r)$ is used to evaluate the average degree of directional alignment among individuals’ velocity fluctuations at a distance r . The correlation function can be calculated as follows:

$$C(r) = \frac{\sum_{i,j=1}^N \hat{\mathbf{u}}_i \cdot \hat{\mathbf{u}}_j \delta(r - r_{ij})}{\sum_{i,j=1}^N \delta(r - r_{ij})}$$

where the $\hat{\mathbf{u}}_i = \hat{\mathbf{v}}_i - \frac{1}{N} \sum_{k=1}^N \hat{\mathbf{v}}_k$ is the velocity fluctuation of individual i . $\delta(r - r_{ij})$ is a smoothed Dirac δ function that is used to select pairs of individuals at the same distance r . r_{ij} is the distance between two individuals. If the relative distance r_{ij} between two individuals is equal to r , we have $\delta(r - r_{ij}) = 1$. Otherwise, $\delta(r - r_{ij}) = 0$. High values of $C(r) > 0$ indicate the strong correlation in the velocity fluctuations among all individuals at a certain distance r . As r increases, the value of $C(r)$ steadily decreases and eventually crosses the x-axis ($C(r) < 0$), indicating that correlation in the velocity fluctuations among individuals exhibits the decaying transition from a strong correlation to a weak correlation (see Supplementary Fig. 33 for the typical curves of $C(r)$ or Fig. 3d,h). In particular, individuals who failed to respond to the spin/turn initiator were excluded from the calculations.

The correlation length is then defined as the distance r_0 at which the correlation function becomes zero $C(r_0) = 0$. At a distance r_0 , the dot product of the velocity fluctuations among individuals $\hat{\mathbf{u}}_i \cdot \hat{\mathbf{u}}_j = 0$, indicating that individuals no longer interact with each other beyond the distance r_0

from the macro-level perspective. The definition of the correlation function and a diagram of the correlation length can be found in Supplementary Fig. 33.”

Point 1.6: There are several other cases where the authors discuss important factors without first defining them or without defining them at all. For example, the use of the correlation matrix is described before the definition of how the matrix is constructed (it is possible to understand it from Figure 1, but that figure is referred to only later in the text. Another example: they refer to spin behaviour and spin results, but never define what such behaviour is (at line 160 they point to Figure 2a, but I could not spot any spinning behaviour --whatever that is supposed to be). Other examples: line 117 "absence of frontal preference ($\alpha = 0$)" without defining what frontal preference and α are; line 163: "Additionally the polarization sharply decreased" without explaining what the polarization is; line 175 "the curve formed by the transfer distance d_{\max} " without ever explaining what that is.

Response: We thank Reviewer #1 very much for carefully reviewing and apologize for not clearly defining these important factors in the previous version of the manuscript. We greatly appreciate the reviewer for giving us the opportunity to clarify these important concepts. In addition to the issues raised by the reviewer, we have carefully reviewed the manuscript and added detailed descriptions of any important factors or methods that were not clearly defined in the revised manuscript. In the following, we first revised the issues raised by the reviewer point by point and then provided details on the additional revisions made.

Point 1.6.1: For example, the use of the correlation matrix is described before the definition of how the matrix is constructed (it is possible to understand it from Figure 1, but that figure is referred to only later in the text.

Response: We thank Reviewer #1 for this significant comment and apologize for not making this clear in the previous manuscript. For the analysis of experimental data from collective U-turn behaviors observed in fish schools, we first constructed a BOC matrix and leader-follower network within a period of U-turn trajectory. Then, we derived two vectors of BOC-based motion salience and leadership of individuals based on the BOC matrix and leader-follower network, respectively. Finally, we conduct Spearman correlation analysis between these two vectors to reveal the significant role of BOC in collective U-turn behaviors. In the following, we have provided the detailed calculation process of the BOC matrix and leader-follower network, respectively.

i. Construction of BOC matrix and derivation of BOC-based motion salience in experimental data analysis of the U-turn behaviors in fish schools

For a given period of U-turn trajectory $[T - \tau, T]$, the entries in the BOC matrix represent the magnitude of each neighbor's BOC as perceived by the focal individual within $[T - \tau, T]$. In particular, a BOC matrix is denoted as $G(T, \tau) = [g_{ij}(T, \tau)]_{N \times N}$, where $g_{ij}(T, \tau)$ is the body orientation change of neighbor j observed from the focal individual i within a period of $[T - \tau, T]$ (defined in the Eq. (1) of the main text, see **pages 2-3, lines 82-130**). The row of a BOC matrix represents the focal individuals (or the viewer), and the column is their perceived neighbors.

Specifically, to get a BOC matrix from a period of U-turn trajectory $[T - \tau, T]$, we first reconstructed the visual field of each individual and then calculated the neighbor's BOC from the perspective of each individual based on Eq. (1) in main text. For example, as

shown in **Fig.R2a-d**, given a period trajectory of U-turn behavior from $T = 2.22s$ to $T = 2.82s$ (**Fig.R2a**), we reconstructed the visual field of individual 8 (**Fig.R2b-c**) and calculated the magnitude of its neighbors' BOC (**Fig.R2d**). As a result, we obtained the eighth column of a BOC matrix (marked by red rectangular shown in **Fig.R2e**). After obtaining the neighbors' BOC of each individual, we could obtain a complete BOC matrix (**Fig.R2e**).

Next, we calculated the column-wise average to get the BOC-based motion salience of each individual $G_i(T, \tau)$ (**Fig.R2f**). From a mathematical standpoint, the BOC-based motion salience is a vector $[G_1(T, \tau), \dots, G_i(T, \tau)]_{i \in [1, \dots, N]}$ that contains how noticeable the movements of the focal individual i are perceived by other individuals in the group based on the BOC. Additionally, we normalized the BOC-based motion salience by dividing with the maximum value in BOC-based motion salience to ensure the analysis of real fish data is not skewed by differences in the data scales (**Fig.R2g**).

ii. Reconstruction of the leader-follower network to quantify the leadership of individuals in experimental data analysis of the U-turn behaviors in fish schools

In brief, the leader-follower network is essentially an unweighted directed graph, where connections flow from leaders at higher hierarchy to followers at lower. Consequently, an individual with more outgoing connections with followers lower in the hierarchy is considered to have stronger leadership within the group.

To quantitatively evaluate the leadership of each individual during collective U-turns, we first reconstructed the leader-follower network in fish schools. To do that, we first derived the leader-follower relationship of all pairs of individuals based on the directional alignment function $\xi_{ij}(\lambda_t)$. For a U-turn trajectory from T_b to T_f , the directional alignment function computes the temporal degree of velocity alignment between the moving directions of the focal individual i and its neighbors in the form of,

$$\xi_{ij}(\lambda_t) = \langle \hat{\mathbf{v}}_i(t) \cdot \hat{\mathbf{v}}_j(t + \lambda_t) \rangle,$$

where t is the time index that varies between T_b and T_f . λ_t is the time lag, which ranges from $[-(T_f - T_b), (T_f - T_b)]$. The operation $\langle \cdot \rangle$ denotes the average function over all the time index t . As the increase of time lag λ_t , we could obtain the curve of ξ_{ij} as a function of λ_t . In particular, we denoted λ_{ij}^* as the value at which the curve of ξ_{ij} reaches its maximum, indicating individual j achieves the highest degree of directional consensus with individual i with a time lag or ahead of $|\lambda_{ij}^*|$ seconds. The sign of λ_{ij}^* determines the leader or follower role of individual j . $\lambda_{ij}^* > 0$ means that the movement of individual j lags behind that of individual i with $|\lambda_{ij}^*|$ seconds, suggesting that the individual j is a follower to individual i . Conversely, $\lambda_{ij}^* < 0$ suggests that individual j leads individual i by $|\lambda_{ij}^*|$ seconds. For example, as shown in **Fig.R3**, for the trajectory of a pair of individual 1 and individual 3 from $T = 2.22s$ to $T = 2.82s$ (**Fig.R3a**), we could obtain the curve of directional alignment function $\xi_{1,3}$ as a function of λ_t (**Fig.R3b**). From the **Fig.R3b**, we found the $\xi_{1,3}(\lambda_t)$ reaches the maximum at $\lambda_{1,3}^* = 0.46s$, which means that individual 3 is the follower to the individual 1 with the lag of $0.46s$. Additionally, as shown in **Fig.R3a**, individual 1 indeed initiate its turn earlier than individual 3.

Moreover, based on the above calculation process, we could obtain the leader-follower relationship for each pair of individuals and reconstruct the leader-follower network within a period of time (**Fig.R3c**). After obtaining the leader-follower network, we could quantify leadership by calculating the normalized out-degree of each node (**Fig.R3d**) based on Eq. (3) in main text. Specifically, the normalized out-distance signifies the hierarchical layer of each node (or individual) in the leader-follower network. In such a hierarchical network, some individuals act as leaders, whereas others act as followers, with directed connections between individuals, i.e., leaders point to their followers. We leverage the number of connections with other nodes, namely, the out-distance (or the out-degree) of nodes in the directed network, to represent the leadership of the focal individual. A higher out-degree indicates that the individual has stronger leadership within the group, as it has more connections with followers.

Particularly, we acknowledged that the mathematical notation of directional alignment function in reconstructing leader-follower network is similar to the correlation function $C(r)$ in the previous version of manuscript. Hence, we have rephrased the mathematical notation of the directional alignment function as $\xi_{ij}(\lambda_t)$ to avoid misleading between these two key concepts.

Additionally, based on the reviewer's valuable suggestions, we have rephrased relevant sentences in **Results** section to clarify the construction of the BOC matrix and derivation of BOC-based motion salience before we referred to (see **main text: page 4, lines 153-162**):

“For each segment $[T - \tau, T]$ of a U-turn trajectory, we quantified the BOC of each individual's neighbor from the first-person perspective according to Eq. (1). This further yielded a BOC matrix in which the rows represent the focal individuals (or viewers) and the columns represent their perceived neighbors. The entries in the BOC matrix represent the magnitude of each neighbor's BOC as perceived by the focal individual within a segment $[T - \tau, T]$ (see Methods for the mathematical definition of the BOC matrix). Then, to evaluate how prominently the focal individual's movements are perceived by its neighbors, we averaged each column in the BOC matrix, resulting in a vector known as the BOC-based motion salience that encapsulates the quantification of each individual's degree of salient movement from the perspective of others within $[T - \tau, T]$ (see Methods for more detailed information of BOC-based motion salience).”

Besides that, based on the reviewer's valuable suggestions, we have included **Fig.R2** and **Fig.R3** in Supplementary information (see **SI: pages 44-45**) to better clarify the computational process and added the following sentences in **Methods** section to facilitate the computational process of BOC matrix (see **main text: page 12, lines 533-548**):

“To obtain the BOC-based motion salience of each individual, we first calculated the BOC of the neighbors from each individual's view according to the definition of Eq. (1) over a period of $[T - \tau, T]$, which forms a BOC matrix denoted as $G(T, \tau) = [g_{ij}(T, \tau)]_{N \times N}$. In a BOC matrix, the rows correspond to the focal individuals (or viewers), whereas the columns represent their respective neighbors. Consequently, we can derive the motion salience of each individual $G_i(T, \tau)$ by averaging each column in BOC matrix $G(T, \tau)$ (Fig. 2d), i.e., the BOC-based motion salience. From a mathematical standpoint, the BOC-based motion salience is a vector $[G_1(T, \tau), \dots, G_i(T, \tau)]_{i \in [1, \dots, N]}$, which contains how noticeable the movements of the focal individual i are perceived by other individuals in the group based on the BOC. Individuals with larger $G_i(T, \tau)$ exhibit more salient directional changes in their movements during $[T - \tau, T]$. Conversely, a lower $G_i(T, \tau)$ suggests more stable movements with fewer directional changes in the

group. In particular, the BOC-based motion salience is normalized by the maximum $G_i(T, \tau)$ value in the group to ensure that the Spearman correlation analysis results are not skewed by differences in the data scales (see Supplementary Fig. 31f-g).”

Moreover, we have added the following sentences to briefly define the leader-follower network and clarify the concept of leadership in this study before we used it in **Results** section (see **main text: page 4, lines 163-169**):

“Next, we quantified the leadership of each individual based on the leader-follower network derived from a segment $[T - \tau, T]$ of the U-turn trajectory. In brief, the leader-follower network is essentially an unweighted directed graph, where connections flow from leaders at higher hierarchy to followers at lower (see Methods for the detailed reconstruction of the leader-follower network). Consequently, an individual with more outgoing connections with followers lower in the hierarchy is considered to have stronger leadership within the group (see Methods for the detailed definition of leadership).”

Furthermore, we have rephrased relevant sentences in **Methods** section to better clarify the definition of leader-follower network and the leadership.

- see **main text: page 12, lines 577-579**:

“In such a hierarchical network, some individuals act as leaders, **whereas others act as followers**, with directed connections between individuals, i.e., leaders point to their followers.”

- see **main text: pages 12-13, lines 579-580**:

“We leverage the number of connections with other nodes, namely, the out-distance (or the out-degree) of nodes in the directed network, to represent the leadership of the focal individual.”

- see **main text: page 13, lines 588-589**:

“Conversely, an individual with a lower $L_i(T, \tau)$ is at a lower hierarchy in the leader-follower network.”

Point 1.6.2: Another example: they refer to spin behaviour and spin results, but never define what such behaviour is (at line 160 they point to Figure 2a, but I could not spot any spinning behaviour --whatever that is supposed to be).

Response: We thank Reviewer #1 for this constructive comment and apologize for not making this clear in the previous manuscript. In fact, the spinning state (or behavior) refers to the in-place rotational movement (shown in **Fig.R4**). Since we used two-wheeled simulated robots in our simulation experiments, the spinning behavior is achieved by controlling simulated robots with a certain angular speed while maintaining a linear speed of zero.

Additionally, in **Fig.2a** of the previous manuscript, we presented the snapshots of collective spin of $N = 200$ when the spin initiator had just spun 2π . Based on the valuable feedback from the reviewer, we acknowledged that these snapshots are unclear in illustrating the collective spin. On the one hand, the absence of marked headings for each individual makes it difficult to determine the orientation of elliptical individuals, which in turn obscures the understanding of the collective spin phenomenon. On the other hand, providing only one single snapshot further impairs the understanding of collective spin. Hence, in the revised manuscript, we have added arrows to each elliptical individual to indicate their current heading,

aiming to enhance the clarity of snapshots. Also, we have provided consecutive snapshots from the activation time of the spin initiator to the time when the initiator had just finished the rotation of 2π to facilitate the understanding of collective spin (see **Fig.3a,e** in main text or **Fig.R5** for snapshots of collective spin).

In the following, we have provided a more detailed description of collective spin and spinning state (or behavior). Specifically, collective spin is an idealized and simplified scenario designed to study the information transfer of BOC-based interaction. In the collective spin, individuals can adopt one of two motion states (or behavior), i.e., either spinning or remaining stationary, and update their motion states based on the states of their selected neighbors, where the spin refers to an in-place rotational movement and the stationary means an individual halts all movement. A spin initiator begins the rotational movement at a predetermined time, serving as the trigger for propagating the spinning state (or behavior) within the group. When individuals adapt their motion states (or behavior) to either spin or stay stationary, it enables the spinning state to disseminate within the group. Importantly, the goal of the collective spin is to achieve in-place rotation collectively throughout the group before the spin initiator completes a 2π rotation. Specifically, the simulation experiments of collective spin have three important aspects: 1) Motion states (or behaviors) of simulated robots; 2) Initiation of collective spin by a spin initiator; 3) Reaction rule of individuals. In the following, we provided the detailed descriptions of these three aspects:

i. Motion States (or behaviors) of simulated robots

In the simulation experiments of collective spin, each simulated robot has two motion states: 1) stationary state; and 2) spinning state (or behavior). For the stationary state, it means that both the linear and angular speeds of simulated robots are set to 0, suggesting that the simulated robot remains in a stationary position and heading. On the contrary, the spinning state (or behavior) indicates that the angular speed of simulated robots is at its maximum value (denoted as ω_{max}^{sim}), but the linear speed is still set to 0, meaning the simulated robot rotates in place.

ii. Initiation of collective spin by a spin initiator

The collective spin is initiated by one informed individual (or spin initiator) selected at the forefront of the group. Specifically, the informed individual starts spinning at a preset activation time and stops spinning after completing a 2π rotation, which serves as a short time stimulus within the group. In other words, simulated robots are required to swiftly transfer the spinning state throughout the entire group within a limited time, i.e., before the spin initiator completes its 2π rotation.

iii. Reaction rule of individuals in collective spin

In the collective spin simulation experiments, the reaction rule of an individual is to switch between the stationary and spinning states (or behavior) based on the state of the selected neighbor. For example, in the context of BOC-based interaction, each simulated robot determines the BOC of its neighbors within a perception range R_{visual} and selects the neighbor with the maximum BOC to react. If the selected neighbor is in a spinning state, the focal individual also enters the spinning state and starts to spin (or rotate) in place with maximal angular speed; if the selected neighbor is stationary, the focal individual switches to the stationary state and stops spinning (or rotating). The velocity of the simulated robot i is calculated in the form of,

$$\hat{\mathbf{v}}_i(t + 1) = \mathbf{R}(\theta_i^{\text{rot}})\hat{\mathbf{v}}_i(t),$$

where the $\mathbf{R}(\theta_i^{\text{rot}})$ is the rotation matrix that characterizes the rotational motion of individuals, which is defined as $\mathbf{R}(\theta_i^{\text{rot}}) = \begin{pmatrix} \cos(\theta_i^{\text{rot}}) & -\sin(\theta_i^{\text{rot}}) \\ \sin(\theta_i^{\text{rot}}) & \cos(\theta_i^{\text{rot}}) \end{pmatrix}$. The rotation angle $\theta_i^{\text{rot}} = \omega_i \Delta t$ corresponds to the angular displacement of individual i during time interval Δt . The angular speed ω_i is determined by the state of the selected neighbor. If the selected neighbor is stationary, then $\omega_i = 0$ and $\theta_i^{\text{rot}} = 0$, indicating the focal individual i ceases spinning. Conversely, if the selected neighbor is in the spinning state, the $\omega_i = \omega_{\text{max}}^{\text{sim}}$, signifying that individual i is about to start rotational movement in place.

Based on the reviewer's important suggestion, we have added the following sentences in **Results** section to better clarify the meaning of the spinning state (or behavior) and stationary state before we refer (see **main text: page 6, lines 239-250**):

“On the one hand, collective spin is an idealized and simplified scenario designed to study information transfer of BOC-based interaction. **In the collective spin, individuals can adopt one of two motion states (or behavior), i.e., either spinning or remaining stationary, and update their motion states based on the states of their selected neighbors, where the spin refers to an in-place rotational movement and the stationary means an individual halts all movement. A spin initiator begins the rotational movement at a predetermined time, serving as the trigger for propagating the spinning state (or behavior) within the group. When individuals adapt their motion states (or behavior) to either spin or stay stationary, it enables the spinning state to disseminate within the group. Importantly, the goal of the collective spin is to achieve in-place rotation collectively throughout the group before the spin initiator completes a 2π rotation (see Methods for detailed descriptions of collective spin experiments).**”

Additionally, we have rephrased the relevant sentences in **Methods** section to better clarify the experimental setup of collective spin and spinning state (or behavior) (see **main text: page 11, lines 499-508**):

“For the **collective spin simulation experiments**, we assumed that each simulated robot has two motion states (or behavior): stationary and spinning. In the stationary state, both the linear and angular speeds of the simulated robots are set to 0. In the spinning state, the angular speed is set to the maximum $\omega_{\text{max}}^{\text{sim}} = 0.83 \text{ rad s}^{-1}$, whereas the linear speed remains 0, **indicating the in-place rotational movement of the robots. The propagation of the spinning state (or behavior) is triggered by an informed individual selected at the front of the group, who starts spinning at a preset activation time (the 25th step in our simulation) and stops spinning after completing a 2π rotation. The simulated robots are required to swiftly transfer the spinning state throughout the entire group within a limited time, i.e., before the spin initiator completes its 2π rotation.**”

Furthermore, after a thorough review of the manuscript, we realized that the description of the collective turn experiments was also not entirely clear and might lead to misunderstandings. In the revised manuscript, we have rephrased the relevant sections in the **Results** and **Methods** to better clarify the experimental setup of collective turn,

- see **main text: page 6, lines 250-256**:

“On the other hand, collective turn represents a more complex and realistic scenario, in which individuals are required to adjust their movement in response to a sudden directional change initiated by one informed individual (or turn initiator) through velocity alignment. Unlike the stationary nature of the collective spin, the collective turn involves the actual movements of individuals and demands rapid reorientation as the group shifts moving direction, posing a more challenging task of information propagation through the BOC-based interaction (see Methods for more detailed information of collective turn simulation experiments).”

- see main text: page 11, lines 518-526:

“For the collective turn simulation experiments, the simulated robots are required to change their direction of movement in response to a sudden turn initiated by one informed individual (or turn initiator). The informed individual, positioned at the forefront of the group, turns with an angle θ_{info} relative to the group’s movement direction at a preset moment (the 50th step in our simulation). The new direction of the informed individuals then triggers the group to start a collective turn. The reaction rule of an individual in the collective turn simulation experiments is a commonly used velocity alignment rule⁴⁵, i.e., average the headings of the selected neighbor and the robot’s own heading (Eq. (7)). For example, in the context of BOC-based interaction, each robot adjusts its movement to align with the neighbor that exhibits the largest BOC.”

Point 1.6.3: Other examples: line 117 "absence of frontal preference (alpha = 0)" without defining what frontal preference and alpha are;

Response: We thank Reviewer #1 for pointing that out and apologize for not making this clear in the previous manuscript. In fact, frontal preference refers to an individual's tendency to interact (or maintain alignment) with its neighbors primarily within its frontal field of view. In the context of perceiving BOC, frontal preference means that the focal individual is more attentive to the body orientation changes of neighbors positioned in front. In this study, to emulate the fact of frontal preference in biological perception, we involve a frontal weighting factor $fp(t) = \left(\frac{1 + \hat{v}_i(t) \cdot \hat{x}_{ij}(t)}{2}\right)^\alpha$ in Eq. (1) of the main text to characterize the anisotropic visual perception of BOC in the form of,

$$\begin{cases} g_{ij}(T, \tau) = \sum_{t=T-\tau+1}^T \frac{|\beta_j(t) - \beta_j(t-1)|}{\Delta t} \cdot fp(t) \\ fp(t) = \left(\frac{1 + \hat{v}_i(t) \cdot \hat{x}_{ij}(t)}{2}\right)^\alpha \end{cases},$$

where $g_{ij}(T, \tau)$ is the body orientation change of neighbor j observed by the focal individual i within a period of $[T - \tau, T]$. Δt is the time interval between two steps $t - 1$ and t . $\frac{|\beta_j(t) - \beta_j(t-1)|}{\Delta t}$ represents the transient measure of the body orientation change of neighbor j . τ represents the time period over which individual i considers the past body orientation change of neighbor j . $fp(t)$ is a frontal weighting factor that characterizes the frontal preference of individuals. The anisotropic parameter α tunes the degree of frontal preference on the neighbor- j .

In the following, we provided the detailed definition and descriptions of $fp(t)$ and explained how the anisotropic parameter α adjusts the degree of frontal preference:

i. Detailed definition of frontal preference of individuals

The $\frac{1+\hat{v}_i(t)\cdot\hat{x}_{ij}(t)}{2}$ quantifies the frontal preference on perceiving neighbor- j 's movement from the first-person perspective of the focal individual i , ranging in $[0, 1]$. The dot product of these two vectors $\hat{v}_i \cdot \hat{x}_{ij}$ measures relative bearing of neighbor- j . It simulates that as the first-person view of neighbor- j moves from the front to the back, the visual perception of focal individual i to the neighbor- j diminishes. For example, as shown in **Fig.R6a**, if $\angle(\hat{v}_i, \hat{x}_{ij}) = 0$, i.e., the dot product of $\hat{v}_i \cdot \hat{x}_{ij} = 1$, it signifies that neighbor- j_1 locates directly in front of the focal individual- i , and the focal individual could perceive the neighbor- j_1 since the $(1 + \hat{v}_i \cdot \hat{x}_{ij})/2$ is 1 (neighbor- j_1 shown in **Fig.R6a**). Once the neighbor- j gradually moves from the front to back with respect to the focal individual i , i.e., $\angle(\hat{v}_i, \hat{x}_{ij})$ goes from 0 to π or $-\pi$, the visual perception to the neighbor- j could diminish since $(1 + \hat{v}_i \cdot \hat{x}_{ij})/2$ decreases from 1 to 0 (neighbor j_2, j_3, j_4 in **Fig.R6a**). By incorporating the $\frac{1+\hat{v}_i(t)\cdot\hat{x}_{ij}(t)}{2}$ with the perception of BOC, we could emulate the anisotropic perception of the neighbor's body orientation change (Eq. (1) in the main text).

ii. Effect of α on tuning frontal preference of individuals

To tune the effect of frontal preference of individuals, we involve the α as an anisotropic parameter in $fp(t)$, denoted as $fp(t) = \left(\frac{1+\hat{v}_i(t)\cdot\hat{x}_{ij}(t)}{2}\right)^\alpha$. When $\alpha = 0$, the focal individual i ignores the forward-oriented perception as the $fp(t)$ always equal to 1 (the green line shown in **Fig.R6b**), which means that the perception of BOC is not relevant to the relative positions of neighbors. In particular, increasing α ($\alpha \geq 1$) make Eq. (1) in main text enhances the anisotropic effect of the visual perception, which causes individuals to gradually narrow the perception of nearby neighbors toward their frontal vision. For example, as shown in **Fig.R6b**, if $\alpha = 10$, it means that the perception of BOC $g_{ij}(T, \tau) \approx 0$ when the relative positions of neighbors are out of the visual sight ($-\pi/2, \pi/2$).

Although the mathematical definitions of these key concepts (e.g., BOC and the effect of frontal preference) were provided in **Methods** section of the previous manuscript, we have recognized that such writing style might have hindered the clarity and accessibility of these definitions in this study. Based on the valuable feedback from the reviewer, we have decided to incorporate these definitions into a new subsection titled “**Quantifying Body Orientation Change to Measure Motion Salience of Neighbors**” at the beginning of the **Results** section. This revision is motivated by two key reasons. On the one hand, the BOC is a central concept proposed in this study to characterize the salient movements of neighbors. On the other hand, it further strengthens the logical flow in **Results** section, from the key definitions of modeling BOC to the crucial role of BOC in the biological swarm, to the investigation of BOC through simulation experiments and to further validation of BOC with real robots. In this new subsection, we provided a detailed definition of body orientation change (BOC), frontal preference and the effect of anisotropic parameter α (see **main text: pages 2-3, lines 81-130**):

“Previous research has revealed that the directional decisions of individuals in biological swarms are influenced by neighbors with salient movement changes^{33–35}. However, the way individuals assess the salience of a neighbor's movements from the first-person visual perception remains not fully elucidated. Here, we introduce a direct observational visual cue, i.e., body orientation change (BOC) to measure the motion salience of neighbors from an individual's own view.

To quantify the BOC of neighbors, we first characterize each individual as a non-transparent ellipse with lengths of major axis a and minor axis b (Fig. 1a), which suggests that neighbors near the focal individual may occlude neighbors that are further away (detailed approaches for identifying the occluded neighbor are presented in Supplementary Sec. 4 and Supplementary Fig. 15). Then, we reconstruct the visual field of each elliptical individual by using the same computational method in the ref³⁶ (see Supplementary Sec. 4 for the detailed computation method). Specifically, the visual field of each individual is defined as the triangular region formed by connecting the focal fish's eye to two points where the rays emanating from its eye are tangent to the edges of each elliptical neighbor. For example, as shown in Fig. 1a, the tangent points on the elliptical neighbor j observed from the focal individual's eye at times $t - 1$ and t are denoted as $[p_1 = (x_1, y_1), p_2 = (x_2, y_2)]$ and $[p_3 = (x_3, y_3), p_4 = (x_4, y_4)]$, respectively. As a result, the visual fields of individual i observing neighbor j at times $t - 1$ and t are the regions shaded in yellow and orange, respectively. Next, we calculate the distance between pairs of tangent points at times $t - 1$ and t , denoted as $\beta_j(t - 1) = \sqrt{(x_1 - x_2)^2 + (y_1 - y_2)^2}$ and $\beta_j(t) = \sqrt{(x_3 - x_4)^2 + (y_3 - y_4)^2}$.

Moreover, we consider the variation between these two distances as the transient measure of the neighbor's body orientation change, i.e., $|\beta_j(t) - \beta_j(t - 1)|$. Finally, we define the accumulation of these transient variations over a period from $T - \tau$ to T as the magnitude of BOC at time T , denoted as $g_{ij}(T, \tau)$. The BOC of neighbor j observed by the focal individual i at time T can be calculated as follows:

$$\begin{cases} g_{ij}(T, \tau) = \sum_{t=T-\tau+1}^T \frac{|\beta_j(t) - \beta_j(t - 1)|}{\Delta t} \cdot \text{fp}(t) \\ \text{fp}(t) = \left(\frac{1 + \hat{\mathbf{v}}_i(t) \cdot \hat{\mathbf{x}}_{ij}(t)}{2} \right)^\alpha \end{cases}$$

where t is the time index and Δt is the time interval between steps $t - 1$ and t . τ represents the time period (or duration) over which individual i considers the past BOC of neighbor j . The accumulation of transient BOC over τ simulates the process by which sensory information is often gathered over a time period before being updated, aligning with findings in biological swarms^{37,38}.

The term $\text{fp}(t) = \left(\frac{1 + \hat{\mathbf{v}}_i(t) \cdot \hat{\mathbf{x}}_{ij}(t)}{2} \right)^\alpha$ in Eq. (1) quantifies the frontal preference of individual i to perceive the movement of neighbor- j from the first-person perspective, and the $\text{fp}(t)$ ranges from $[0, 1]$. $\hat{\mathbf{v}}_i(t)$ is the velocity of individual i and $\hat{\mathbf{x}}_{ij}(t)$ is the relative position vector between individual i and j at time t . Here, frontal preference refers to an individual's tendency to interact (or maintain alignment) with its neighbors primarily within its frontal field of view. In the context of perceiving BOC, the frontal preference causes an individual to be more attentive to the BOC of neighbors positioned in front. Specifically, the dot product of $\hat{\mathbf{v}}_i(t) \cdot \hat{\mathbf{x}}_{ij}(t)$ measures the relative bearing of neighbor j . For example, for the neighbor- j_1 shown in Fig. 1b, $\angle(\hat{\mathbf{v}}_i, \hat{\mathbf{x}}_{ij}) = 0$ signifies that neighbor- j_1 is located directly in front of focal individual i , and the focal individual i could perceive the neighbor- j_1 since the $\frac{1 + \hat{\mathbf{v}}_i(t) \cdot \hat{\mathbf{x}}_{ij}(t)}{2}$ is 1. Once the neighbor j moves from the front to back, i.e., $\angle(\hat{\mathbf{v}}_i, \hat{\mathbf{x}}_{ij})$ goes from 0 to π or $-\pi$ (neighbor- j_2, j_3, j_4 , as shown in Fig. 1b), the ability of the focal individual i to perceive neighbor j is reduced as the $\text{fp}(t)$ decreases from 1 to 0.

Additionally, α is an anisotropic parameter that tunes the effect of the forward-oriented preference on the perception of BOC. Specifically, when $\alpha = 0$, the focal individual i ignores its forward-oriented preference in perceiving BOC. As α increases ($\alpha \geq 1$), individuals gradually narrow their perception of nearby neighbors towards the front of their visual field. For example, as shown in Fig. 1b-c, if $\alpha = 10$, the perception of BOC decreases to $g_{ij}(T, \tau) \approx 0$ when the relative positions of the neighbors are out of the visual sight ($-\pi/2, \pi/2$)."

Point 1.6.4: line 163: "Additionally the polarization sharply decreased" without explaining what the polarization is;

Response: We thank Reviewer #1 for this important comment and apologize for not making this clear in the previous manuscript. In fact, the polarization evaluates the degree of velocity consensus within the group, which is defined as follows:

$$\phi = \frac{1}{N} \left\| \sum_{i=1}^N \frac{\hat{v}_i}{\|\hat{v}_i\|} \right\|,$$

where N is the group size and \hat{v}_i is the velocity of individual i . The polarization ϕ ranges from $[0, 1]$. The higher ϕ indicated the high-level velocity alignment among individuals. Conversely, the lower ϕ represents a diffused group, reflecting the lack of alignment in moving direction. As the mathematical definition of polarization is well-established in the study of collective behavior and not originally proposed in our work, we have opted to include the detailed computational process of ϕ in Supplementary Information (see **SI: page 5, lines 129-134**). In the revised manuscript, to clarify the definition of polarization, we have introduced a brief descriptions of polarization ϕ in **Results** section before using it (see **main text: page 6, lines 266-269**):

“In brief, the polarization ϕ is calculated as the normalized sum of the unit velocity vectors of all individuals in the group, with values ranging from 0 to 1. High polarization indicates a strong alignment of velocities among individuals, whereas low polarization indicates a dispersed group with weaker directional alignment.”

Point 1.6.5: line 175 "the curve formed by the transfer distance d_{\max} " without ever explaining what that is.

Response: We thank Reviewer #1 for pointing that out and apologize for not making this clear in the previous manuscript. We acknowledged that the mathematical definition of information transfer distance is ambiguous and has misleading descriptions for both simulation experiments of collective spin and collective turn. Indeed, due to different reaction rules in simulation experiments of collective spin and collective turn, we defined the information transfer distance differently for these two kinds of experiments. In the revised manuscript, to better distinguish the definition of information transfer distance in these two simulation experiments, we have relabeled the information transfer distance as d_i^{spin} for collective spin and d_i^{turn} for collective turn, respectively. Additionally, we have revised the descriptions of the computational process and redefined the mathematical formulations to clarify the computational process of d_i^{spin} and d_i^{turn} in the new version of manuscript. In the following, we provided detailed definitions of information transfer distance d_i^{spin} and d_i^{turn} in collective spin and collective turn, respectively.

i. Definition of information transfer distance d_i^{spin} in simulation experiments of collective spin

To calculate the information transfer distance d_i^{spin} in collective spin simulation experiments, we started with the definition of spinning lag and spinning rank s_i . Specifically, the spinning lag is defined as the time delay that an individual starts spinning lags behind the spin initiator. Then, the spinning rank s_i is determined based on the value

of spinning lag for each individual, where the longer lag corresponds to the higher spinning rank. For example, the spin initiator has the spinning rank of $s_i = 1$, while the last individual who starts spinning has $s_i = N$. N is the group size. Finally, the information transfer distance d_i^{spin} of individual i is defined as the distance over which information has traveled to reach the individual with spinning rank s_i . d_i^{spin} is estimated as the relative Euclidean distance between the individual i and the spin initiator in the form of,

$$d_i^{\text{spin}} = \|\hat{\mathbf{x}}_i - \hat{\mathbf{x}}_{\text{spin}}^{\text{ini}}\|,$$

where the $\hat{\mathbf{x}}_{\text{spin}}^{\text{ini}}$ is the position of a spin initiator.

ii. **Definition of information transfer distance d_i^{turn} in simulation experiments of collective turn**

In the context of collective turn, the estimation of information transfer distance d_i^{turn} is more complicated due to dynamic heading adjustments in response to the selected neighbor and collision avoidance among individuals. First of all, we introduced the definition of turning rank κ_i and turning delay t_i . To determine the turning rank κ_i of each individual, we calculated the time lag λ_{ij}^* based on the directional alignment function $\xi_{ij}(\lambda_t) = \langle \hat{\mathbf{v}}_i(t) \cdot \hat{\mathbf{v}}_j(t + \lambda_t) \rangle$ to determine the leader-follower relation between the individual i and j , where λ_{ij}^* is the value that makes the $\xi_{ij}(\lambda_t)$ reaches the maximum and the sign of λ_{ij}^* determines the leader or follower role of individual i . λ_{ij}^* is defined at subsection titled as “**Reconstructing the leader-follower network to quantify the leadership during collective U-turns**” in **Methods** section (see **main text: pages 12-13, lines 547-590**).

Then, we assigned a score w_{ij} to each individual based on the λ_{ij}^* of their neighbors. Specifically, we set $w_{ij} = 1$ if $\lambda_{ij}^* < 0$, indicating that individual i is the follower of individual j . Additionally, we set $w_{ij} = -1$ if $\lambda_{ij}^* > 0$, suggesting that individual i is the leader of individual j . Upon calculation of all pairs of individuals within the group, the total score of individual i is calculated as $W_i = \sum_{i \neq j} w_{ij}$. The smaller W_i means that individual i turns earlier than a larger number of other individuals in the group. Next, the turning rank κ_i is determined in ascending order of their W_i in the group. For example, the first individual to turn has the smallest W_i , which is assigned $\kappa_i = 1$. Conversely, the last turning individual is $\kappa_i = N$ as it has the largest W_i . N is the group size.

Once the turning rank κ_i is obtained for each individual, the turning delay t_i is calculated as $t_i = \sum_{\kappa_j < \kappa_i, \kappa_i > 1} (t_j + |\lambda_{ij}^*|) / (\kappa_i - 1)$ for individuals with $\kappa_i > 1$. t_i represents the time delay at which an individual begins to turn behind the turn initiator. Particularly, for the individual with the turning rank $\kappa_i = 1$ (i.e., turning initiator), we set $t_i = 0$. Finally, we defined the information transfer distance d_i^{turn} as the distance that information has traveled to the individual with turning rank κ_i . d_i^{turn} is then estimated as being proportional to the radius of the subgroup containing the individual whose turning delay less than t_i , which is calculated as follows:

$$d_i^{\text{turn}} = l_r(t_i) / \rho,$$

where $l_r(t_i) = \frac{\max_{(j_1, j_2 \in \mathcal{H}_i)} \|\hat{\mathbf{x}}_{j_1} - \hat{\mathbf{x}}_{j_2}\|}{2}$ is the estimated radius of the subgroup $\mathcal{H}_i = j | t_j \leq t_i, j \in [1, \dots, N]$, which is calculated as the half of maximum relative distance within the subgroup \mathcal{H}_i . \mathcal{H}_i contains individuals whose turning delay is less than t_i . $\hat{\mathbf{x}}_{j_1}$ and $\hat{\mathbf{x}}_{j_2}$ represents the position vector of individual j_1 and individual j_2 , respectively. ρ is the density of the group. Due to the initial position of simulated robots are uniformly and the position of simulated robots cannot be abruptly changed, we set $\rho = 1$ in the calculation of d_i^{turn} to facilitate the straightforward understanding of information transfer distance d_i^{turn} in collective turn. In addition, the above calculation of turning rank, turning delay and information transfer distance d_i^{turn} have been widely adopted to understand the process of information transfer in starling and jackdaw flocks, for example,

- Ling H, Mclvor G E, Westley J, et al. Collective turns in jackdaw flocks: kinematics and information transfer, *Journal of the Royal Society Interface*, 2019, 16(159): 20190450.
- Attanasi A, Cavagna A, Del Castello L, et al. Information transfer and behavioural inertia in starling flocks, *Nature physics*, 2014, 10(9): 691-696.

Particularly, we recognized that the mathematical notation of distance r in correlation function $C(r)$ is similar to the notation of turning rank in the previous manuscript. In the revised manuscript, we have rephrased the mathematical notation of turning rank as κ_i to distinguish with the notation of distance r in correlation function $C(r)$.

Indeed, our primary aim in defining the information transfer distance (d_i^{spin} or d_i^{turn}) and other intermediate variables (e.g., spinning lag, spinning rank s_i , turning delay t_i and turning rank κ_i) is to calculate the information transfer speed V_s , which estimates the rate at which behavioral changes or information, such as the spinning state or direction of an informed individual, propagate through a group. For example, V_s is estimated by the slope of curves formed by spinning lag and d_i^{spin} in the context of collective spin, while the V_s is estimated by the slope of curves formed by turning delay and d_i^{turn} defined in collective turn experiments. The relationships of these key factors in calculating V_s for simulation experiments of collective spin and collective turn are shown in **Fig.R7**.

While we had previously provided definitions of these important factors or variables in the previous version of Supplementary Information, we acknowledged that such writing style has impeded the readers' understanding of these definitions, particularly with respect to the definition of information transfer speed. This further makes it difficult to understand our findings of BOC in this study. Additionally, we recognized that sequentially presenting these definitions in the **Results** section could further complicate the readability of the manuscript. Hence, given that our primary objective is to calculate the information transfer speed V_s , we have integrated detailed definitions of information transfer speed V_s and other intermediate variables (e.g., information transfer distance $d_i^{\text{spin}}, d_i^{\text{turn}}$, spinning lag, spinning rank, turning rank and turning delay) in **Methods** section with a subtitle as “**Estimation of information transfer speed**” to ensure that the focus remains on the key indicators of information transfer speed V_s but also clarify other methodology and formulas (see **main text: pages 15-16, lines 673-719**):

“The information transfer speed V_s measures how quickly behavioral changes or information, such as the spinning state (or behavior) and direction of an informed individual, spreads within a group, reflecting the efficiency of information transfer among individuals. Due to differences in the reaction rules governing collective spin and collective turn, the estimation method of information transfer speed V_s is adapted for each of these simulation experiments.

In the collective spin simulation experiments, the estimation of V_s begins with defining the spinning lag and spinning rank s_i . The spinning lag is defined as the time delay by which an individual's start of spinning lags behind that of the spin initiator. The spinning rank s_i is determined based on the value of spinning lag, where a longer lag corresponds to a higher spinning rank. For example, the spin initiator has $s_i = 1$, whereas the last individual to start spin spinning has $s_i = N$. Then, we defined the information transfer distance d_i as the distance over spin which information has traveled to reach the individual with spinning rank s_i , which is estimated as the relative Euclidean distance between individual i and the spin initiator, i.e., $d_i = \|\hat{\mathbf{x}}_i - \hat{\mathbf{x}}_{\text{spin}}^{\text{ini}}\|$, where the $\hat{\mathbf{x}}_{\text{spin}}^{\text{ini}}$ is the position of the spin initiator. Finally, based on the computational method adopted in Ref. 65, information transfer speed V_s in the collective spin experiment is estimated as the slope of the spin curve formed by the spinning lag and the corresponding d_i^{spin} (see Supplementary Fig. 27 for the curves in collective spin simulation experiments).

For the collective turn simulation experiments, first, we followed the method proposed in Ref. 65 to calculate the turning rank κ_i of each individual based on the leader-follower relationship among all pairs of individuals, which are determined by the sign of time lag λ_{ij}^* (see detailed calculation of λ_{ij}^* in Eq.(2) in Methods). Specifically, to obtain the turning rank κ_i , we first assigned scores w_{ij} to individual i based on the λ_{ij}^* of its neighbor j . In particular, we set $w_{ij} = 1$ if $\lambda_{ij}^* < 0$, indicating that individual i is the follower of individual j , whereas we set $w_{ij} = -1$ if $\lambda_{ij}^* > 0$, suggesting that individual i is the leader of individual j . Then, the total score of individual i is calculated as $W_i = \sum_{i \neq j} w_{ij}$. Consequently, a smaller W_i means that individual i turns earlier than a larger number of other individuals in the group. As a result, each individual's turning rank κ_i is determined in ascending order of their W_i within the group. For example, the first individual to turn has the smallest W_i , which is assigned $\kappa_i = 1$. Conversely, the last turning individual is assigned $\kappa_i = N$, as it has the largest W_i . Once the turning rank κ_i is obtained for each individual, the corresponding turning delay t_i is calculated as $t_i = \sum_{\kappa_j < \kappa_i, \kappa_i > 1} (t_j + |\lambda_{ij}^*|) / (\kappa_i - 1)$, which represents the time delay at which the individual begins to turn relative to the turn initiator. Particularly, for the individual with turning rank $\kappa_i = 1$, we set its turning delay as $t_i = 0$.

Next, we defined d_i^{turn} as the distance that information has traveled to the individual with turning rank κ_i (i.e., within the turning delay t_i). Following the computational approach proposed turn in Ref. 66, we estimated the information transfer distance d_i^{turn} as being proportional to the radius of the subgroup containing the individual with a turning delay less than t_i . The information transfer distance in the collective turn simulation experiments is estimated as:

$$d_i^{\text{turn}} = l_r(t_i) / \rho$$

where $l_r(t_i) = \frac{\max_{(j_1, j_2 \in \mathcal{H}_i)} \|\hat{\mathbf{x}}_{j_1} - \hat{\mathbf{x}}_{j_2}\|}{2}$ is the estimated radius of subgroup $\mathcal{H}_i = \{j | t_j \leq t_i, j \in [1, \dots, N]\}$, which is calculated as half of the maximum relative distance within the subgroup \mathcal{H}_i . $\hat{\mathbf{x}}_{j_1}$ and $\hat{\mathbf{x}}_{j_2}$ represents the position vector of individual j_1 and individual j_2 , respectively. ρ is the group density. Given the uniform initial distribution of individuals in simulation experiments and no abrupt positional changes among simulated robots, we set $\rho = 1$ to facilitate a straightforward understanding of the d_i^{turn} . Finally, the V_s in the collective turn simulation experiments is estimated by fitting the slope of the curve formed by the turning delay t_i and the corresponding d_i^{turn} (see Supplementary Fig. 28 for the curves in collective turn simulation experiments)."

Additionally, we have removed the definitions of these intermediate variables (e.g., information transfer distance) in **Results** section, as they are not directly related to the interpretation of our findings. In particular, recognizing the concept of information transfer speed is inherently general, we have chosen to use a single mathematical symbol V_s , to represent it throughout the manuscript, which is intended to minimize the introduction of additional mathematical symbols and unnecessary complexity or confusion in **Results** section. Moreover, we have also briefly clarified the definition of information transfer speed V_s in **Results** section before referencing it, helping readers to better grasp the concept of V_s . For example,

- clarify the concept of information transfer speed in the context of collective spin in **Results** section before we referred to (see **main text: page 6, lines 283-287**):

“Moreover, we analyzed the information transfer speed V_s governed by BOC-based interaction in simulation experiments of collective spin, which is a measure of how quickly a new state or information, is disseminated throughout a group. In the context of collective spin, V_s evaluates how fast the spinning state (or behavior) triggered by one initiator spreads in the group. The detailed computation process of V_s in simulation experiments of collective spin can be found in Methods.”

- clarify the concept of information transfer speed in the context of collective turn in **Results** section before we referred to (see **main text: page 8, lines 362-365**):

“Particularly, in the context of collective turn, V_s evaluates how fast the direction of one informed individual spreads within the group (see Methods for the detailed definition of V_s in simulation experiments of collective turn).”

In addition to the issues raised by the reviewer regarding undefined or partially defined concepts, we have found additional issues upon a thorough review of the manuscript, such as the definition of response accuracy, responsiveness, baseline swarm model with random-based interaction, etc. While we have provided these definitions in the previous Supplementary Information, we acknowledged that such a writing style may not facilitate easy reading and understanding of these definitions for readers. To clarify these definitions of these important concepts or methods while maintaining the readability of our manuscript, we have opted to provide concise definitions and explanations of these concepts in **Results** section before we used them, while presenting the detailed descriptions and mathematical definitions in the **Methods** section. For example,

- clarify the definition of response accuracy and responsiveness in **Results** section (see **main text: pages 7-8, lines 338-345**):

“In addition, to compare BOC-based interaction with random-based interaction quantitatively, we analyzed the response accuracy $\delta_{resp}(t)$ and responsiveness R in collective turn experiments (see Methods for detailed mathematical definitions). On the one hand, response accuracy $\delta_{resp}(t)$ measures how closely the group aligns with the direction of an informed individual. In particular, $\delta_{resp}(t) = 1$ indicates the most accurate response, whereas $\delta_{resp}(t) = -1$ signifies the worst response. On the other hand, responsiveness R evaluates how quickly and efficiently the group responds to the direction change initiated by the informed individual. Notably, the lower the R , the higher the responsiveness of the group.”

- clarify the mathematical definition of response accuracy and responsiveness in **Methods** section (see **main text: page 16, lines 720-736**):

“To evaluate the quality of the collective response in both the simulations and the robotic experiments of collective turn, we defined the response accuracy $\delta_{resp}(t)$ as follows:

$$\delta_{resp}(t) = \frac{1}{N} \sum_{i=1}^N (\hat{\mathbf{v}}_i(t) \cdot \hat{\mathbf{n}}) \in [-1,1],$$

where $\hat{\mathbf{v}}_i(t)$ is the velocity of individual i at time t . $\hat{\mathbf{n}}$ represents the direction of the informed individual, which also corresponds to the new direction that triggers the collective turn. $\delta_{resp}(t) = 1$ indicates that the group successfully responds to the informed individual and moves in the same direction as the informed individual does. Conversely, when the group moves in a direction opposite to the informed individual, $\delta_{resp}(t) = -1$.

To evaluate the efficiency of the group response to the informed individual, we followed the similar indicator used in the Ref 4, i.e., the responsiveness. The responsiveness R is defined as follows:

$$R = \frac{1}{t_1 - t_0} \int_{t_0}^{t_1} (1 - \hat{\mathbf{V}}(t) \cdot \hat{\mathbf{n}}) dt,$$

where the duration of the collective response is from the t_0 to t_1 . The group moving direction is $\hat{\mathbf{V}}(t) = \frac{1}{N} \sum_{i=1}^N \hat{\mathbf{v}}_i(t)$. Notably, R is the cumulative evaluation of the group response. The lower the value of R , the higher the group’s responsiveness. The value of R ranges from 0 to 2. $R = 0$ means that all individuals follow the new direction $\hat{\mathbf{n}}$ without any delay during the response process, whereas $R = 2$ suggests the worst response to the informed individual, i.e., the group moves in the direction opposite to $\hat{\mathbf{n}}$.”

- clarify the definition of random-based interaction in **Results** section (see **main text: page 6, lines 258-260**):

“In the random-based interaction, an individual randomly selects a neighbor to react within the perception range irrespective of the neighbors’ BOC (see Methods for the detailed description of the baseline swarm model with random-based interaction).”

- clarify the detailed description of baseline swarm model with random-based interaction in **Methods** section (see **main text: page 14, lines 634-652**):

“To show the significant role of BOC in facilitating the emergence of scale-free correlation, we involved the random-based interaction as a baseline mechanism in both simulation experiments and robotic experiments. Specifically, in the random-based interaction, the focal individual selects one neighbor to react based on a uniform random distribution within the perceptual range R_{visual} , which is a fair and unbiased baseline interaction mechanism. By comparing with the random-based interaction, on the one hand, we can show the effectiveness of BOC-based interaction in information transfer. On the other hand, we can explore whether the BOC is a critical factor responsible for the emergence of scale-free correlation.

Similar to the swarm model with BOC-based interaction, we incorporated the random-based interaction into a self-propelled model to conduct the collective spin and collective turn simulation experiments. For the collective spin simulation experiments, the motion state (or behavior) of the focal individual i depends on the current state (or behavior) of the randomly selected neighbor. For the collective turn in simulation and robotic experiments, individual i

updates its velocity by aligning with the randomly selected neighbor, calculated as $\hat{v}_i(t+1) = \hat{v}_i(t) + k_a \cdot \hat{v}_j(t)$, where the neighbor j is randomly selected from the distribution Uniform (1, N). Unless otherwise specified, the parameter selection in the random-based interaction is the same as the swarm model with BOC-based interaction, which can be found in Supplementary Tables 1-3 for different experiments.”

- clarify the definition of pinhole camera model in **Results** section (see **main text: page 9, lines 393-396**):

“On the other hand, we adopt the pinhole camera model to simulate vision-based sensing of robots, which is a simplified representation of how a camera projects a 3D scene onto a 2D image plane (see Methods for detailed definition of pinhole camera model).”

- clarify the mathematical definition of pinhole camera model in **Methods** section (see **main text: page 17, lines 764-767**):

“The pinhole camera model describes the mathematical relationship between the coordinates of a point $p = (p_x, p_y, p_z)$ in three-dimensional space and its projection onto the image plane. The position on the image plane is calculated as $(u_x = f \cdot \frac{p_x}{p_z}, u_y = f \cdot \frac{p_y}{p_z})$, where f represents the focal length of the camera.”

- clarify the effect and role of distance, bearing change-based motion salience in the analysis of U-turn fish data in **Results** section (see **main text: page 5, lines 203-210**):

“Following the same procedures used to calculate the BOC-based motion salience (the detailed procedure can be found in Fig.2a-f or Supplementary Fig. 31), we derived the corresponding distance-based motion salience and bearing change-based motion salience. On the one hand, the correlation analysis with distance-based motion salience examines whether neighbors closer to the focal individual exert a greater influence on shaping leadership during U-turns. On the other hand, the correlation analysis with bearing change-based motion salience investigates whether the neighbors with greater relative position changes tend to be leaders within the group.”

- clarify the Spearman correlation analysis between the leadership and the effect of frontal preference in **Results** section (see **main text: page 5, lines 194-196**):

“Through an examination of the sole correlation between leadership and the effect of frontal preference, i.e., omitting the influence of the neighbor’s BOC from Eq. (1) (described as $g_{ij}(T, \tau) = \sum_{t=T-\tau+1}^T fp(t)$),”

- clarify the relationship between the BOC-based interaction and swarm model with BOC-based interaction and add a brief description of the self-propelled model in **Results** section (see **main text: page 5, lines 229-236**):

“To explore how this bio-inspired mechanism drives the overall collective response, we introduced BOC-based interaction into the self-propelled model^{9,12}, a theoretical framework of swarm model used to simulate how individuals move based on certain local interaction mechanisms, which suggests that the motion decisions of individuals are guided by the most influential neighbor characterized by BOC and the focal individual can adjust its motion by behavior imitation or velocity alignment with this selected neighbor to achieve the collective response (see Methods for detailed definitions of the swarm model with BOC-based interaction).”

- clarify the definition of the self-propelled model in **Methods** section (see **main text: page 13, lines 596-598**):

“The self-propelled model is a widely used theoretical framework in swarm dynamics to simulate the autonomous movement of individuals, which enables the simulation of collective spin and collective turn arising from BOC-based interaction.”

- clarify the effect of additional repulsive interactions in **Results** section (see **main text: page 7, lines 335-338**):

“These repulsive interactions occur when individuals come too close to each other, causing them to move apart gently (see Supplementary Sec. 7 for a detailed mathematical definition).”

- clarify the definition of the aspect ratio of individuals in **Results** section (see **main text: page 5, lines 216-219**):

“Moreover, it is noteworthy that the aspect ratio of the elliptical individual might be an influential factor in the experimental data analysis of real fish schools, which is defined as the ratio of the major axis length (parameter a shown in Fig. 1a)) to the minor axis length (parameter b shown in Fig. 1a)) of an ellipse, i.e., denoted as a/b .”

Point 1.7: Finally, the Discussion section needs to be drastically reduced to a concise discussion of the results (significance, limitations, possible developments etc.). As it stands it includes a recapitulations of some results, state of the art, and repetition of claims and considerations already made elsewhere in the manuscript.

Response: We thank Reviewer #1 for these constructive suggestions for **Discussion** Section. We fully agreed the reviewer noted, “the Discussion section needs to be drastically reduced to a concise discussion of the results (significance, limitations, possible developments etc.)”. Following the reviewer’s valuable comments, in the revised **Discussion** section, we begin with a brief restatement of our research aims and new information our work provides. Then, we elaborate on the significance of results from the perspective of the biological relevance and realistic factors in modeling BOC, the emergence of scale-free correlation in the absence of critical dynamics and the effectiveness and practicability of BOC in swarm robotics applications. Next, we discuss the limitations of this study, particularly the challenges posed from the field of control theory and engineering. Finally, we provide the possible developments and potential benefits to related research areas. In the following, we provided a rewritten **Discussion** section for each part:

- Statements of research aim and new information our work provides (see main text: pages 9-10, lines 433-440):**

“This study aimed to present a visual cue from the first-person perspective that could serve as sensory input to facilitate the emergence of collective responses in vision-based swarm models. It was hypothesized that body orientation change of neighbors might be a crucial visual cue in enhancing the efficiency of information transfer within groups. Based on a comprehensive research chain from biological data analysis to bionic mechanism modeling and further to swarm robotics application, we confirmed that BOC is not only associated with the emergence of leadership during U-turn behaviors in fish schools but also enables rapid information transfer in swarm robotic systems and gives rise to the non-trivial phenomenon of scale-free correlation.”

ii. Discussion of the significance of results (see main text: page 10, lines 441-473):

“Our work represents an effort to model collective behavior based on visual observations rather than explicit physical measurements, which is crucial for moving beyond swarm models derived from computational theories and handcrafted designs^{9,13,14,47}. In particular, we characterized each individual as a non-transparent ellipse to investigate BOC in both fish schools and artificial swarms. This further enabled us to involve several important realistic factors in local interactions, such as the aspect ratio of the individuals¹⁹ and visual occlusions^{48,49}. Additionally, we highlighted the significance of conducting a comprehensive research chain in extracting interaction mechanisms from animal groups, as it ensures such bio-inspired mechanisms (e.g., BOC-based interaction) are not only theoretically sound with biological plausibility but also applicable in real-world contexts.

Moreover, our work sheds new light on the emergence of scale-free correlation which is a macro-level phenomenon observed in various biological and artificial systems^{31,32,50,51}. The scale-free correlation suggests the velocity fluctuations among individuals are directionally aligned over longer distances in larger groups, which typically occurs in the critical regime of a self-organized system. However, it is important to note that BOC-based interaction exhibits scale-free correlation with generic parameters that are qualitatively consistent with those observed in biological swarms³¹ and facilitates rapid information transfer within the swarm. This further suggests the presence of scale-free correlation may be essential to the enhancement of collective responses within groups. In addition to our findings, researchers have also found the emergence of scale-free correlation in self-organized systems operating outside the critical regime. For example, Huepe et al. provided an alternative mechanism for generating this macro-level phenomenon based on positional interaction with the absence of critical dynamics⁵².

Furthermore, we believed that BOC stands out as a significant visual cue in the context of swarm robotics due to its sole dependence on the orientation change-induced maneuver, which provides immediate and explicit visual feedback on the movements of neighboring robots. In contrast to other cues, such as the neighbors’ bearing change, which requires additional time to accumulate velocity into displacement before identifying the neighbor’s salient movement. Notably, as swarm robotics expect to make decisions and actions based on raw and immediate observations through simple local sensing devices^{53,54}, this advantage becomes particularly valuable because BOC can be effectively estimated via onboard cameras through standard computer vision techniques⁵⁵⁻⁵⁸. For example, robots can effectively estimate the magnitude of the BOC by calculating changes in the bounding box area over consecutive frames obtained with onboard optical cameras (see Supplementary Sec. 8 and Supplementary Fig. 21 for detailed information).”

iii. Discussion of limitations in this study (see main text: page 10, lines 472-482):

“Although the BOC-based interaction is a novel mechanism with considerable potential in swarm robotics, it still remains much challenging work from the perspective of control theory and engineering, such as addressing the consensus and stability issues of BOC-based interaction^{59,60}, mitigating effects of noise and time delays in real-world applications^{61,62}, etc. Additionally, we openly admitted that elucidating the role of BOC in collective response addresses only one aspect of collective behavior modeling in this work, i.e., determining which visual cues individuals should focus on during collective response. Another fundamental issue that warrants further investigation is how biological individuals integrate BOC from visual neurobiological circuits² and convert them into motion to achieve collective responses^{63,64}.”

iv. Discussion of possible developments and potential benefits of this study (see main text: page 10, lines 483-489):

“Overall, this study offers an essential interaction mechanism for swarm robotics to perform complex and sophisticated collective tasks^{5,65} that demand high maneuverability. Moreover, our results provide valuable insights not only for biologists and researchers in complex systems but also for computer scientists and engineers specializing in swarm robotics. In particular, our comprehensive research chain in this work encourages interdisciplinary collaboration across biology, physics, and engineering for the development of high-performance and cost-effective swarm robotic systems.”

Finally, we thank Reviewer #1 again for her/his very insightful and constructive comments and appreciate her/his carefully reviewing our manuscript. We hope our responses above have addressed those very legitimate issues/concerns in a satisfactory manner.

Response to Reviewer #2

Dear authors,

after reviewing the changes you made to the manuscript, I am exceptionally impressed and satisfied by the extensiveness of the revision. In my opinion, the article is now greatly improved, also in response to the other reviewer's comments. I therefore I have no more objections to publication.

Reviewer #2 (Remarks on code availability):

I thank the authors for providing much more code now for reproducing the work.

Response: We thank Reviewer #2 very much for reviewing our paper again. We are very pleased to know that he/she is now happy with the revised version.

Response to Reviewer #3

In this paper, authors propose an algorithm based on body orientation change of neighbors. They implemented this algorithm both numerically and using simulated robots. This is the second round of the review process. I had several concerns about the paper and implementation with simulated robots. All my concerns are satisfied and I think that the paper is good enough to be published.

Reviewer #3 (Remarks on code availability):

I haven't checked the code.

Response: We thank Reviewer #3 very much for reviewing our paper again. We are very pleased to know that she/he is now happy with the revised version.

Response Figure

Fig.R1 | The definition of the correlation length based on the curve of correlation function with the increasing distance r . **a**, the dot product between two individual's velocity fluctuation $\hat{\mathbf{u}}_i$ and $\hat{\mathbf{u}}_j$, where $\hat{\mathbf{u}}_i = \hat{\mathbf{v}}_i - \frac{1}{N} \sum_{k=1}^N \hat{\mathbf{v}}_k$ and $\hat{\mathbf{v}}_i$ is the velocity of individual i . When these two vectors are aligned with each other, we have $\hat{\mathbf{u}}_i \cdot \hat{\mathbf{u}}_j = 1$. When the deviation between these two vectors gradually increases, the dot product value $\hat{\mathbf{u}}_i \cdot \hat{\mathbf{u}}_j$ decreases. Particularly, when the two vectors are perpendicular to each other, $\hat{\mathbf{u}}_i \cdot \hat{\mathbf{u}}_j = 0$. Additionally, when two vectors of velocity fluctuations are completely opposite, the dot product between them is $\hat{\mathbf{u}}_i \cdot \hat{\mathbf{u}}_j = -1$. **b**, The correlation function of velocity fluctuations $C(r)$ as a function of distance r . $C(r)$ is calculated as $C(r) = \frac{\sum_{i,j=1}^N \hat{\mathbf{u}}_i \cdot \hat{\mathbf{u}}_j \delta(r-r_{ij})}{\sum_{i,j=1}^N \delta(r-r_{ij})}$, where the $\hat{\mathbf{u}}_i = \hat{\mathbf{v}}_i - \frac{1}{N} \sum_{k=1}^N \hat{\mathbf{v}}_k$ is the velocity fluctuation of individual i . In particular, r is a variable that represents the distance between two individuals, which can be varied to investigate how the directional correlation between velocity fluctuations evolves with increasing distance. $\delta(r-r_{ij})$ is a smoothed Dirac δ function used to select pairs of individuals at a mutual distance r . r_{ij} is the distance between two individuals. If the relative distance r_{ij} between two individuals is equal to the variable r , $\delta(r-r_{ij}) = 1$, otherwise, $\delta(r-r_{ij}) = 0$. High values of $C(r) > 0$ indicate the strong correlation in the velocity fluctuations among all individuals at a certain distance r . As r increases, the value of $C(r)$ steadily decreases and eventually crosses the x-axis ($C(r) < 0$), indicating that correlation in the velocity fluctuations among individuals exhibits the decaying transition from a strong correlation to a weak correlation. The correlation length is defined as the distance r_0 that makes $C(r_0) = 0$, which provides an efficient estimate of the size of the correlated domains.

Fig.R2 | Construction of BOC matrix and derivation of BOC-based motion salience. To get a BOC matrix, we start with the calculation of neighbors' BOC from the view of each individual based on the reconstruction of their visual field. For example, given a period trajectory of U-turn behavior (a), we reconstructed the visual field of individual 8 (b-c) and calculated the magnitude of its neighbors' BOC based on Eq. (1) in main text from $[2.22s, 2.82s]$ ($T = 2.82s, \tau = 0.6s$) (d). As a result, we obtained the eighth column of a BOC matrix (marked by red rectangular shown in e). After obtaining the neighbors' BOC of each individual, we could obtain a complete BOC matrix (e). Then, we calculated the column-wise average to obtain the BOC-based motion salience (f). From the mathematical standpoint, the BOC-based motion salience is a vector that contains how noticeable the movements of the focal individual i are perceived by other individuals in the group based on the BOC. Additionally, the BOC-based motion salience is normalized by its maximum value to ensure that the Spearman correlation analysis results are not skewed by differences in the data scales (g). Particularly, the trajectory and BOC matrix presented in panels (a) and (e) are identical to those shown in Fig. 2a-b of the main text.

Fig.R3 | Reconstruction of leader-follower network to derive the leadership of each individual within the group. **a**, The trajectory of individual 1 and individual 3 from $T = 2.22s$ to $T = 2.82s$ ($T = 2.82s, \tau = 0.6s$). The trajectory in panel (a) is drawn with a gradient color that increases with time. From this trajectory, we found that the individual 1 turns earlier than the individual 3. **b**, The curve of directional alignment function $\xi_{1,3}$ between individual 1 and individual 3 with the increasing time lag λ_t , which reaches its maximum at $0.46s$. This means that the individual 1 leads the movement of individual 3 by $0.46s$ ahead, which aligns with the leader-follower relationship shown in trajectory (a). **c**, After obtaining the leader-follower relations among all pairs of individuals, we could reconstruct the leader-follower network within $[2.22s, 2.82s]$. **d**, Based on the leader-follower network, we adopted the normalized out-distance of each node (or individual) in the network to characterize each fish's leadership according to the Eq. (3) in main text. The normalized out-distance signifies the hierarchical layer of each node (individual) in the leader-follower network. In such a hierarchical network, some individuals act as leaders, whereas others act as followers, with directed connections between individuals, i.e., leaders point to their followers. We leverage the number of connections with other nodes, namely, the out-distance (or the out-degree) of nodes in the directed network, to represent the leadership of the focal individual. For example, as depicted in panels (c-d), fish-1 occupies the top level in the leader-follower network, thus its leadership $L_1 = 1$.

Fig.R4 | The illustration of the spinning motion state (or behavior) in the simulation experiments of collective spin. The spinning motion state (or behavior) refers to a rotational movement in place. Since we used two-wheeled simulated robots in our simulation experiments, the spinning is achieved by controlling the simulated robots with a certain angular speed while maintaining a linear speed of zero.

Fig.R5 | The snapshots of collective spin in groups with BOC-based and random-based interaction from the beginning to the end. a, The snapshots of collective spin in a group with BOC-based interaction from $t = 25$ step to $t = 58$ step. **b,** The snapshots of collective spin in a group with random-based interaction from $t = 25$ step to $t = 58$ step. The activation time of the spin initiator is $t = 25$ step. The time that the spin imitator had just spun 2π is $t = 58$ step. The spinning lag is defined as the time delay by which an individual's spinning initiation lags behind that of the spin initiator.

Fig.R6 | The involvement of frontal preference and effect of the anisotropic parameter α on tuning the forward-oriented perception of BOC. **a**, the $fp(t) = \left(\frac{1+\hat{\mathbf{v}}_i(t)\cdot\hat{\mathbf{x}}_{ij}(t)}{2}\right)^\alpha$ quantifies the frontal preference on perceiving neighbor- j 's movement from the first-person perspective of the focal individual i , ranging in $[0, 1]$. It simulates that as the first-person view of neighbor- j moves from the front to the back, the visual perception of the focal individual i to the neighbor- j diminishes. The dot product of these two vectors $\hat{\mathbf{v}}_i \cdot \hat{\mathbf{x}}_{ij}$ measures the relative bearing of neighbor- j . As shown in **a**, if $\angle(\hat{\mathbf{v}}_i, \hat{\mathbf{x}}_{ij}) = 0$, i.e., the dot product of $\hat{\mathbf{v}}_i \cdot \hat{\mathbf{x}}_{ij} = 1$, it signifies that neighbor- j_1 locates directly in front of the focal individual- i , and the focal individual could perceive the neighbor- j_1 as the $(1 + \hat{\mathbf{v}}_i \cdot \hat{\mathbf{x}}_{ij})/2$ is 1 (neighbor- j_1 shown in **a**). Once the neighbor- j gradually moves from the front to back respect to the focal individual i , i.e., $\angle(\hat{\mathbf{v}}_i, \hat{\mathbf{x}}_{ij})$ goes from 0 to π or $-\pi$, the visual perception to the neighbor- j could diminish since $(1 + \hat{\mathbf{v}}_i \cdot \hat{\mathbf{x}}_{ij})/2$ decreases from 1 to 0 (Neighbor j_2, j_3, j_4 in **a**). By incorporating the $\left(\frac{1+\hat{\mathbf{v}}_i(t)\cdot\hat{\mathbf{x}}_{ij}(t)}{2}\right)^\alpha$ with the perception of BOC, we could emulate the anisotropic perception of the neighbor's body orientation change ($g_{ij}(T, \tau)$ defined in Eq. (1) of the main text). **b**, to tune the effect of frontal preference, we involve an anisotropic parameter α , i.e., $fp(t) = \left(\frac{1+\hat{\mathbf{v}}_i(t)\cdot\hat{\mathbf{x}}_{ij}(t)}{2}\right)^\alpha$. When $\alpha = 0$, the focal individual i ignores the forward-oriented perception as the $fp(t)$ always equal to 1 (the green line shown in **b**), which means that the perception of BOC is not relevant to the relative positions of neighbors. In particular, increasing α ($\alpha \geq 1$) makes Eq. (1) in main text enhances the anisotropic effect of visual perception, which causes individuals to gradually narrow the perception of nearby neighbors toward their frontal vision. For example, as shown in **b**, if $\alpha = 10$, it means that the perception of BOC $g_{ij}(T, \tau) \approx 0$ when the relative positions of neighbors are out of the visual sight ($-\pi/2, \pi/2$).

Fig.R7 | The relationships of key intermediate variables in calculating the information transfer speed V_s for both simulation experiments of collective spin and collective turn. a, To determine the information transfer speed in collective spin, we first define the spinning lag and then calculate the spinning rank based on it. Subsequently, the information transfer distance d_i^{spin} is defined as the distance when the information reaches the individual with a certain spinning rank. Finally, the slope of the curve formed by the information transfer distance d_i^{spin} and the corresponding spinning lag is used to define the information transfer speed V_s in the collective spin simulation experiments. **b,** For the simulation experiments of collective turn, we first define the turning rank based on the leader-follower relationship among individuals, then define the turning delay accordingly, and further calculate the information transfer distance d_i^{turn} corresponding to a certain turning rank based on Eq. (9) in main text. Finally, the information transfer speed V_s in collective turn is defined as the slope of the curve formed by the information transfer distance d_i^{turn} and turning delay.